# Assessment of pre-industrial to present-day anthropogenic climate forcing in UKESM1

Fiona M. O'Connor[1], N. Luke Abraham[2,3], Mohit Dalvi[1], Gerd Folberth[1], Paul Griffiths[2,3], Catherine Hardacre[1], Ben T. Johnson[1], Ron Kahana[1], James Keeble[2,3], Byeonghyeon Kim[4], Olaf Morgenstern[5], Jane P. Mulcahy[1], Mark Richardson[6], Eddy Robertson[1], Jeongbyn Seo[4], Sungbo Shim[4], Joao C. Teixeira[1], Steven Turnock[1], Jonny Williams[5], Andy Wiltshire[1], and Guang Zeng[5]

[1]Met Office, Exeter, United Kingdom
[2]National Centre for Atmospheric Science, University of Cambridge, United Kingdom
[3]Department of Chemistry, University of Cambridge, United Kingdom
[4]National Institute of Meteorological Sciences, Seogwipo-si, Jeju-do, Korea
[5]National Institute, for Water and Atmospheric Research, Wellington, New Zealand
[6]Centre for Environmental Modelling And Computation, University of Leeds, United Kingdom

*Correspondence to*: Fiona M. O'Connor (fiona.oconnor@metoffice.gov.uk)

**Abstract.** Quantifying forcings from anthropogenic perturbations to the Earth System (ES) is important for understanding changes in climate since the pre-industrial (PI) period. Here, we quantify and analyse a wide range of present-day (PD) anthropogenic effective radiative forcings (ERFs) with the UK's Earth System Model (ESM), UKESM1, following the protocols defined by the Radiative Forcing Model Intercomparison Project (RFMIP) and the Aerosol and Chemistry Model Intercomparison Project (AerChemMIP). In particular, quantifying ERFs that include rapid adjustments within a full ESM enables the role of various climate-chemistry-aerosol-cloud interactions to be investigated.

Global mean ERFs for the PD (year 2014) relative to the PI (year 1850) period for carbon dioxide ($CO_2$), nitrous oxide ($N_2O$), ozone-depleting substances (ODSs), and methane ($CH_4$) are $1.89 \pm 0.04$, $0.25 \pm 0.04$, $-0.18 \pm 0.04$, and $0.97 \pm 0.04$ W m$^{-2}$, respectively. The total greenhouse gas (GHG) ERF is $2.92 \pm 0.04$ W m$^{-2}$.

UKESM1 has an aerosol ERF of $-1.09 \pm 0.04$ W m$^{-2}$. A relatively strong negative forcing from aerosol-cloud interactions (aci) and a small negative instantaneous forcing from aerosol-radiation interactions (ari) from sulphate and organic carbon (OC) are partially offset by a substantial forcing from black carbon (BC) absorption. Internal mixing and chemical interactions imply that neither the forcing from ari nor aci is linear, making the aerosol ERF less than the sum of the individual speciated aerosol ERFs.

Ozone ($O_3$) precursor gases consisting of volatile organic compounds (VOCs), carbon monoxide (CO), and nitrogen oxides (NOx) but excluding $CH_4$, exert a positive radiative forcing due to increases in $O_3$. However, they also lead to oxidant changes,

which in turn cause an indirect aerosol ERF. The net effect is that the ERF from PD-PI changes in NOx emissions is negligible at $0.03 \pm 0.04$ W m$^{-2}$, while the ERF from changes in VOC and CO emissions is $0.33 \pm 0.04$ W m$^{-2}$. Together, aerosol and O$_3$ precursors (called near-term climate forcers (NTCFs) in the context of AerChemMIP) exert an ERF of $-1.03 \pm 0.04$ W m$^{-2}$, mainly due to changes in the cloud radiative effect (CRE). There is also a negative ERF from land use change ($-0.17 \pm 0.04$ W m$^{-2}$). When adjusted from year 1850 to 1700, it is more negative than the range of previous estimates, and is most likely due to too strong an albedo response. In combination, the net anthropogenic ERF ($1.76 \pm 0.04$ W m$^{-2}$) is consistent with other estimates.

By including interactions between GHGs, stratospheric and tropospheric O$_3$, aerosols, and clouds, this work demonstrates the importance of ES interactions when quantifying ERFs. It also suggests that rapid adjustments need to include chemical as well as physical adjustments to fully account for complex ES interactions.

## 1 Introduction

In order to have a quantitative understanding of past and future climate change, and attribute climate change and its impacts to different anthropogenic and natural drivers, it is important to have a process-based understanding of critical aspects of the pathway from anthropogenic (or natural) activity through to climate response and its impacts. A recent international effort for understanding climate change is the 6th Coupled Model Intercomparison Project (CMIP6; Eyring et al., 2016), which designs experiments and distributes data from multi-model simulations. These simulations, with state-of-the-art climate models or Earth System Models (ESMs), are aimed at addressing climate science questions directly or via dedicated CMIP6-endorsed model intercomparison projects (MIPs) such as the Aerosol and Chemistry Model Intercomparison Project (AerChemMIP; Collins et al., 2017). An important part of this cause-effect chain from activity to climate response, mediated through the atmosphere and the land surface, is quantifying changes to the Earth's radiation budget, often termed radiative forcing (RF). RF is a direct measure of the response in the Earth's radiation budget by changes in anthropogenic (or natural) activities.

Successive assessment reports of the Intergovernmental Panel on Climate Change (IPCC) have used the concept of RF as a metric to quantify the effects of different anthropogenic and natural drivers on the Earth's radiation balance. For this purpose, RF, or more precisely, the stratospherically-adjusted RF (SARF) is defined at the tropopause (Myhre et al., 2013a) as:

$$RF \ = \ IRF \ + \ A_{strattemp}, \tag{1}$$

where IRF is the instantaneous radiative forcing and $A_{strattemp}$ is the additional change in the net downward radiative fluxes at the tropopause solely due to stratospheric temperature adjustment (Hansen et al., 1997), while holding all other variables fixed. Including the stratospheric temperature adjustment can significantly affect the magnitude of a forcing (e.g., Smith et al., 2018)

and even change the sign of the forcing in the case of ozone ($O_3$) depletion (Shine et al., 1995). As a result, RF rather than IRF is a better predictor of the drivers of global mean temperature response. RF is a concept, based around energy budget analyses, that has split perturbations to the Earth's radiative balance from climate response (Boucher et al., 2013; Myhre et al., 2013a; Sherwood et al., 2015). It has been used extensively to evaluate and compare the strength of various forcings, both anthropogenic and natural, affecting the Earth's radiation balance and hence, their contribution to climate change (e.g., Hansen

et al., 1997; Shine and Forster, 1999). However, despite the extensive use of RF as a metric for climate change, it is often calculated inconsistently between the different drivers of climate change (e.g., Myhre et al., 2013a). Participating models in the 5th Coupled Model Intercomparison Project (CMIP5; Taylor et al., 2012) can show large differences in $CO_2$ forcing (Andrews et al., 2012a; Forster et al., 2013) due to model diversity and/or calculation method. For example, the RF attributed to long-lived greenhouse gases (LLGHGs) is typically based on changes in observed concentrations between the pre-industrial

(PI) and the present-day (PD) periods and uses line-by-line radiative transfer calculations and/or simple, yet justified, expressions for RF based on, e.g., Myhre et al. (1998) and Ramaswamy et al. (2001). These expressions have been updated for some LLGHGs recently (Etminan et al., 2016). However, the observed concentrations themselves may be subject to biogeochemical feedbacks (e.g., Arneth et al., 2010; O'Connor et al., 2010).

In contrast to the quantification of the LLGHG RF, the RF from PD-PI changes in tropospheric $O_3$ (Stevenson et al., 2013) is based solely on models. It has been calculated using the ensemble of models (Young et al., 2013) participating in the Atmospheric Chemistry and Climate Model Intercomparison Project (ACCMIP; Lamarque et al., 2013) providing input to offline radiative transfer models (e.g., Edwards and Slingo, 1996). The simulations used the corresponding sea surface temperatures (SSTs) and sea ice (SI) conditions for the time periods of interest (PI and PD), therefore allowing some climate

response and feedbacks at the PD, implying that the resulting estimate is not consistent with the RF definition. It does not fit into the simple forcing-feedback concept, whereby feedbacks are related to global mean temperature change and forcings are not (Sherwood et al., 2015). There are also additional uncertainties associated with the estimate of tropospheric $O_3$ RF due to the lack of robust and reliable observational constraints for PI $O_3$ concentrations (e.g., Stevenson et al., 2013), the diversity in modelled PD tropospheric $O_3$ burden across models (e.g., Young et al., 2013; Young et al., 2018), uncertainties in historical

emissions of $O_3$ precursors, and the apparent inability of current state-of-the-art chemistry-climate models to replicate near-recent observed trends in tropospheric $O_3$ (Parrish et al., 2014; Young et al., 2018) although recently, isotopic measurements seem to corroborate the modelled trends (Yeung et al., 2019). Other uncertainties in Stevenson et al. (2013) arise from neglecting the change in $O_3$ in the lower stratosphere attributable to changes in $O_3$ precursors and the contribution from stratospheric $O_3$ depletion on the modelled changes in tropospheric $O_3$ (e.g., Søvde et al., 2011; 2012). Despite these difficulties

in estimating tropospheric $O_3$ RF, the even larger uncertainty in aerosol forcing (Myhre et al., 2013a; Bellouin et al., 2020) accounts for the majority of the uncertainty in the total anthropogenic forcing.

Aerosol forcing involves a wide range of physical processes. These include (i) direct changes to the radiation budget through scattering and absorption of both shortwave (SW) and longwave (LW) radiation (e.g., Haywood and Boucher, 2000), (ii) indirect impacts on the radiation budget by changing the microphysical properties of clouds (Twomey et al., 1977), and (iii) changes in the distribution of cloud cover or condensate that follow on from perturbations in cloud microphysics (Albrecht 1989) or radiative heating by aerosols (Hansen et al., 1997). Direct aerosol RF can be calculated using offline radiative transfer models in a similar manner to greenhouse gas (GHG) and $O_3$ forcing, whereas assessing impacts of aerosols on clouds requires simulations in atmospheric models. The 5th assessment report (AR5) of the IPCC recommended the effective radiative forcing (ERF) framework as a suitable metric for assessing the overall aerosol forcing as it enables the more complex cloud impacts to be evaluated as part of the climate's rapid adjustments (RAs) (Myhre et al., 2013a). To simplify terminology, AR5 also made a clear distinction between components of the forcing driven by aerosol-radiation interactions (ari; i.e., the direct or IRF) and aerosol-cloud interactions (aci) (that include all indirect or semi-direct cloud-related forcings). Despite wide-ranging and on-going research, the role of aerosols remains the leading source of uncertainty in estimates of climate forcing, due to the difficulty in constraining the sensitivity of clouds to changing microphysical processes (Bellouin et al., 2020).

In the case of land use, RF estimates have been made using single general circulation model (GCM) simulations with a double-call to the radiation scheme (e.g., Betts et al., 2007) or by comparing paired simulations that include RAs (e.g., Andrews et al., 2016). However, the choice of RF calculation is not the major source of differences in RF estimates. Similar to $O_3$, uncertainty in PI land cover is a major source of uncertainty in land use RF (e.g., de Noblet-Ducoudré et al., 2012). Historically, deforestation has been the dominant type of land use change, and this causes a positive RF due to increased carbon dioxide ($CO_2$) emissions and a negative RF due to increased surface albedo. Here, we include the effects of land use $CO_2$ emissions in the $CO_2$ ERF estimates and the land use ERF is due to biophysical changes, predominately albedo. Deforestation has a much larger effect on albedo in snowy regions and model biases in snow cover also contribute to uncertainty in land use RF (Pitman et al., 2011). Land use RF estimates also vary due to different time periods being considered (Myhre et al., 2013a), because, unlike many other forcing agents, there was substantial land use change before the industrial revolution.

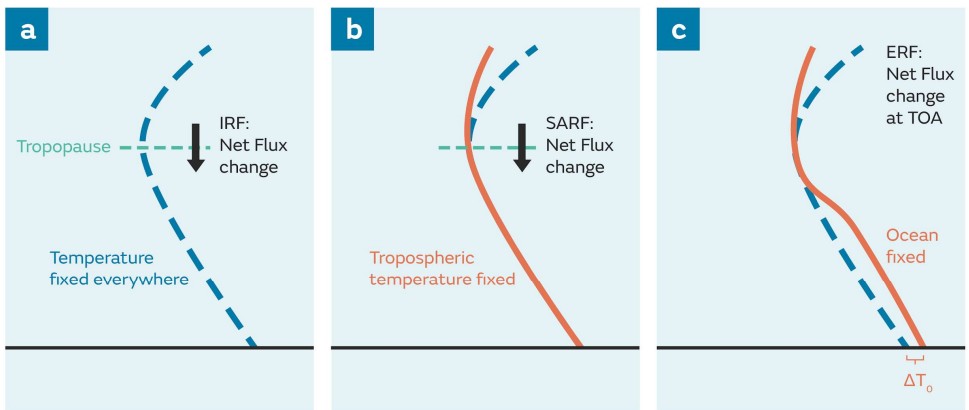

**Figure 1: Vertical profiles of temperature, showing the different definitions of radiative forcing: a) instantaneous radiative forcing (IRF), b) stratospherically-adjusted radiative forcing (SARF), and c) effective radiative forcing (ERF). IRF and SARF are defined at the tropopause whereas ERF is defined at the top of atmosphere (TOA). Adapted from Fig. 8.1 of the IPCC AR5 (Myhre et al., 2013a), which in turn was updated from Hansen et al. (2005).**

As indicated previously, the IPCC AR5 recommended an extension to the definition of RF to include RAs other than the stratospheric temperature adjustment. This updated definition, the ERF, is defined at the top of the atmosphere (TOA), following Chung and Soden (2015), as:

$$\text{ERF} = \text{IRF} + \sum_i A_i \ , \tag{2}$$


where IRF is now defined at the TOA and $A_i$ is a RA in the atmosphere or over the land surface that alters the net downward radiative flux at the TOA either positively or negatively. These RAs include changes in stratospheric temperatures (as included in the definition of RF or SARF above; Eqn. (1)) as well as adjustments such as tropospheric temperatures, water vapour, clouds, land surface temperature, and land surface albedo, as examples, but global ocean conditions remain unchanged. When

comparing the ERF and IRF for black carbon (BC), as an example, RAs lead to the ERF being half that of its IRF (Stjern et al., 2017; Smith et al., 2018). On the other hand, Xia et al. (2016) found in their model that cloud and sea ice adjustments driven by stratospheric $O_3$ recovery included in the ERF definition lead to the ERF being different in both sign and magnitude from the SARF. And when comparing different forcing metrics, Hansen et al. (2005) found that the efficacy of a climate forcing (i.e., the change in global mean temperature relative to that from $CO_2$ for an equivalent forcing at the TOA or the

tropopause) is less sensitive to the forcing agent and closer to 1 for ERFs than for IRFs or SARFs. Although the uncertainty associated with an ERF tends to be larger than that of RF, it is more representative of the climate response than the traditional RF (Hansen et al., 2005; Forster et al., 2016). As a result, it is now the preferred metric of choice for ranking the drivers of climate change (Boucher et al., 2013; Forster et al., 2016).

The SARF and ERF differ in where the change in radiative fluxes is diagnosed. The SARF, diagnosed at the tropopause, requires the tropopause to be defined but the ERF has the advantage of being diagnosed at the TOA, with no need for a tropopause. While the SARF at the tropopause and TOA are, by definition, identical, this is not the case for the ERF; climate forcings can lead to adjustments in stratospheric circulation and therefore changes to dynamical heating above the tropopause. Changes in dynamical heating are thus balanced by radiative divergence across the stratosphere, explaining the change in net

fluxes between the tropopause and TOA.

    The use of ERF as a metric also offers the advantage that it can be readily calculated using a pair of parallel simulations with standard model TOA radiative flux diagnostics (Forster et al., 2016) albeit with a requirement to run for relatively long periods (~30 years) to reduce the uncertainty associated with meteorological variability (e.g., Shindell et al., 2013a). Figure 1 illustrates

the various definitions of RF: IRF, SARF, and ERF. An overview of the historical evolution of the RF concept, its quantification for different forcing agents, and applications of RF can be found in Ramaswamy et al. (2019).

    The aim of the current study is to quantify PD (year 2014) ERF from anthropogenic drivers of climate change with an atmosphere-only configuration of an ESM. Using the experimental protocol recommended for the Radiative Forcing Model

Intercomparison Project (RFMIP; Pincus et al., 2016), PD ERFs will be quantified relative to the PI period from PD-PI changes in emissions, concentrations, and/or land use due to anthropogenic activities. This approach offers a consistent methodology for diagnosing the ERFs of all forcing agents or Earth System (ES) perturbations and, if applied consistently across all models as part of CMIP6 (Eyring et al., 2016), should help to address some of the deficiencies and uncertainties associated with previous estimates of forcing (e.g., Myhre et al., 2013a) and improve our understanding of how the ES responds to forcing

(Pincus et al., 2016). The paper is organised as follows. Section 2 provides a brief description of the UK's ESM, UKESM1, used in this study. Section 3 outlines the experimental setup and the simulations carried out. Anthropogenic ERFs are presented in Sect. 4, while conclusions can be found in Sect. 5.

## 2 Model Description

    The model used in this study consists of the atmospheric and land components of the UK's ESM, UKESM1 (Sellar et al.,

2019). UKESM1 is based on the Global Atmosphere 7.1/Global Land 7.0 (GA7.1/GL7.0; Walters et al., 2019) configuration of the Hadley Centre Global Environment Model version 3 (HadGEM3; Hewitt et al., 2011), herein referred to as HadGEM3-GA7.1, to which terrestrial carbon/nitrogen cycles (Sellar et al., 2019) and interactive stratosphere-troposphere chemistry (Archibald et al., 2020) from the UK Chemistry and Aerosol (UKCA; Morgenstern et al., 2009; O'Connor et al., 2014) model have been coupled. The model resolution is N96L85; this is equivalent to a horizontal resolution of roughly 135 km and the

85 terrain-following model levels cover an altitude range up to 85 km above sea level. The physical atmosphere model, HadGEM3-GA7.1, already includes the UKCA prognostic aerosol scheme called GLOMAP-mode (Mann et al., 2010;

Mulcahy et al., 2018), in which secondary aerosol formation makes use of prescribed oxidant fields. Here, the UKCA chemistry and aerosol schemes are coupled, with oxidants from the stratosphere-troposphere chemistry scheme (Archibald et al., 2020) influencing secondary aerosol formation rates. A full description and evaluation of the UKCA chemistry and aerosol schemes in UKESM1 can be found in Archibald et al. (2020) and Mulcahy et al. (2020), respectively. Mulcahy et al. (2018) also implemented a number of aerosol process improvements in HadGEM3-GA7.1 which helped to reduce the aerosol ERF at the present day (year 2000) from $-2.75 \pm 0.06$ W m$^{-2}$ in its predecessor model (i.e., HadGEM3-GA7.0; Walters et al., 2019) to $-1.45 \pm 0.04$ W m$^{-2}$; the aerosol ERF in HadGEM3-GA7.0 previously led to an unrealistic negative total anthropogenic ERF of $-0.60$ W m$^{-2}$.

Differences between HadGEM3-GA7.1 and the atmosphere-only configuration of UKESM1 can be found in Table A1 in the Appendix.

## 3 Model Setup and Experiments

To calculate the PD (year 2014) ERFs relative to the PI (year 1850) period due to a PD-PI perturbation (e.g., change in emissions), timeslice experiments with fixed SSTs and SI were carried out, following the protocol defined by RFMIP (Pincus et al., 2016). The experimental setup was also consistent with recommendations from Forster et al. (2016) and the protocol for timeslice ERF experiments defined in AerChemMIP (Collins et al., 2017).

| Experiment ID | MIP | CO$_2$ concn. | N$_2$O concn. | ODS concns. | Other HC concns. | CH$_4$ concn. | O$_3$ Precursor (VOC, CO, NOx) emissions | Aerosol and aerosol precursor (BC, OC, SO$_2$) emissions | Land use |
|---|---|---|---|---|---|---|---|---|---|
| *piClim-control* | AerChemMIP; RFMIP | 1850 | 1850 | 1850 | 1850 | 1850 | 1850 | 1850 | 1850 |
| *piClim-CO2* | N/A | 2014 | 1850 | 1850 | 1850 | 1850 | 1850 | 1850 | 1850 |
| *piClim-CO2phys[a]* | N/A | 2014 (land surface) 1850 (radiation) | 1850 | 1850 | 1850 | 1850 | 1850 | 1850 | 1850 |
| *piClim-4xCO2* | RFMIP | 4x 1850 value | 1850 | 1850 | 1850 | 1850 | 1850 | 1850 | 1850 |

| | | | | | | | | | |
|---|---|---|---|---|---|---|---|---|---|
| *piClim-4xCO2phys*[a] | N/A | 4x 1850 value (land surface) 1850 (radiation) | 1850 | 1850 | 1850 | 1850 | 1850 | 1850 | 1850 |
| *piClim-N2O* | AerChemMIP | 1850 | 2014 | 1850 | 1850 | 1850 | 1850 | 1850 | 1850 |
| *piClim-HC*[b] | AerChemMIP | 1850 | 1850 | 2014 | 1850 | 1850 | 1850 | 1850 | 1850 |
| *piClim-CH4* | AerChemMIP | 1850 | 1850 | 1850 | 1850 | 2014 | 1850 | 1850 | 1850 |
| *piClim-GHG* | RFMIP | 2014 | 2014 | 2014 | 1850 | 2014 | 1850 | 1850 | 1850 |
| *piClim-SO2* | AerChemMIP | 1850 | 1850 | 1850 | 1850 | 1850 | 1850 | 2014 (SO2); 1850 (BC, OC) | 1850 |
| *piClim-BC* | AerChemMIP | 1850 | 1850 | 1850 | 1850 | 1850 | 1850 | 2014 (BC); 1850 (OC, SO₂) | 1850 |
| *piClim-OC* | AerChemMIP | 1850 | 1850 | 1850 | 1850 | 1850 | 1850 | 2014 (OC); 1850 (BC, SO₂) | 1850 |
| *piClim-aer* | AerChemMIP | 1850 | 1850 | 1850 | 1850 | 1850 | 1850 | 2014 | 1850 |
| *piClim-NOx* | AerChemMIP | 1850 | 1850 | 1850 | 1850 | 1850 | 2014 (NOx); 1850 (VOC, CO) | 1850 | 1850 |
| *piClim-VOC* | AerChemMIP | 1850 | 1850 | 1850 | 1850 | 1850 | 2014 (VOC, CO); 1850 (NOx) | 1850 | 1850 |
| *piClim-O3* | AerChemMIP | 1850 | 1850 | 1850 | 1850 | 1850 | 2014 (VOC, CO, NOx) | 1850 | 1850 |
| *piClim-NTCF*[c] | AerChemMIP | 1850 | 1850 | 1850 | 1850 | 1850 | 2014 | 2014 | 1850 |
| *piClim-LU* | RFMIP | 1850 | 1850 | 1850 | 1850 | 1850 | 1850 | 1850 | 2014 |
| *piClim-Anthro* | RFMIP | 2014 | 2014 | 2014 | 2014 | 2014 | 2014 | 2014 | 2014 |


**Table 1: List of all the atmosphere-only experiments carried out with the UK's Earth System Model, UKESM1, to quantify PD ERFs from PD-PI changes in emissions, concentrations and/or land use. Each simulation was 45 years in length, with analysis based on the last 30 years. [a]In the *piClim-CO2phys* and *piClim-4xCO2phys* simulations, only the land surface sees the perturbation in $CO_2$, the radiation scheme still sees the PI concentration. [b]The AerChemMIP experiment piClim-HC changes concentrations of ozone depleting substances or halocarbons (HCs) from PI to PD levels. [c]The AerChemMIP experiment *piClim-NTCF* is also known as *piClim-aerO3* in RFMIP.**

Effectively, this involves running a PI timeslice experiment, called *piClim-control* here, in which SSTs, SI and all other boundary conditions were fixed at year-1850 levels. The SSTs and SI used in *piClim-control* were monthly mean climatologies derived from 30 years (i.e., years 2156-2185 inclusive) of output from the UKESM1 PI coupled control experiment (*piControl*) characterised in Sellar et al. (2019) and one of an underpinning set of coupled experiments for CMIP6 (Eyring et al., 2016). It also used 30-year monthly mean climatologies for the vegetation distribution, canopy height, leaf area index, and surface seawater dimethyl sulphide (DMS) and chlorophyll concentrations derived from the same period of *piControl*. Fixing the vegetation distribution was not part of the RFMIP protocol and any potential vegetation RAs will be somewhat constrained. This is due to the simulations being based on the configuration of UKESM1 used for the Atmosphere Model Intercomparison Project (AMIP) simulation (which prescribed vegetation characteristics). Although the AerChemMIP protocol (Collins et al., 2017) requested use of the maximum capability possible, interactive vegetation was not a model requirement. The extra RFMIP experiments carried out here were only done as a late addition and the same experimental setup was kept for internal consistency.

The model was initialised using output from the start of the 30-year period used to produce the PI climatologies (i.e., January 2156 of *piControl*). All the other experiments are perturbation experiments, parallel to *piClim-control*, in which selected emissions, concentrations and/or land use were changed from year-1850 to year-2014 values. Although AerChemMIP and RFMIP recommend 30 years for fixed SST (fSST) timeslice ERF experiments, the perturbations to the LLGHGs took up to 15 years to propagate fully into the stratosphere due to the turnover timescale associated with the Brewer-Dobson circulation (e.g., Butchart, 2014). Therefore, all simulations were 45 years in length. Using the latter 30 years of the paired simulations, the ERFs were diagnosed as the time-mean global-mean PD-PI difference in the TOA net radiative fluxes. Running for 30 years when the model has reached steady state reduces the uncertainty associated with meteorological variability (e.g., Shindell et al., 2013a) and improves the estimate of the ERF. Details on how the ERF was further decomposed can be found in Sect. 4.1.

In all cases, the GHG concentrations for 1850 and/or 2014 were taken from Meinhausen et al. (2017). However, the recommended concentrations for the different GHGs in UKESM1 are implemented differently. In the case of $CO_2$, the prescribed concentration is uniform in mass mixing ratio throughout the model domain. For methane ($CH_4$) and nitrous oxide ($N_2O$), the recommended concentrations are treated as lower boundary conditions (LBCs); their 3D distributions are modelled interactively by the UKCA chemistry scheme (Archibald et al., 2020) and coupled to radiation. For $O_3$ depleting substances

(ODSs) or halocarbons (HCs) in *piClim-HC*, their concentrations are prescribed separately (and consistently) in UKCA and in the radiation scheme. For the UKCA chemistry scheme, LBCs are prescribed for trichlorofluoromethane (CFC11), dichlorodifluoromethane (CFC12), and methyl bromide ($CH_3Br$), all of which include contributions from other chlorine- and bromine-containing source gases not explicitly treated in UKCA. This approach ensures that the correct stratospheric chlorine and bromine loadings are used for the PD period. Further details on the species included in the CFC11, CFC12, and $CH_3Br$ LBCs can be found in Archibald et al. (2020). For the radiation scheme, the radiative effects of ODSs are handled by prescribing the mass mixing ratio of a lumped species (CFC12-eq) uniformly throughout the atmosphere, consistent with the UKCA LBCs. Finally, the *piClim-GHG* experiment collectively perturbs all the GHGs from PI to PD levels, including non $O_3$-depleting halocarbons (HCs). For these latter GHGs, a uniform mass mixing ratio of a lumped species (HFC134a-eq), provided by Meinhausen et al. (2017), is prescribed in the radiation scheme.

In all the experiments in Table 1, the emissions of primary aerosol and aerosol precursors (BC, Organic carbon (OC), and sulphur dioxide ($SO_2$)) and $O_3$ precursors (volatile organic compounds (VOCs), carbon monoxide (CO), and nitrogen oxides (NOx)) excluding $CH_4$ for 1850 and/or 2014 were taken from Hoesly et al. (2018) and van Marle et al. (2017). In the case of the NOx emissions perturbation experiment (*piClim-NOx*), both aircraft and surface anthropogenic emissions were changed to PD levels. In *piClim-VOC*, both VOC *and* CO anthropogenic emissions were changed to PD levels while the experiment *piClim-O3* perturbs emissions of VOC, CO, and NOx *only*, with the $CH_4$ concentration remaining at PI levels. Finally, although near-term climate forcers (NTCFs) include $CH_4$ and short-lived halocarbons (e.g., Myhre et al., 2013a), in the context of the AerChemMIP protocol (Collins et al., 2017), the experiment *piClim-NTCF* does not perturb concentrations of $CH_4$ or other short-lived GHGs. It *only* changes anthropogenic emissions of aerosol and aerosol precursors (BC, OC, and $SO_2$) and $O_3$ precursors (VOC, CO, NOx) to PD levels; it is also referred to as *piClim-aerO3* in the RFMIP protocol (Table 1).

For prescribing the anthropogenic land use change at 2014, the difference in vegetation between 1850 and 2014 was taken from a UKESM1 coupled historical simulation, in which the only transient forcing was anthropogenic land use change. Natural volcanic and solar forcings were fixed in all simulations at 1850 levels (Arfeuille et al., 2014; Thomason et al., 2018; Matthes et al., 2017) using those specified for CMIP6 (Eyring et al., 2016). Table 1 gives a full list of the fSST ERF experiments carried out with UKESM1 for this study. The only experiment omitted from the  fSST ERF experiments specified in the RFMIP and AerChemMIP protocols is *piClim-NH3*; this is because UKESM1's aerosol scheme, GLOMAP-mode (Mann et al., 2010; Mulcahy et al., 2018; Mulcahy et al., 2020), does not include any treatment for nitrate aerosol.

Through a partnership between the Met Office Hadley Centre (MOHC; https://www.metoffice.gov.uk/climate-guide/science/science-behind-climate-change/hadley), the UK's National Centre for Atmospheric Science (NCAS; https://www.ncas.ac.uk/en/), New Zealand's National Institute for Water and Atmospheric Research (NIWA; https://www.niwa.co.nz/), and the National Institute of Meteorological Science/Korean Meteorological Administration

(NIMS-KMA; http://nims.go.kr/MA/main.jsp), the fSST experiments carried out with UKESM1 were spread across multiple high performance computing (HPC) platforms. Due to the non-linearity of the equations being solved, ESMs are sensitive to the propagation of small perturbations, resulting in a lack of bit reproducibility. As a result, we aimed to verify that the differences in model output were not statistically significant from each other.


To test this, we created an ensemble of short runs on each machine by perturbing selected variables in their initial conditions, using a perturbation with a numerical value comparable to the machine's precision. The spread of results (at each point in time and space) on each platform was then used to determine whether they could each have been sampled from the same ensemble of results generated on either machine. A permutation method was used to ensure statistical independence between 280 neighbouring points according to the work described by Wilks (1997). A paper, describing this protocol in more detail, is in preparation (Teixeira et al., 2020). In addition to this, a number of perturbation experiments were carried out in duplicate to test the sensitivity of the ERFs to differences in HPC platform. These duplicate experiments are listed in Table 2 and will be available through the Earth System Grid Federation (ESGF; https://esgf.llnl.gov/) archive as different realisations of the same experiment.


| HPC | Compiler | Experiment ID | Realisation ID |
|---|---|---|---|
| Met Office CrayXC40 | Cray compiling environment 8.3.4 | *piClim-control* | R1 |
| | | *piClim-SO2* | R2 |
| | | *piClim-OC* | R2 |
| | | *piClim-NTCF* | R2 |
| NIWA XC50 | Intel Compilers 17.0.4 20170411 | *piClim-SO2* | R1 |
| | | *piClim-OC* | R1 |
| KMA Cray XC40 | Cray compiling environment 8.3.7 | *piClim-NTCF* | R1 |

**Table 2: List of atmosphere-only PI control (*piClim-control*) and duplicate perturbation experiments (*piClim-X*) carried out with UKESM1 on different HPC platforms.**

## 4 Anthropogenic Effective Radiative Forcings (ERFs)

The ERF has been calculated from the difference ($\Delta$) in the net TOA radiative flux ($F$) between the perturbed simulation (e.g., *piClim-CH4;* Table 1) and the *piClim-control* simulation as follows:

$$ERF = \Delta F, \tag{3}$$

where $\Delta F$ is in response to whole-atmosphere PD-PI changes in composition and/or other RAs; no masking of the response is

applied. For example, the ERF quantified from *piClim-HC* minus *piClim-control* includes the direct radiative effect from the increase in ODS concentrations, the indirect radiative effect of the whole-atmosphere $O_3$ response as well as other changes to the TOA radiative fluxes due to whole-atmosphere and land surface RAs. The ERF can be decomposed into the clear-sky (CS) ERF ($ERFcs$) and the change in the cloud radiative effect ($\Delta CRE$) using the diagnosed CS radiative flux ($Fcs$):

$$ERF = \Delta Fcs + \Delta(F - Fcs) \tag{4}$$
$$= ERFcs + \Delta CRE. \tag{5}$$

However, many of the experiments in this study either directly perturb aerosol emissions and/or alter aerosol concentrations via chemical and dynamical interactions. Changes in aerosol can bias the diagnosed CRE as aerosol scattering and absorption

typically reduce the contrast in SW reflection between cloudy and CS scenes; a process termed "cloud masking" (e.g., Zelinka et al., 2014). In consideration of this, we have calculated the change in the CRE from "clean" radiation calls that exclude ari, as recommended in Ghan (2013):

$$ERF = \Delta(F - Fclean) + \Delta Fcs,clean + \Delta(Fclean - Fcs,clean) \tag{6}$$
$$= Aerosol\ IRF + ERFcs,clean + \Delta CRE' \tag{7}$$
$$= ERFcs' + \Delta CRE'. \tag{8}$$

The ERF is, thus, separated into a component due to cloud property changes ($\Delta CRE'$) and the non-cloud forcing ($ERFcs'$). Here, $ERFcs'$ is the sum of the aerosol IRF and any non-aerosol changes in CS fluxes and differs slightly from $ERFcs$ in Eqn.

(5), in that it can include the impact of aerosol scattering and absorption in the clear-air above or below clouds. One acknowledged limitation is that variations in gaseous absorption and emission between clear and cloudy scenes also lead to cloud masking effects (e.g., Soden et al., 2008). Although Ghan's method removes the very prominent influence of aerosols, cloud masking from $O_3$ and GHGs may still affect the separation of ERF into the CS and CRE components.

### 4.1 Overview of ERFs

The ERF, and its CS (*ERFcs'*), and *ΔCRE'* contributions, following Eqn. (8) are listed in Table 3 for all perturbation experiments relative to *piClim-control*, and are further decomposed into the SW (solar), LW (terrestrial) and net (SW + LW) components. Table 3 also includes estimates of the model-derived uncertainty in the different components by calculating the standard error based on Forster et al. (2016). Although the errors quoted are small (i.e., less than 0.04 W m$^{-2}$), they do not represent the true uncertainty in the quantified ERFs; uncertainties due to emissions (e.g., Hoesly et al., 2018), model biases

(e.g., Archibald et al., 2020), incorrect sensitivity to changing emissions (e.g., Archibald et al., 2010; Wild et al., 2020), radiative transfer schemes (e.g., Pincus et al., 2020), and/or missing or unresolved processes are not taken into account.

  The ERFs are also plotted in Fig. 2. Together, they show that the ERF from GHGs is $2.92 \pm 0.04$ W m$^{-2}$, which is offset by an aerosol ERF of $-1.09 \pm 0.04$ W m$^{-2}$. The GHG ERF estimate is lower and the aerosol ERF is consistent with estimates of 3.09

and $-1.10$ W m$^{-2}$, respectively (Andrews et al., 2019), from the HadGEM3 GC3.1 (Williams et al., 2017) physical model (herein referred to as HadGEM3-GC3.1) upon which UKESM1 is based but are consistent with the range of previous estimates (e.g., Myhre et al., 2013a). The net anthropogenic ERF is $1.76 \pm 0.04$ W m$^{-2}$, again consistent with the range of estimates from AR5 (Myhre et al., 2013a) and the estimate from HadGEM3-GC3.1 (1.81 W m$^{-2}$; Andrews et al., 2019). The net anthropogenic ERF quantified here is also narrowly inside the range of net anthropogenic ERF of 2.3 W m$^{-2}$ (1.7 to 3.0 W m$^{-2}$; $5 - 95$ % confidence

interval) derived from a top-down energy budget constraint based on measurements of historical global mean temperature change and the Earth's heat uptake, and model estimates of the Earth's radiative response (Andrews and Forster, 2020).

| PD-PI Perturbation | Present day (PD; year 2014) effective radiative forcings (ERFs) relative to the pre-industrial (PI; year 1850) period (W m$^{-2}$) | | | | | | |
|---|---|---|---|---|---|---|---|
| | NET ERF | LWcs' | SWcs' | LW ΔCRE' | SW ΔCRE' | NETcs' | NET ΔCRE' |
| $CO_2$ concn. | 1.89 ±0.04 | 1.61 ±0.02 | 0.09 ±0.02 | -0.31 ±0.02 | 0.50 ±0.02 | 1.70 ±0.02 | 0.19 ±0.02 |
| $CO_2$ phys concn. | 0.03 ±0.04 | -0.07 ±0.03 | -0.03 ±0.02 | -0.02 ±0.02 | 0.16 ±0.02 | -0.11 ±0.03 | 0.14 ±0.02 |
| $4xCO_2$ concn. | 7.97 ±0.04 | 6.83 ±0.03 | 0.46 ±0.02 | -1.51 ±0.02 | 2.18 ±0.02 | 7.29 ±0.03 | 0.68 ±0.02 |
| $4xCO_2$ phys concn. | 0.13 ±0.03 | -0.30 ±0.02 | -0.10 ±0.02 | 0.00 ±0.01 | 0.53 ±0.02 | -0.40 ±0.02 | 0.53 ±0.02 |
| $N_2O$ concn. | 0.25 ±0.04 | 0.28 ±0.03 | -0.04 ±0.02 | -0.08 ±0.01 | 0.09 ±0.03 | 0.25 ±0.03 | 0.01 ±0.02 |

| | | | | | | | |
|---|---|---|---|---|---|---|---|
| ODS concns. | -0.18 ±0.04 | 0.45 ±0.02 | -0.45 ±0.02 | 0.23 ±0.02 | -0.40 ±0.03 | 0.00 ±0.02 | -0.18 ±0.02 |
| $CH_4$ concn. | 0.97 ±0.04 | 0.74 ±0.02 | 0.11 ±0.02 | -0.39 ±0.02 | 0.50 ±0.02 | 0.85 ±0.03 | 0.12 ±0.02 |
| GHG ($CO_2$, $N_2O$, ODSs, Other HCs, $CH_4$) concns. | 2.92 ±0.04 | 3.08 ±0.02 | -0.18 ±0.02 | -0.63 ±0.02 | 0.65 ±0.03 | 2.90 ±0.02 | 0.02 ±0.03 |
| $SO_2$ emissions | -1.37 ±0.03 | 0.15 ±0.03 | -0.61 ±0.02 | 0.17 ±0.02 | -1.08 ±0.03 | -0.46 ±0.03 | -0.91 ±0.02 |
| BC emissions | 0.37 ±0.03 | -0.02 ±0.02 | 0.36 ±0.02 | -0.16 ±0.01 | 0.15 ±0.02 | 0.38 ±0.02 | -0.01 ±0.02 |
| OC emissions | -0.22 ±0.04 | 0.03 ±0.02 | -0.18 ±0.02 | -0.02 ±0.02 | -0.05 ±0.03 | -0.14 ±0.03 | -0.07 ±0.02 |
| Aerosol and aerosol precursor (BC, OC, $SO_2$) emissions | -1.09 ±0.04 | 0.16 ±0.03 | -0.26 ±0.02 | 0.01 ±0.02 | -1.00 ±0.03 | -0.10 ±0.02 | -1.00 ±0.02 |
| NOx emissions | 0.03 ±0.04 | 0.05 ±0.03 | 0.03 ±0.02 | -0.03 ±0.01 | -0.02 ±0.02 | 0.08 ±0.03 | -0.05 ±0.02 |
| VOC and CO emissions | 0.33 ±0.04 | 0.10 ±0.03 | 0.03 ±0.02 | -0.09 ±0.01 | 0.28 ±0.02 | 0.13 ±0.03 | 0.20 ±0.02 |
| $O_3$ precursor (VOC, CO, NOx) emissions | 0.21 ±0.04 | 0.07 ±0.03 | 0.06 ±0.02 | -0.07 ±0.02 | 0.15 ±0.03 | 0.13 ±0.02 | 0.08 ±0.02 |
| NTCF (BC, OC, $SO_2$, VOC, CO, NOx) emissions | -1.03 ±0.04 | 0.23 ±0.03 | -0.26 ±0.02 | -0.08 ±0.02 | -0.92 ±0.03 | -0.03 ±0.03 | -1.00 ±0.02 |
| Land Use | -0.17 ±0.04 | 0.02 ±0.03 | -0.30 ±0.02 | 0.03 ±0.01 | 0.09 ±0.03 | -0.28 ±0.03 | 0.11 ±0.03 |
| Anthro (GHG concns., Aer. and aerosol | 1.76 ±0.04 | 3.34 ±0.02 | -0.72 ±0.02 | -0.64 ±0.01 | -0.22 ±0.03 | 2.63 ±0.03 | -0.86 ±0.03 |

| precursor (BC, OC, SO₂) ems., O₃ precursor (VOC, CO, NOx) ems., and land use | | | | | | |
|---|---|---|---|---|---|---|

**Table 3: Present-day (year 2014) ERFs relative to the pre-industrial (year 1850) period derived from Eqn. (8), and including an estimate of the standard error. Where duplicate experiments exist (e.g., *piClim-SO2*), the quoted ERFs use realisations 2. Units in W m⁻².**

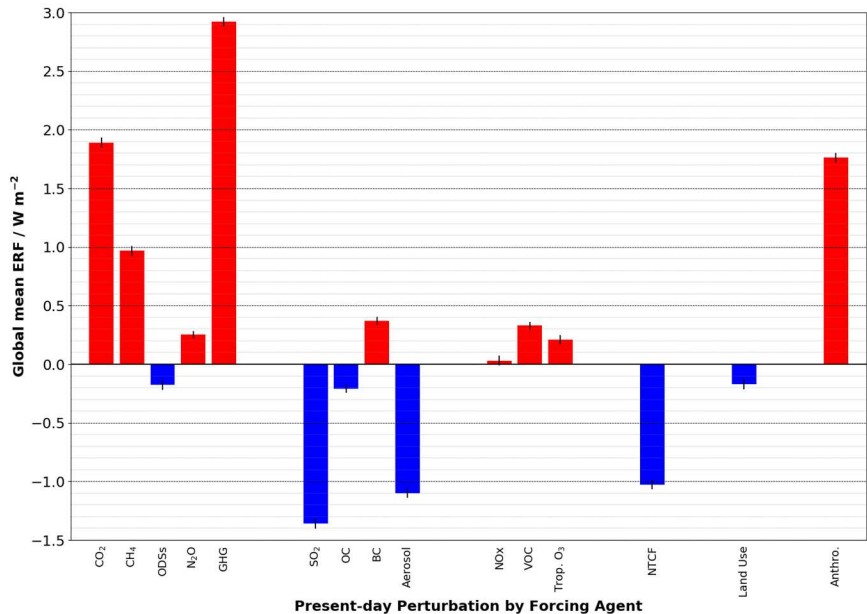

**Figure 2: Present day (year 2014) anthropogenic ERFs relative to the pre-industrial (year 1850) period derived from Eqn. (8), and including an estimate of the standard error. Units in W m⁻².**

Figure 3 shows global and zonal mean distributions of the ERFs from PD-PI changes in GHG concentrations, aerosol and aerosol precursor (BC, OC. SO₂) emissions, O₃ precursor (VOC, CO, NOx) emissions, land use change, and total anthropogenic sources. It shows that the ERF from GHGs (Fig. 3a) has a robust signal over 93 % of the globe; it is positive everywhere except for the southern high latitudes (Fig. 3b). This negative ERF is due to the negative ERF from the *piClim-HC* perturbation experiment (Table 3) which offsets the positive ERFs from the other LLGHG experiments. The breakdown of the GHG ERF into its individual speciated contributions will be discussed further in Sect. 4.2 and results from *piClim-HC*

will be discussed in Sect. 4.2.3. The aerosol ERF (Figure 3c and d) is robust over a smaller area (52 %) of the globe and is more spatially heterogeneous than the GHG ERF due to the shorter aerosol lifetime. However, the distribution of the aerosol ERF is mostly negative, except for over bright surfaces and regions where the positive forcing from black carbon (BC) emissions outweighs the negative forcing from scattering aerosols such as sulphate and OC. A breakdown of the aerosol ERF between constituents and between ari and aci will be presented and discussed in Sect. 4.3.


Figures 3e and 3f show the global and zonal mean distributions of the ERF from emissions of $O_3$ precursors (VOC, CO, and NOx), excluding $CH_4$. It shows that the ERF from changes in VOC, CO, and NOx emissions alone is positive ($0.21 \pm 0.04$ W $m^{-2}$; Table 3), with only 10 % of the globe showing a robust signal. Further analysis of the ERF from $O_3$ precursor emissions and their contribution, along with $CH_4$, to tropospheric $O_3$ RF, can be found in Sect. 4.4. The distribution of a robust land use

ERF (Fig. 3g; Fig. 3h) is also limited in spatial extent (~12 % of the globe); much of the negative ERF is concentrated over the northern hemisphere (NH) continental regions (e.g., North America, South East Asia). Together with aerosols, the combined ERF outweighs the positive ERF from GHGs, leading to a negative total anthropogenic ERF over parts of the NH continents (Fig. 3i). As was the case with the GHG ERF, the negative ERF over the SH high latitudes is still evident in the total anthropogenic ERF (Fig. 3i; Fig. 3j).


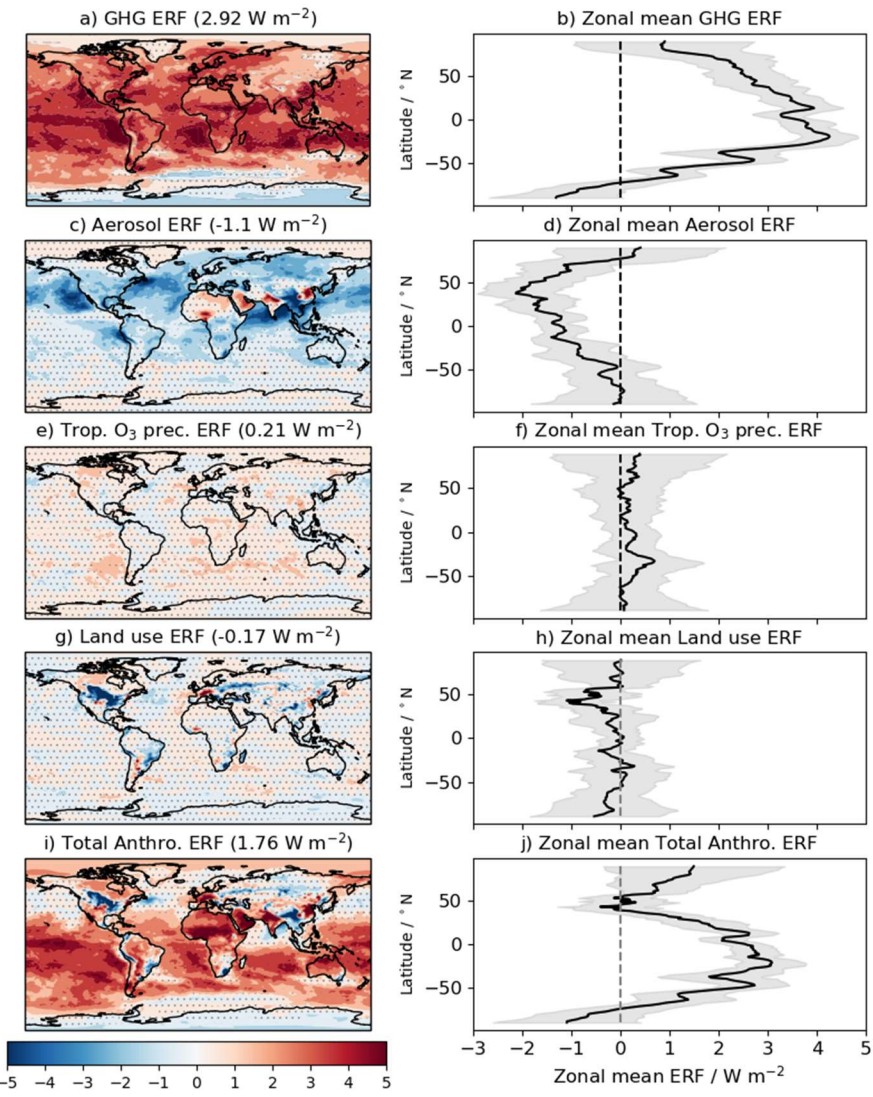

**Figure 3: Geographical and zonal mean distributions of the present-day (PD; year 2014) ERFs relative to the pre-industrial (PI; year 1850) period in a) and b) for GHGs, in c) and d) for aerosols, in e) and f) for O₃ precursors, in g) and h) for land use change, and in i) and j) for total anthropogenic, respectively. In the global distribution plots, global mean ERFs are included and areas are stippled, where the ERF is not statistically significant at the 95 % confidence interval. In the zonal mean plots, the grey shading shows the ± 1 standard deviation in the zonal mean ERF. Units in W m⁻².**

As discussed in Sect. 3, the UKESM1 fSST experiments were run on multiple HPC platforms. Statistical methods ensure that the model is not scientifically different on the different HPC platforms, but such duplicate experiments still produce slightly different results. This raises the question of the impact of such differences on the quantification of ERFs with UKESM1. To address this question, the experiments described in Table 2 were used to compare the difference in TOA radiative fluxes of

equivalent realisation experiment pairs. The two-sample Kolmogorov–Smirnov test between the monthly TOA radiative fluxes from the realisation pairs (null hypothesis that the samples are drawn from the same distribution) shows that the TOA radiative

fluxes between the two realisation pairs are statistically identical at a confidence level ($\alpha$) of 5 % (Table 4). Despite the fact that one cannot conclude that the distributions are identical, for each pair of experiments there is no evidence suggesting that the two distributions are different.

| Pair | Differences in ERF and its components (W m$^{-2}$) | | | | | | |
|---|---|---|---|---|---|---|---|
| | NET ERF | LWcs' | SWcs' | LW $\triangle$CRE' | SW $\triangle$CRE' | NETcs' | NET $\triangle$CRE' |
| *piClim-SO2* R1 vs R2 | 0.01 ±0.03 | -0.01 0.02 | 0.01 ±0.02 | 0.01 ±0.02 | -0.01 ±0.02 | 0.01 ±0.02 | 0.01 ±0.02 |
| *piClim-OC* R1 vs R2 | 0.01 ±0.03 | 0.01 ±0.02 | -0.01 ±0.01 | 0.03 ±0.01 | -0.02 ±0.02 | -0.01 ±0.02 | 0.01 ±0.02 |
| *piClim-NTCF* R1 vs R2 | 0.01 ±0.03 | -0.01 ±0.02 | 0.01 ±0.02 | 0.01 ±0.02 | -0.01 ±0.03 | 0.01 ±0.02 | 0.01 ±0.02 |

**Table 4: Differences in the present-day ERF and its components between two realisations (R1 and R2) of the same perturbation experiment relative to *piClim-control*. Units in W m$^{-2}$.**

## 4.2 Long-Lived Greenhouse Gases (LLGHGs)

### 4.2.1 Carbon Dioxide (CO$_2$)

Atmospheric CO$_2$ concentrations have risen from 284.3 to 397.5 ppm between 1850 and 2014 (Meinshausen et al., 2017) resulting in an ERF of 1.89 ± 0.04 W m$^{-2}$ (Table 3) and is the largest individual contribution to the total historical forcing. Myhre et al. (2013a) report a SARF of 1.82 ± 0.19 W m$^{-2}$ for the period from 1750 to 2011 which, coincidentally, has a near identical rise in CO$_2$ of 113 ppm to that assessed here, although CO$_2$ forcing has a logarithmic dependency on concentration. An updated assessment based on line-by-line calculations (Etminan et al., 2016) increased SARF for 2015 relative to 1750 to

1.95 W m$^{-2}$ but for a larger CO$_2$ rise of 121 ppm. Applying Etminan et al. (2016) to our case reveals a SARF of 1.80 W m$^{-2}$. Our ERF estimate for 2014 is, therefore, larger than the SARF by 0.09 W m$^{-2}$. As expected for a GHG, the ERF is dominated by the LWcs' component (1.61 ± 0.02 W m$^{-2}$) with an additional contribution from the SWcs' component (0.09 ± 0.02 W m$^{-2}$) coming from direct effects and a RA in snow cover across the northern latitudes. There is also a NET $\triangle$CRE' contribution (0.19 ± 0.02 W m$^{-2}$) from a reduction in cloud cover. The SW direct effect occurs in the near-infrared and is small, equivalent

to around 2 % of the LW forcing (Pincus et al., 2020). On that assumption, the SWcs' component is dominated by the rapid albedo adjustment over the direct effect.

Rising atmospheric $CO_2$ also exerts an indirect forcing through rapid changes in plant stomatal conductance (Doutriaux-Boucher et al., 2009; Richardson et al., 2018) enhancing plant water use efficiency and reducing evapotranspiration leading to an increase in sensible heating at the surface and corresponding drying of the boundary layer and reduction in low clouds. This mechanism is known as a physiological forcing. In UKESM1 (*piClim-CO2phys*; Table 3), this forcing is small at PD and 4xCO$_2$ levels (0.03 ± 0.04 and 0.13 ± 0.03 W m$^{-2}$) and scales in line with the total $CO_2$ ERF. It results from a balance between a negative LWcs' component (-0.07 ± 0.03; -0.30 ± 0.02 W m$^{-2}$) associated with surface warming and a positive SW $\triangle$CRE' component (0.16 ± 0.02; 0.53 ± 0.02 W m$^{-2}$) associated with a reduction in low-level clouds and from smaller terms including a negative SWcs' component from reduced water vapour. Previous Hadley Centre models at 4xCO$_2$ found a similar but larger effect: 1.1 W m$^{-2}$ in HadCM3LC (Doutriaux-Boucher et al., 2009) and 0.25 W m$^{-2}$ in HadGEM2-ES (Andrews et al., 2012b). We find that UKESM1 has a similar LWcs' component compared to HadGEM2-ES implying that UKESM1 has a similar surface warming adjustment associated with the physiological effect but offset by a weaker SW $\triangle$CRE' component. The inclusion of the physiological effect and rapid albedo adjustment in the $CO_2$ ERF acts to increase the forcing slightly. These additional adjustments likely account for our slightly higher ERF relative to the SARF from line-by-line estimates (Etminan et al., 2016).

### 4.2.2 Nitrous Oxide ($N_2O$)

The ERF due to changes in $N_2O$ concentration from the PI period (273 ppbv in 1850) to the present day (327 ppbv in 2014) is calculated as 0.25 ± 0.04 W m$^{-2}$ (Table 3), following Eqn. (8) described above. The predominant contribution to the $N_2O$ ERF is the LWcs' component (0.28 ± 0.03 W m$^{-2}$), with a small offset by the SWcs' component (-0.04 ± 0.02 Wm$^{-2}$); the NETcs' component sums up to 0.25 ± 0.03 W m$^{-2}$. The NET $\triangle$CRE' component is insignificant (0.01 ± 0.03 W m$^{-2}$), with SW $\triangle$CRE' and LW $\triangle$CRE' contributions of 0.09 ± 0.03 W m$^{-2}$ and -0.08 ± 0.02 W m$^{-2}$, respectively. In comparison, the net ERF calculated here (0.25 ± 0.04 W m$^{-2}$) is higher than the SARF values of 0.17 ± 0.03 W m$^{-2}$ from AR5 for 2011 (Myhre et al., 2013a) and of 0.18 W m$^{-2}$ for 2014 based on the updated expression from Etminan et al. (2016). This is likely to be due to the effect of adjustments associated with changing $N_2O$ that were not considered as part of the SARF in AR5 (Myhre et al., 2013a) or Etminan et al. (2016), including $O_3$ depletion and fast cloud adjustments. The UKESM1 estimate agrees well with the AerChemMIP multi-model mean of 0.23 ± 0.05 W m$^{-2}$ (Thornhill et al., 2020). Previously, Hansen et al. (2005) calculated an IRF and a SARF of 0.15 W m$^{-2}$ due to the change in $N_2O$ from 278 to 316 ppbv.

The global distribution of the individual components contributing to the $N_2O$ ERF (Fig. 4) shows that the SWcs' component is negligible. The LWcs' component is predominantly positive, due mainly to $N_2O$ direct forcing. The SW $\triangle$CRE' and LW

△CRE' components of the ERF are largely noise (at 95 % confidence level; Fig 4), with a net contribution to the ERF of close to zero.

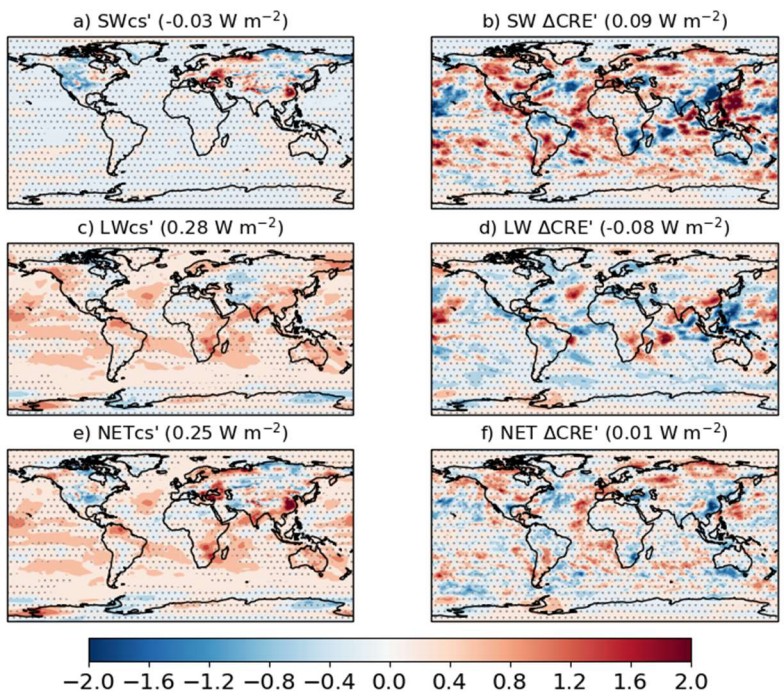

**Figure 4: Global distributions of the PD N₂O ERF components relative to the PI period, i.e., *piClim-N2O* minus *piClim-Control*; (a) SWcs', (b) SW △CRE', (c) LWcs', (d) LW △CRE', (e) NETcs', and (f) NET △CRE', based on Eqn. (8). Global mean values are shown in brackets. Regions where the ERF components are outside the 95 % confidence level are stippled. Units in W m⁻².**

### 4.2.3 Ozone ($O_3$) Depleting Substances (ODSs)

The ERF from the PD-PI change in ODSs is quantified, using *piClim-HC* relative to *piClim-control*, as $-0.18 \pm 0.04$ W m⁻², which is dominated by the NET △CRE' component ($-0.18 \pm 0.02$ W m⁻²; Table 3). Figure 5 shows global distributions of the SWcs', SW △CRE', LWcs', LW △CRE', NETcs', and NET △CRE' components, respectively. For CS conditions, the SW component (SWcs') is characterised by negative values over the southern high latitudes (to a lesser extent in the northern high latitudes) which is linked to pronounced Antarctic $O_3$ depletion and some decreases in Arctic $O_3$ (not shown). The LWcs' component is predominantly positive, reflecting the direct effect of ODSs acting as GHGs in the *piClim-HC* simulation. The positive LWcs' component at high latitudes contains an offset caused by $O_3$ depletion. Overall, the global mean NETcs' component is negligible ($0.00 \pm 0.02$ W m⁻²; Table 3). Under cloudy conditions, the negative SW △CRE' component and the

positive LW $\triangle$CRE' component of the ERFs are anti-correlated, especially over the Southern Ocean, summing up to a net
contribution to the ERF of -0.18 ± 0.02 W m$^{-2}$.

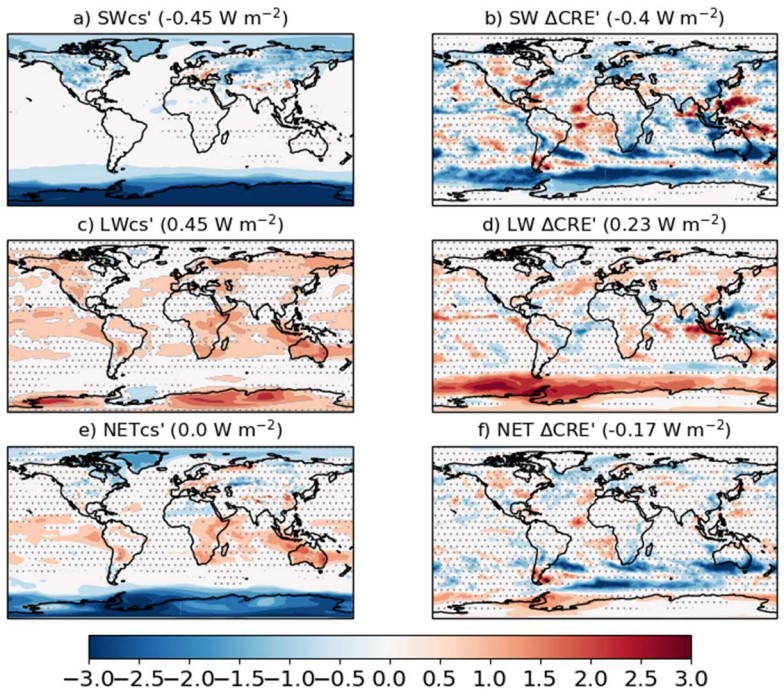

**Figure 5: Global distributions of the PD ODS ERF components relative to the PI period, i.e., *piClim-HC* minus *piClim-Control*; (a)**
**SWcs', (b) SW $\triangle$CRE', (c) LWcs', (d) LW $\triangle$CRE', (e) NETcs', and (f) NET $\triangle$CRE', based on Eqn. (8). Global mean values are**
**shown in brackets. Regions where the ERF components are outside the 95 % confidence level are stippled. Units in W m$^{-2}$.**

As mentioned previously, the ERF includes the direct effect of the ODSs acting as GHGs, the indirect effect of the $O_3$ depletion
that they cause, and any resulting TOA changes due to other RAs. Quantifying the historical evolution of the total $O_3$ RF *alone*
from the CMIP6 models, Skeie et al. (2020) found that UKESM1 was the only model with both tropospheric and stratospheric
chemistry that had a negative total $O_3$ RF at the present day. This is likely due to UKESM1 having a global $O_3$ decline during
the period of increasing ODSs that is stronger than other models and observations by at least a factor of 1.4 (Table 2 in Keeble
et al., 2020). Likewise, in the AerChemMIP multi-model ODS ERF assessment, Morgenstern et al. (2020) used observed $O_3$
trends as a constraint on the modelled ODS ERF; they found that the ERF is likely to be between -0.05 and 0.13 W m$^{-2}$, with
the UKESM1 estimate of -0.18 ± 0.04 W m$^{-2}$ outside of this range. Although Keeble et al. (2020), Skeie et al. (2020) and
Morgenstern et al. (2020) suggest that the negative contribution from $O_3$ depletion to the ODS ERF is too strong in UKESM1,
there is an additional negative offset to the direct radiative effect of the ODSs through the NET $\triangle$CRE' component (Figure 5).

Furthermore, the ODSs will have an impact on tropospheric $O_3$; their contribution to the tropospheric $O_3$ RF will be quantified in Section 4.4.2.

### 4.2.4 Methane ($CH_4$)

The global mean $CH_4$ concentration changed from 808.3 ppbv in the PI (year 1850) period to 1831.5 ppbv in the PD (year 2014) period, resulting in an ERF of $0.97 \pm 0.04$ W m$^{-2}$ (Table 3; Fig. 6). Most of the ERF ($0.74 \pm 0.02$ W m$^{-2}$; Table 3) is due to the LWcs' component, with an additional positive contribution ($0.11 \pm 0.02$ W m$^{-2}$) from the SWcs' component, which is consistent with the growing recognition of the importance of the SW absorption bands in $CH_4$ forcing (Collins et al., 2006; Li et al., 2010; Etminan et al., 2016). There are additional SW and LW $\triangle$CRE' components but these partly cancel out, leading to a small NET $\triangle$CRE' ($0.12 \pm 0.02$ W m$^{-2}$) in addition to the NETcs' component ($0.85 \pm 0.03$ W m$^{-2}$). Estimates of the direct $CH_4$ ERF from HadGEM2 model simulations (Andrews, 2014) and the updated RF expression for $CH_4$ based on line-by-line calculations (Etminan et al., 2016) are in the order of 0.50-0.56 W m$^{-2}$ but the ERF calculated here is higher by more than 0.4 W m$^{-2}$. However, it is consistent with other studies (e.g., Hansen et al., 2005; Shindell et al., 2009; Myhre et al., 2013a), who concluded that the total climate forcing by $CH_4$ is almost double that of the direct forcing and is due to indirect effects. The UKESM1 estimate is also larger than the 0.69 W m$^{-2}$ radiative impact of an increase in $CH_4$ concentration of 1800 ppbv above PD levels quantified by Winterstein et al. (2019) with the ECHAM/MESSy Atmospheric Chemistry (EMAC) coupled model. Although the Winterstein et al. (2019) estimate included indirect forcings from $O_3$ and stratospheric water vapour (SWV), their direct $CH_4$ forcing in the LW is low relative to other models (Lohmann et al., 2010).

The UKESM1 ERF quantified here is at the upper end of estimates from the recent study of AerChemMIP multi-model ERFs by Thornhill et al. (2020). They found that the multi-model mean ERF was 0.70 W m$^{-2}$, with a standard deviation of 0.22 W m$^{-2}$. They attributed part of the inter-model spread to different complexities in the representation of interactive chemistry in the respective models, i.e., some models only captured the direct radiative effect of $CH_4$ (e.g., NorESM2) while others (e.g., UKESM1) also included indirect contributions from $CH_4$-driven changes in $O_3$ and SWV. However, the contribution to the ERF from tropospheric adjustments differed in both magnitude and sign between the models, with UKESM1 being the only model with a positive contribution to the ERF from tropospheric RAs. The relative contributions of the direct and indirect contributions to the total $CH_4$ ERF quantified here and the mechanism behind the positive tropospheric RA can be found in O'Connor et al. (2019).

### 4.2.5 Total Greenhouse Gases (GHGs)

The major drivers of anthropogenic climate change are GHGs, whose forcing is offset by aerosols (Myhre et al., 2013a). Therefore, the total GHG ERF and the aerosol ERF are key values in understanding observed and modelled changes in the climate system since the PI period. As a result, a separate timeslice simulation with all GHG concentrations (*piClim-GHG*; Table 1) at PD levels was conducted following the RFMIP protocol (Pincus et al., 2016).

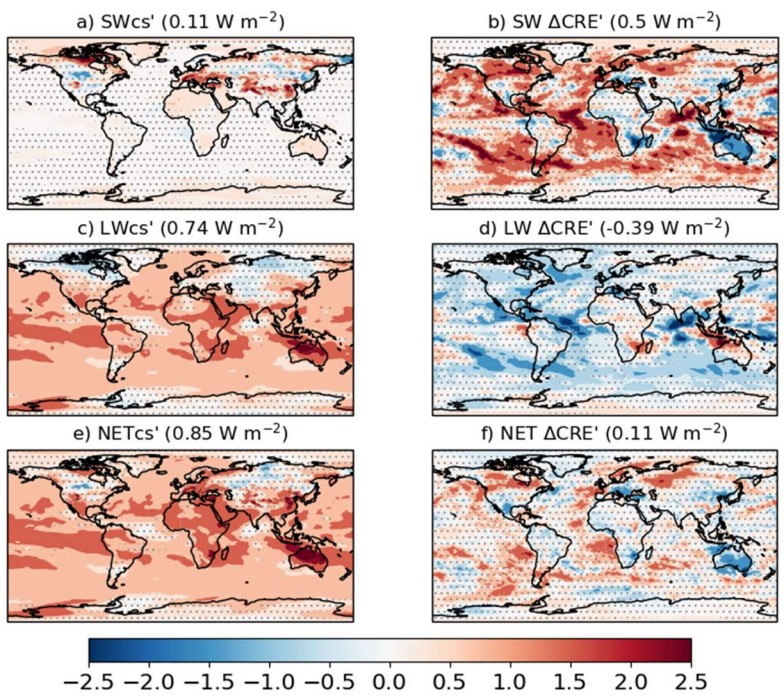

**Figure 6: Global distributions of the PD CH$_4$ ERF components relative to the PI period, i.e., *piClim-CH4* minus *piClim-Control*; (a) SWcs', (b) SW △CRE', (c) LWcs', (d) LW △CRE', (e) NETcs', and (f) NET △CRE', based on Eqn. (8). Global mean values are shown in brackets. Regions where the ERF components are outside the 95 % confidence level are stippled. Units in W m$^{-2}$.**

The UKESM1 *piClim-GHG* experiment leads to an ERF of 2.92 ± 0.04 W m$^{-2}$ (Table 3), which is dominated by a positive LWcs' component (3.08 ± 0.02 W m$^{-2}$) that is partially offset by a negative SWcs' component (-0.18 ± 0.02 W m$^{-2}$). There are significant positive and negative contributions from SW △CRE' (0.65 ± 0.03 W m$^{-2}$) and LW △CRE' (-0.63 ± 0.02 W m$^{-2}$) components but these largely cancel out and contribute little (0.02 ± 0.03 W m$^{-2}$) to the total ERF (Table 3). This GHG ERF is lower than the ERF of 3.09 W m$^{-2}$ estimated from the physical model HadGEM3-GC3.1 (Andrews et al., 2019). Part of this discrepancy may be due to the inclusion in UKESM1 of indirect forcings by O$_3$ and/or aerosols from O$_3$-depleting substances (Morgenstern et al., 2020) and CH$_4$ (O'Connor et al., 2020), for example. The UKESM1 estimate, however, is consistent with the AR5 year-2011 SARF estimate of 2.82 W m$^{-2}$. The latter estimate of 2.82 W m$^{-2}$ has been adjusted to an 1850 baseline from 1750, and taking stratospheric O$_3$ depletion, CH$_4$-driven SWV, and half of the tropospheric O$_3$ forcing (based on the attribution by Stevenson et al., 2013) into account, although some GHG concentrations (e.g., CH$_4$, CO$_2$) have increased between 2011 and 2014 (Nisbet et al., 2016) while others (e.g., ODSs) have declined (Engel et al., 2018).

The simulation *piClim-GHG* perturbs non $O_3$-depleting halocarbons but *piClim-HC* does not. Nevertheless, the small positive RF from non $O_3$-depleting halocarbons of 0.02 W m$^{-2}$ in 2011 (Myhre et al., 2013a) allows one to use the combination of GHG simulations to test for linearity. The sum of the ERFs from *piClim-CO2*, *piClim-CH4*, *piClim-N2O*, and *piClim-HC* is 2.93 ± 0.08 W m$^{-2}$, indicating that the ERFs add linearly and agree with the ERF from *piClim-GHG* (2.92 ± 0.04 W m$^{-2}$).

### 4.3 Aerosols and Aerosol Precursors

Figure 7a summarizes the results from the anthropogenic aerosol experiments, including a breakdown of the ERF into IRF and RAs for each of the anthropogenic aerosol experiments (*piClim-SO2*, *piClim-OC*, *piClim-BC*, and *piClim-aer*). The IRF was calculated using the double-call system where ari (i.e., scattering and absorption) are withdrawn from the second call to the radiation scheme, as in Ghan et al. (2012). The RAs have then been derived as the residual between the ERF and IRF and include all aerosol-induced changes in cloud radiative effects. For completeness, Table 3 also summarises the contributions to the ERF from the SWcs', SW △CRE', LWcs', LW △CRE', NETcs', and NET △CRE' components quantified using Eqn. (8).

Uncertainties in aerosol ERF are driven partly by uncertainties in aerosol lifetime, spatial and temporal distributions, the historical change in aerosol loading, and the cloud response to aerosols (Bellouin et al., 2020). Other contributing factors include uncertainties in the PI aerosol state (Carslaw et al., 2013) as well as the oxidising capacity of the atmosphere (Karset et al., 2018). The additional interactive sources of natural aerosol in UKESM1 from marine DMS, terrestrial and marine biogenic emissions, along with the inclusion of a fully interactive chemistry scheme (Archibald et al., 2020) are generally found to improve the evaluation of PD aerosol in UKESM1 (Mulcahy et al., 2020). This provides some confidence in the underlying physical processes driving the PI aerosol state in this model.

The anthropogenic aerosol ERF evaluated from the "all" (*piClim-aer*) experiment is -1.09 ± 0.04 W m$^{-2}$, which is identical to the ERF of -1.10 W m$^{-2}$ from the physical model HadGEM3-GA7.1 (Andrews et al., 2019) and lower in magnitude than the ERF of -1.45 W m$^{-2}$ derived from HadGEM3-GA7.1 with CMIP5 emissions (Mulcahy et al., 2018). The estimate fits well within the likely range of -1.60 to -0.65 W m$^{-2}$ (16-84 % confidence level) provided by a recent major assessment of aerosol ERF (Bellouin et al., 2020) and the -1.5 to -0.4 W m$^{-2}$ likely range previously assessed by AR5 (Myhre et al., 2013a). The global distribution of aerosol ERF, IRF and aci are shown in Fig. 8. The aerosol ERF (Fig. 8a) is negative over most regions that have a robust signal, and is strongest over the cloudy ocean regions of the Northern Hemisphere where the forcing is dominated by the RA term (Fig. 7a) driven by aci (Fig. 8c). Some areas of Asia and North Africa have a positive aerosol ERF due to BC-rich aerosol loadings that give locally positive aerosol IRF (Fig. 8b). However, globally, the IRF is rather small and negative (-0.15 ± 0.01 W m$^{-2}$) (Fig. 7a) due to scattering by sulphate and OC that is partially offset by absorption from BC.

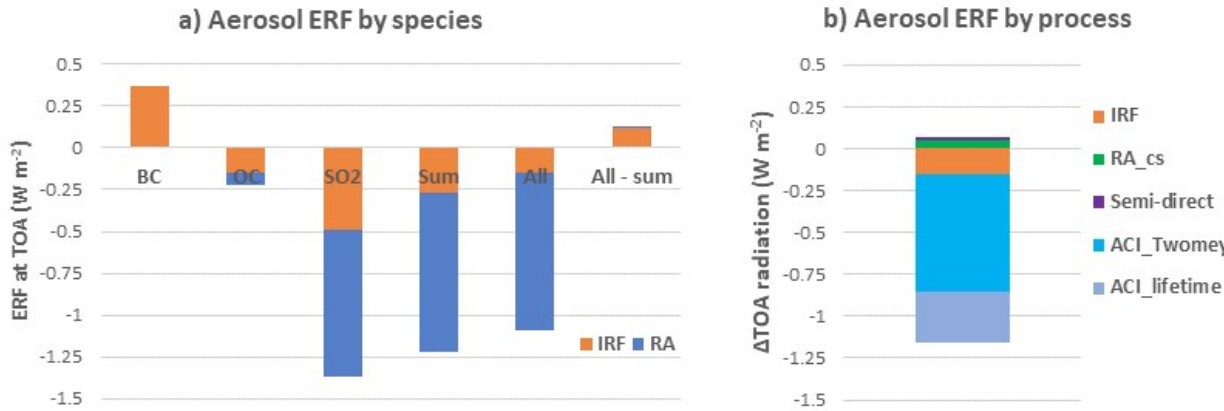

**Figure 7: Aerosol ERF, broken down by species and process. (a) Results are from *piClim-BC*, *piClim-OC*, *piClim-SO2*, from summing those three experiments (Sum), and from *piClim-aer* (All aerosol). (b) Aerosol ERF decomposed into various contributing processes from the *piClim-aer* and accompanying sensitivity tests: IRF = instantaneous radiative forcing, RA = rapid adjustments, aci = aerosol-cloud interactions, RA_cs = clear-sky rapid adjustments (e.g., surface albedo, atmospheric temperature and water vapour). Units in W m$^{-2}$.**

The SO$_2$ emissions are the largest individual contributor to the aerosol ERF and have a strong RA (aci) component leading to an SO$_2$ (equivalent to sulphate) ERF of -1.37 ± 0.03 W m$^{-2}$ (Table 3, Fig. 7a). The IRF component of the sulphate ERF is -0.49 ± 0.01 W m$^{-2}$, which agrees well with the best estimate from AR5 (-0.40 ± 0.2 W m$^{-2}$) and from AEROCOM Phase II (-0.58 to -0.11 W m$^{-2}$) (Myhre et al., 2013b). The BC ERF is 0.37 ± 0.03 W m$^{-2}$, coming mostly from the IRF (0.38 ± 0.01 W m$^{-2}$) and a small negative offset of -0.01 ± 0.02 W m$^{-2}$ from the RA term. As noted in Johnson et al. (2019), BC absorption leads to strong cloud adjustments but the SW and LW components of these almost cancel in HadGEM3-GA7.1 and UKESM1 (see Table 3). This contrasts with many other models where the combination of low cloud enhancements and reductions in upper-level clouds typically result in more substantial negative adjustments, making the BC ERF on average about half the magnitude of the IRF (Stjern et al., 2017). The BC ERF given by UKESM1 is however well within the range assessed by AR5 (0.05 to 0.8 W m$^{-2}$), which took into consideration the possibility that BC emissions and/or absorption efficiency were underestimated in CMIP5 models (Bond et al., 2013). The anthropogenic emissions of BC were specified as 5 Tg yr$^{-1}$ in CMIP5 (year 2000 as PD) but have increased to 8 Tg yr$^{-1}$ in CMIP6 (year 2014 as PD). With CMIP5 emissions, the BC ERF from HadGEM3-GA7.1 was found to be 0.17 W m$^{-2}$ (Johnson et al., 2019) and is comparable to direct BC forcing from other CMIP5 model estimates (Myhre et al., 2013a). It is worth noting that the aerosol absorption was in fairly good agreement with AERONET observations in HadGEM3-GA7.1 simulations that used the CMIP5 emission set (Mulcahy et al., 2018). The slightly higher CMIP6-based estimate of 0.37 ± 0.03 W m$^{-2}$ provided in the present study could therefore be an overestimate, although this is difficult to judge given the uncertainties in comparing models with absorption measurements.

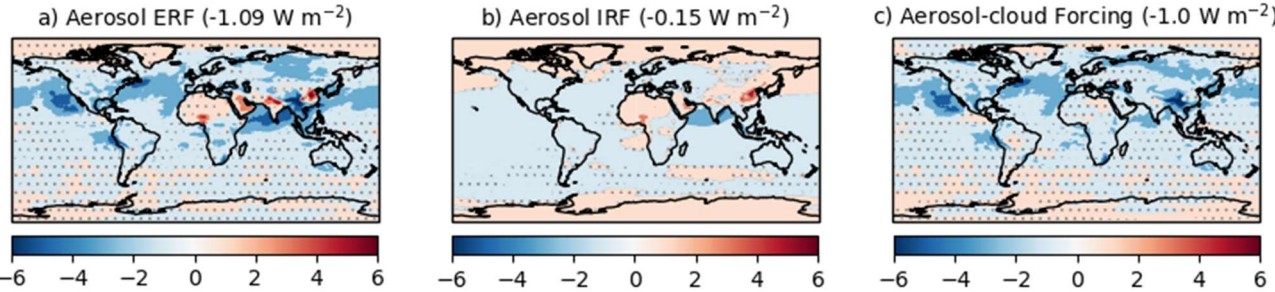

**Figure 8: Changes in TOA net radiation from piClim-aer (all anthropogenic aerosols), including (a) the ERF, (b) the IRF, (c) aerosol-cloud forcing due to aci and the semi-direct aerosol effect. Global mean values are shown in brackets. Regions where the ERF, IRF, and the aerosol-cloud forcing are outside the 95 % confidence level are stippled. Units in W m$^{-2}$.**

The OC ERF is $-0.22 \pm 0.04$ W m$^{-2}$, with $-0.14 \pm 0.01$ W m$^{-2}$ from the IRF and $-0.07 \pm 0.02$ Wm$^{-2}$ from RAs (Fig. 7a, Table 3). The OC IRF estimate agrees fairly well with AR5 that assessed the RF of primary and secondary OC to be $-0.12$ W m$^{-2}$ ($-0.4$ to $0.1$ W m$^{-2}$). Note that OC is non-absorbing in UKESM1 and, as such, neglects the role of brown carbon. This potentially misses a small positive contribution to the aerosol forcing (e.g., Feng et al., 2013), although biomass burning emissions are the dominant global source of brown carbon and these do not change significantly from 1850 to 2014 (Saleh et al., 2014). The contribution of brown carbon to aerosol ERF is, therefore, likely to be small compared to the overall uncertainty in modelling aerosol absorption by BC (Bond et al., 2013). Nitrate aerosols are also not represented in HadGEM3-GA7.1 (Mulcahy et al., 2018; Walters et al., 2019) or UKESM1 (Sellar et al., 2019; Mulcahy et al., 2020) so the ERF associated with ammonia (NH$_3$) emissions could not be evaluated here. The aerosol ERF in UKESM1 would presumably be more strongly negative if the role of nitrate aerosol was included (e.g., Bellouin et al., 2011) and this should be borne in mind when making comparisons with other models or estimates based on observational constraints.

The "all" aerosol forcing experiment combines the increases in BC, OC and SO$_2$ emissions together in one simulation, and interestingly, the ERF is $0.13$ W m$^{-2}$ weaker (less negative) than the sum of the ERFs from the experiments that perturb those emissions separately. The main reasons for this lack of linearity are: (i) Cloud droplet numbers do not increase linearly with aerosol loading (e.g., Jones et al., 1994) and begin to saturate in the "all" experiment (*piClim-aer*), which means OC and BC emissions no longer contribute significantly to aci once co-emitted with year-2014 levels of SO$_2$. (ii) The absorption of upwelling SW radiation by BC is enhanced by increases in aerosol scattering and cloud brightness due to aci, making the IRF less negative in the "all" experiment. And (iii) internal mixing creates an interdependency between different aerosol sources, meaning that the aerosol size distributions, optical scattering efficiency and hygroscopicity evolve differently depending on

the absolute and relative abundance of different mass components (sulphate, OC, BC and sea salt mass) and differing rates of new particle production via primary emission or nucleation.

To further understand which processes contribute most to the aerosol ERF, a series of additional control and perturbation experiments were conducted with aci processes selectively disabled. In these tests, the cloud droplet number concentrations (CDNCs) used for the calculation of cloud droplet effective radius (Reff) and/or autoconversion were prescribed via a 3D monthly-mean PI climatology constructed from the final 30 years of the *piClim-control* simulation. The resulting ERFs are

summarized in Fig. 7b. In one pair of simulations, CDNCs were prescribed for the Reff calculation to disable the so-called Twomey effect (Twomey, 1977). By comparison with the main *piClim-aer*/*piClim-control* experiment pair, this indicated a Twomey effect (ACI_Twomey) of $-0.70 \pm 0.05$ W m$^{-2}$. A similar pair with CDNCs prescribed only for the autoconversion process led to an estimate of $-0.31 \pm 0.05$ W m$^{-2}$ for the cloud lifetime effect (ACI_lifetime) (Albrecht, 1989). By prescribing CDNCs for both Reff and autoconversion both microphysical aci processes are disabled and only ari are included. A pair of

simulations with this setup provided an estimate for ERF$_{ari}$ of $-0.15 \pm 0.03$ W m$^{-2}$. The change in CRE in the ari-only experiment was only 0.02 W m$^{-2}$, which is not statistically significant at the 95 % confidence level and indicates that the semi-direct aerosol effect is small or approximately neutral in this model. To complete the breakdown, the method in Ghan (2013) was applied to the main *piClim-aer*/*piClim-control* experiment pair to derive the contribution from changes in "clean" (aerosol-free) CS radiation (RA_cs). This term was found to be $0.05 \pm 0.02$ W m$^{-2}$, and arises due to changes in surface albedo and atmospheric

temperature and humidity. This overall breakdown suggests an aerosol-cloud forcing of $-0.99 \pm 0.05$ W m$^{-2}$, with a roughly 70/30 split between the Twomey effect and aerosol effects on cloud cover and water content (lifetime and semi-direct effects).

The estimated aerosol-cloud forcing sits well within the 90 % likelihood ranges assessed by AR5 (-1.2 to 0.0 W m$^{-2}$) and Bellouin et al. (2020) (-2.0 to -0.35 W m$^{-2}$), although these broad ranges reflect the large uncertainties involved in constraining

global estimates with observations (e.g., Ghan et al., 2016). At present there are no definitive constraints on the proportionate contributions that the Twomey and other aerosol-cloud effects make towards the aerosol-cloud forcing, but recent observational evidence (Malavelle et al., 2017; Toll et al., 2017) supports the supposition that the Twomey effect is the dominant process and that some models overestimate cloud lifetime effects. Toll et al. (2017) indicated that HadGEM3 can indeed overestimate the cloud lifetime effect in marine stratocumulus, whereas Malavelle et al. (2017) found HadGEM3 to

correctly simulate only weak cloud lifetime effects for mixed cloud regimes over the North Atlantic. The UKESM1 estimate of ERF$_{ari}$ of $-0.15$ W m$^{-2}$ is within the 5-95 % confidence range ($-0.45 \pm 0.5$ W m$^{-2}$) from AR5 but slightly weaker than the range of values estimated by Bellouin et al. (2020) (-0.60 to -0.25 W m$^{-2}$). Possible reasons include the lack of nitrate, the relatively strong BC forcing compared to CMIP5 models, and a slight underestimation of aerosol optical depth (AOD) in PD simulations relative to some satellite products (Mulcahy et al., 2020).

 **4.4 Ozone (O₃) precursor (VOC, CO, and NOx) gases**

The ERF from emissions of O₃ precursors (VOC, CO, and NOx) excluding CH₄ is weakly positive ($0.21 \pm 0.04$ W m⁻²; Table 3; Figure 3), although spatially heterogeneous, with sparse regions of the globe showing a statistically significant ERF (Figure 3). O₃ precursor emissions affect the ERF both through changes in O₃, a GHG, and by changing tropospheric oxidants such as the hydroxyl (OH) radical, which in turn affect aerosols (Karset et al., 2018) and CH₄. Here, we explore the composition and forcings resulting from O₃ precursor emissions changes by comparing the *piClim-O3* simulation with *piClim-control* (Table 1) and the separate effects of NOx and VOC/CO emissions changes using the *piClim-NOx* and *piClim-VOC* simulations, respectively.

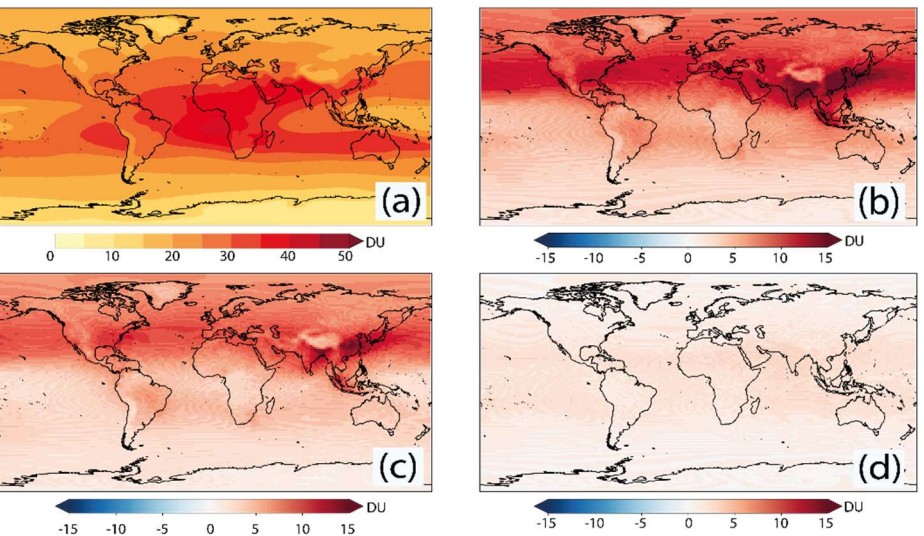

**Figure 9: Global distributions of tropospheric O₃ column (a) in *piClim-control*. Differences with respect to *piClim-control* from *piClim-O3*, *piClim-NOx* and *piClim-VOC* can be seen in panels b), c) and d), respectively. Units in Dobson Units (DU).**

**4.4.1 Tropospheric Ozone (O₃) Changes**

The tropospheric O₃ column for the *piClim-control* simulation, and tropospheric O₃ column differences between *piClim-control* and the *piClim-O3*, *piClim-NOx* and *piClim-VOC* simulations are shown in Fig. 9. O₃ levels in the troposphere are the result of competing production and loss processes. Production occurs in the presence of NOx and VOC/CO, with most regions of the troposphere being NOx-limited. PI tropospheric column O₃ values (Fig. 9a) show maxima over central Africa and the Eastern tropical Pacific, the result of relatively large emissions of NOx from soil, biomass burning and lightning in this region, and minima over regions remote from NOx sources, such as the Western equatorial Pacific (where O₃ loss is efficient), consistent with Young et al. (2018). Tropospheric column O₃ is also low over regions of high surface elevation where the atmospheric column is shallower.

Tropospheric $O_3$ column values increase when the model is perturbed with the larger PD $O_3$ precursor emissions in *piClim-O3*, with the largest increases in the northern hemisphere, particularly in southern and eastern Asia. The tropospheric $O_3$ burden increases from 280.9 Tg in *piClim-control* to 355.5 Tg in *piClim-O3*. As can be seen in Fig. 9, the dominant driver of these changes are increased NOx emissions, and there is a similar pattern of changes in the *piClim-O3* and *piClim-NOx* simulations, although the change in $O_3$ is smaller in the latter; the burden in *piClim-NOx* is 337.5 Tg. Small increases in $O_3$ are modelled in *piClim-VOC*, with some hotspots in regions such as South East Asia and the $O_3$ burden increases to 296.8 Tg.

The tropospheric $O_3$ burden difference between *piClim-O3* and *piClim-control* of 74.6 Tg is slightly larger than the sum of the individual $O_3$ burden changes in *piClim-NOx* (56.6 Tg) and *piClim-VOC* (15.9 Tg) which total 72.5 Tg. However, these differences due to the non-linear nature of tropospheric chemistry are small, on the order of < 5 %. Similar behaviour is seen in the patterns of the tropospheric $O_3$ column difference between *piClim-control* and other experiments.

### 4.4.2 Tropospheric $O_3$ Stratospherically-Adjusted Radiative Forcing (SARF)

As seen from earlier sections, it can be difficult to compare ERFs estimated from these experiments against estimates of SARF (e.g., Stevenson et al., 2013; Myhre et al., 2013a), due to the inclusion of indirect forcings and/or RAs other than the stratospheric temperature adjustment although Shindell et al. (2013b) and Skeie et al. (2020) noted that for $O_3$, SARF and ERF estimates are comparable. Nevertheless, in order to compare against Stevenson et al. (2013), we estimate the tropospheric $O_3$ SARF from the *piClim-CH4*, *piClim-VOC*, and the *piClim-NOx* experiments by adopting a radiative kernel approach (e.g., Soden et al., 2008). This involves applying the tropospheric $O_3$ radiative kernel from Rap et al. (2015) to the diagnosed change in tropospheric $O_3$ (using the 150 ppbv $O_3$ isoline in *piClim-control* as a tropospheric mask, as used in Young et al., 2013, Stevenson et al., 2013, and Rap et al., 2015) to calculate a SARF. While the ERF captures changes in radiative fluxes at the TOA due to whole-atmosphere responses, here, we mask off the stratosphere and focus *solely* on the tropospheric $O_3$ response. In this way, we can directly compare against the best estimate of the 1850-2010 tropospheric $O_3$ SARF of 364 mW m$^{-2}$ by Stevenson et al. (2013) and quantify the contribution of different $O_3$ precursors (including CH$_4$) to the change in tropospheric $O_3$ and its SARF. We can also compare with more recent estimates from Checa-Garcia et al. (2018) and Yeung et al. (2019).

Figure 10 shows the global distribution of the tropospheric $O_3$ SARF from *piClim-CH4*, *piClim-NOx*, and *piClim-VOC* experiments and their sum using the kernel approach. It shows that the tropospheric $O_3$ SARF is strongest over the northern hemisphere sub-tropics and weakest over the southern hemisphere high latitudes. The strongest SARF occurs in regions of warm surface temperatures and high albedo, coinciding with the largest tropospheric $O_3$ change (Shindell et al., 2013a). As was the case in Stevenson et al. (2013), the SARF is weaker over regions of high altitude (e.g., Tibetan Plateau) due to there being less $O_3$ column aloft to absorb in the LW. The tropospheric $O_3$ SARF from the kernel approach is 414 mW m$^{-2}$. However, the increase in tropospheric $O_3$ (and its resulting SARF) are offset by decreases due to ODSs (e.g., Søvde et al., 2011; Søvde et al., 2012; Shindell et al., 2013a). Applying the kernel method to the diagnosed decrease in tropospheric $O_3$ from the *piClim-*

*HC* experiment (Sect. 4.2.3), we find a SARF offset of -101 mW m$^{-2}$. Although larger in magnitude by nearly a factor of 2 than the estimates from Søvde et al. (2011; 2012) and Shindell et al. (2013a) due to the strong O$_3$ depletion in UKESM1 (Keeble et al., 2019; Skeie et al., 2020; Morgenstern et al., 2020), it reduces our original estimate to 313 mW m$^{-2}$. This revised estimate is within the 30 % uncertainty of the Stevenson et al. (2013) estimate, albeit lower than their central estimate by 13 %. It is also consistent with a number of other estimates: the CMIP6 historical O$_3$ dataset (312 mW m$^{-2}$ from Checa-Garcia et

al., 2018), a recent study in which observational isotopic data was used as a constraint on historical increases in tropospheric O$_3$ (330 mW m$^{-2}$ derived from the GEOSChem model in Yeung et al., 2019), and a parametric model based on multi-model source-receptor relationships (290 ± 3 mW m$^{-2}$ from Turnock et al., 2019). However, in masking off the stratosphere, the UKESM1 estimate for the O$_3$ SARF attributable to O$_3$ precursors is likely to be underestimated. For example, Søvde et al. (2011; 2012) estimate that approximately 15 % of the O$_3$ response from changes in CH$_4$ and other O$_3$ precursors may be in the

stratosphere, and hence not considered here.

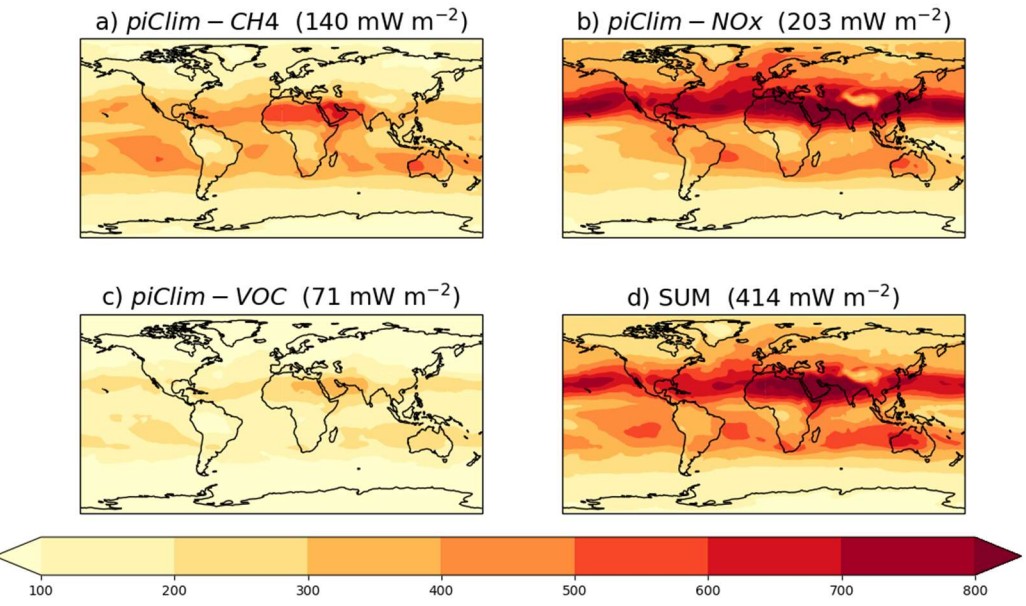

**Figure 10: Global distribution of tropospheric O$_3$ SARF diagnosed from a) *piclim-CH4*, b) *piClim-NOx*, c) *piClim-VOC*, and d) their sum relative to *piClim-control*, based on the diagnosed change in tropospheric O$_3$ and the tropospheric O$_3$ radiative kernel of Rap et**
**al. (2015). Global mean values are included. Units in mW m$^{-2}$.**

In considering the attribution of the tropospheric O$_3$ SARF to its precursors, initial estimates (Fig. 10; Table 5) indicate that it is predominantly NOx driven (49 %), followed by CH$_4$ (34 %), with the smallest contribution from VOCs and CO (17 %). This is qualitatively consistent with Stevenson et al. (2013). However, as outlined in detail in that paper, these estimates do
not account for potential CH$_4$ (and O$_3$) changes that would occur were these experiments driven by CH$_4$ emissions rather than

concentrations (e.g., Shindell et al., 2005). Taking the same approach as Stevenson et al. (2013), we calculate an equilibrium CH$_4$ concentration for the perturbation experiments, using the total CH$_4$ lifetime relative to *piClim-control* based on the following (Fiore et al., 2009):


$$[CH_4]_{piClim-X} = [CH_4]_{piClim-cont} * \left(\frac{\tau_{piClim-X}}{\tau_{piClim-cont}}\right)^f, \qquad (9)$$

where $[CH_4]_{piClim-X}$ is the global mean equilibrium CH$_4$ concentration in the *piClim-X* experiments, $[CH_4]_{piClim-cont}$ is the prescribed global mean CH$_4$ concentration in *piClim-control*, $\tau_{piClim-cont}$ and $\tau_{piClim-X}$ are the whole-atmosphere CH$_4$ lifetimes in the *piClim-control* and *piClim-X* perturbation experiments, respectively. The CH$_4$-OH feedback factor (Prather,

1996) is denoted by *f* and is defined as:

$$f = \frac{1}{1-s}, \qquad (10)$$

where *s*, called the sensitivity coefficient, is calculated from the following:


$$s = \frac{\delta ln\, \tau}{\delta\, ln\, [CH_4]}. \qquad (11)$$

Using the whole-atmosphere CH$_4$ burden and its removal by OH, the whole-atmosphere CH$_4$ lifetime is 8.1 and 9.8 years in *piClim-control* and *piClim-CH4,* respectively, when adjusted for stratospheric removal (120 yr lifetime) and soil uptake (160

yr lifetime). From Eqns. (10) and (11), the UKESM1 feedback factor *f* is 1.28, consistent with the range of other estimates (Prather et al., 2001; Shindell et al., 2005; Fiore et al., 2009; Stevenson et al., 2013; Voulgarakis et al., 2013; Turnock et al., 2018) and within 5 % of the observationally constrained best estimate of 1.34 (Holmes et al., 2013). Then, using the equilibrium minus prescribed difference in surface CH$_4$ concentrations (Table 5), we calculate the additional tropospheric O$_3$ SARF in the *piClim-CH4*, *piClim-NOx* and *piClim-VOC* experiments by applying the tropospheric O$_3$ radiative kernel (Rap et al., 2015) to

the O$_3$ response derived from scaling the (*piClim-CH4* minus *piClim-control*) O$_3$ response based on the relationship in Turnock et al. (2018); this additional contribution to the tropospheric O$_3$ SARF is shown in Table 5.

Changing from a concentration-based perspective to an emissions-based view increases the tropospheric O$_3$ SARF from 414 to 469 mW m$^{-2}$ (an increase of 13 %), which agrees better with the central estimate from Stevenson et al. (2013) once the offset

by ODSs is accounted for (-101 mW m$^{-2}$). It also changes the relative contributions of the different O$_3$ precursors. The contribution of CH$_4$ now dominates (45 %), with NOx playing a smaller role (37 %), while the contribution from VOC/CO emission increases is relatively unchanged (18 %). These emissions-based contributions are well within the spread of estimates

from Stevenson et al. (2013) who quantified contributions from 6 of the ACCMIP models: $CH_4$ (44 ± 12 %), NOx (31 ± 9 %), and VOC/CO (25 ± 3 %) although there are some differences with Shindell et al. (2005) and Shindell et al. (2009). In both Shindell et al. studies, the contribution from NOx is lower at 15 ± 8 and 11 %. The difference may be due to the strong sensitivity of the $CH_4$ lifetime to NOx in the GISS model compared with other models (Wild et al., 2020) and/or could be due to differences in VOC chemistry; Archibald et al. (2010) showed that the response of OH to increasing NOx strongly depends on the treatment of VOC chemistry. Nevertheless, this approach demonstrates the importance of an emissions-based view of climate forcing and is more directly relevant to policy makers than a concentration-based view (Shindell et al., 2005; Shindell et al., 2009).

| Pair | Trop. $O_3$ SARF / mW m$^{-2}$ | $\Delta CH_4$ concentration / ppbv | Additional Trop. $O_3$ SARF from $\Delta CH_4$ / mW m$^{-2}$ | Total Trop. $O_3$ SARF / mW m$^{-2}$ |
|---|---|---|---|---|
| *piClim-NOx* minus *piClim-control* | 203 | -246.2 | -31 | 172 |
| *piClim-VOC* minus *piClim-control* | 71 | 123.2 | 16 | 87 |
| *piClim-CH4* minus *piClim-control* | 140 | 533.0 | 70 | 210 |
| SUM minus *piClim-Control* | 414 | N/A | N/A | 469 |

**Table 5: Contribution to the tropospheric $O_3$ SARF from the different perturbation experiments (*piClim-NOx*, *piClim-VOC*, and *piClim-CH4*) relative to the pre-industrial control (*piClim-control*). Also shown is the absolute difference between the equilibrium and prescribed $CH_4$ concentrations in the different experiments and the resulting additional contribution to the total tropospheric $O_3$ SARF from the $CH_4$-driven response in $O_3$.**

### 4.4.3 ERF: Role of Other Oxidants and Aerosols

In addition to $O_3$ being a GHG, it is also important for secondary aerosol formation along with other oxidants: the OH radical, hydrogen peroxide ($H_2O_2$) and the nitrate ($NO_3$) radical. The OH radical in UKESM1 (Sellar et al., 2019) is involved in aerosol nucleation of gas-phase sulphuric acid ($H_2SO_4$) via the reaction with sulphur dioxide ($SO_2$), leading to new particle formation (Mulcahy et al., 2020). $O_3$ and $H_2O_2$ are important for $SO_2$ oxidation in cloud and aerosol droplets, creating sulphate aerosol mass but not number. Likewise, oxidation of monoterpenes by $O_3$, OH, and $NO_3$ determines the rate of formation of secondary organic aerosol (SOA) in UKESM1 (Kelly et al., 2018; Mulcahy et al., 2020) although it does not lead to new particle formation. Thus, sulphate aerosol alone, through changes in $O_3$, OH, and $H_2O_2$, has potentially important impacts on cloud and aerosol radiative properties. Indeed, Karset et al. (2018) found that oxidant changes in the CAM5.3-Oslo model alter the

relative importance of different chemical reactions, leading to changes in aerosol size distribution, cloud condensation nuclei (CCN), and the aerosol ERF.

Figure 11 (bottom row) shows a global distribution of OH at 1 km altitude in *piClim-control* and changes in the perturbation experiments relative to *piClim-control*. The *piClim-control* experiment shows an OH maximum in the equatorial humid regions, where photolytic production of OH from excited oxygen atoms ($O^1D$) and water vapour ($H_2O$) is at a maximum. In *piClim-O3*, OH increases throughout the northern hemisphere due to increases in $O_3$, which is the precursor of $O^1D$. These increases in OH are driven largely by increases in NOx. However, the *piClim-VOC* experiment shows the opposite behaviour - while VOC and CO emissions increases serve to increase $O_3$, they also remove OH via direct reaction with OH. This latter effect outweighs the small increase in $O_3$ in *piClim-VOC* and there are decreases in OH throughout the troposphere.

When OH is lower, we anticipate a decrease in the number of CCN, a decrease in CDNC, leading to larger cloud droplets (Twomey, 1977), and an increase in Reff. The middle rows of Fig. 11 show these effects at work. In *piClim-control*, the distribution of CDNC shows large values in equatorial regions, in regions of continental outflow and regions of deep convection. Large increases in CDNC are seen in *piClim-NOx*, with large decreases in *piClim-VOC*. These results reflect the changes in OH in these experiments - increases in OH lead to increases in CDNC, and vice versa, but it should be noted that the effect of OH on CDNC is seen over a larger region downwind, particularly in East Asia and over the North Atlantic. The impact of NOx emissions appears to dominate, given the similarity between *piClim-O3* and *piClim-NOx*.

While tropospheric column $O_3$ increases in both *piClim-NOx* and *piClim-VOC*, leading to positive contributions to the tropospheric $O_3$ SARF (Sect. 4.4.2), the ERFs are very different between the two simulations ($0.33 \pm 0.04$ W m$^{-2}$ for *piClim-VOC* and $0.03 \pm 0.04$ W m$^{-2}$ for *piClim-NOx* relative to *piClim-control*). Further, the spatial pattern of the ERF does not match those regions of largest $O_3$ changes. Instead, the dominant driver of the ERF differences are changes to OH and the subsequent impacts on aerosol particle formation and clouds. In particular, the positive tropospheric $O_3$ SARF in *piClim-NOx* (0.2 W m$^{-2}$) appears to be nearly completely offset by a negative aerosol forcing, largely from aci driven by changes in oxidants and aerosol nucleation; the contribution to the global mean ERF from ari is only $-0.03 \pm 0.01$ W m$^{-2}$, despite the strong regional changes in AOD at 550 nm (Fig. 11; top row). Similarly, the aerosol IRF in *piClim-VOC* is negligible (less than $0.01 \pm 0.01$ W m$^{-2}$). However, the forcing in *piClim-VOC* due to aci, particularly from the SW ΔCRE' component (Table 3), enhances the positive tropospheric $O_3$ SARF (0.07 W m$^{-2}$), leading to an ERF of $0.33 \pm 0.04$ W m$^{-2}$.

A negative SARF attributable to NOx emissions has been found in other studies (Shindell et al., 2009; Collins et al., 2010) from a balance between the direct $O_3$ response (positive SARF), the NOx-driven $CH_4$ response (negative SARF), and the subsequent $O_3$ response to $CH_4$ changes (negative SARF). Inclusion of ari and aci from sulphate and nitrate aerosol further increases the magnitude of the net negative forcing or cooling (Shindell et al., 2009; Collins et al., 2010). However, a study by

Fry et al. (2012) found that the chemistry response is sensitive to the location of the emissions, with so large an uncertainty that it is difficult to determine whether NOx emissions cause a warming or cooling. Indeed, other indirect effects such as NOx deposition to the terrestrial biosphere leading to fertilisation (Collins et al., 2010) and/or NOx-driven $O_3$ damage (Sitch et al., 2007; Collins et al., 2010) increase the uncertainty further through changes in $CO_2$. Thornhill et al. (2020) show that the ERF from changes in NOx emissions among the AerChemMIP models differ in both sign and magnitude. Here, the longer time-

scale $CH_4$ response to NOx emissions (Collins et al., 2010) is constrained, nitrate aerosol is neglected (Sellar et al., 2019; Mulcahy et al., 2020), and $CO_2$ is concentration-driven (Sellar et al., 2019). Nevertheless, in UKESM1, the negative forcing due to aci from sulphate aerosol offsets the positive NOx-driven $O_3$ SARF, leading to a negligible ERF overall.

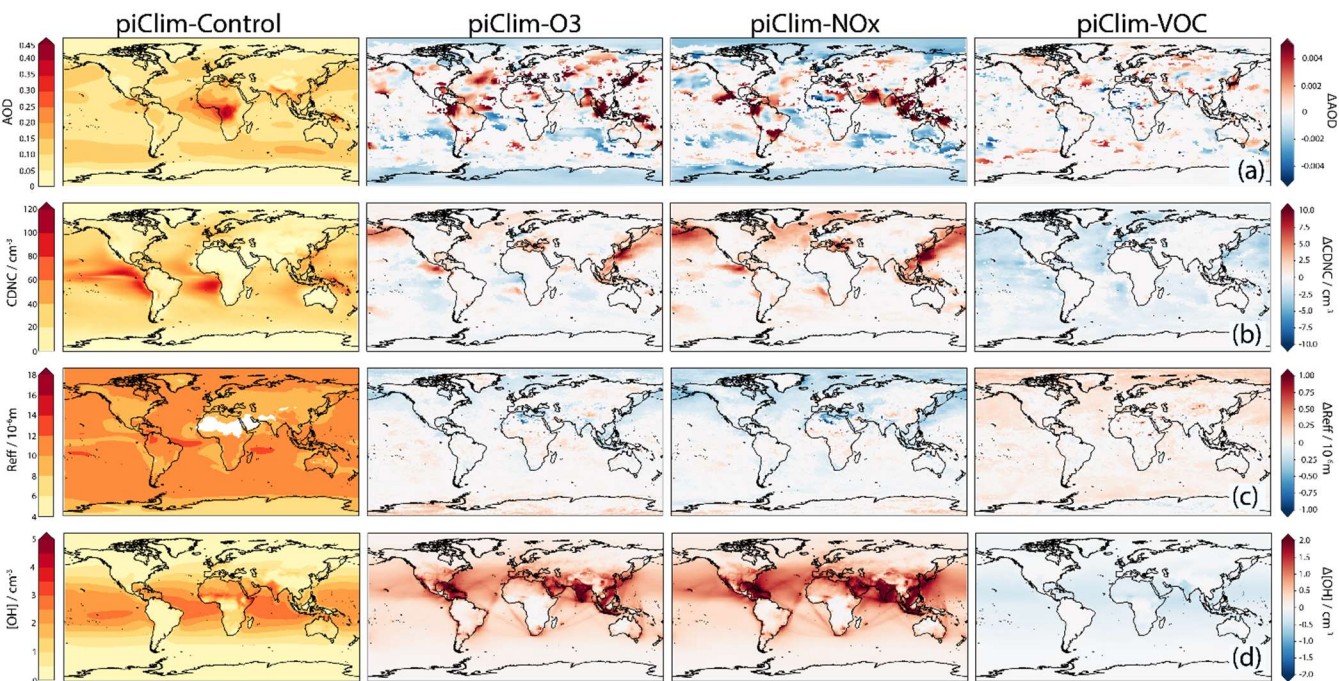

**Figure 11: Global distributions of AOD at 550 nm (row (a); left column), cloud droplet number concentration (CDNC; cm$^{-3}$) at 1 km altitude (row (b); left column), cloud droplet effective radius (Reff; μm) at 1 km altitude (row (c); left column) and OH number concentration (cm$^{-3}$) at 1 km altitude (row (d); left column) in *piClim-control*. Differences with respect to *piClim-control* from *piClim-O3*, *piClim-NOx* and *piClim-VOC* can be seen in the second, third and right hand columns, respectively.**

Previous work has also found a positive SARF from VOC and CO emissions, due to the combined indirect forcings by $O_3$ and $CH_4$ (Shindell et al., 2005; Forster et al., 2007); they estimate a global mean SARF of $0.21 \pm 0.10$ W m$^{-2}$ at the PD (year 1998) relative to the PI (year 1750) period. Via the same mechanisms as was the case with NOx, the magnitude of this SARF increases (to $0.25 \pm 0.04$ W m$^{-2}$ for the year 2000) when additional indirect forcings from sulphate, nitrate, and $CO_2$ are included (Shindell et al., 2009). More recently, Stevenson et al. (2013) found the SARF from VOC/CO emissions to be marginally higher, at 0.29

W m$^{-2}$, excluding aerosols, with contributions of 0.09, 0.08, and 0.12 W m$^{-2}$ from $O_3$, $CH_4$, and $CO_2$, respectively. The SARF

contribution from $O_3$ alone quantified here (87 mW m$^{-2}$; Table 5) is consistent with the Stevenson et al. (2013) estimate. Despite excluding the longer-term $CH_4$ and $CO_2$ responses, the ERF from VOC/CO emissions of $0.33 \pm 0.03$ W m$^{-2}$ is higher than previous estimates of SARF due to the additional positive contribution from aci driven by OH changes. However, Fry et al. (2014) found that the SARF from VOCs is sensitive to the location of emissions and could influence the strength of the contribution from aerosols. Interestingly, other AerChemMIP models show a negative ERF from VOC and CO emissions (Thornhill et al., 2020); these differences in sign of the ERF warrant further investigation.

As a result of the very different chemical response between NOx and VOC/CO emissions, both in terms of the magnitude of the $O_3$ changes and the different impacts on OH, aerosols and clouds, a comparison of the ERFs (Table 3) indicates that the ERF from *piClim-O3* is not a linear combination of that from *piClim-NOx* and *piClim-VOC* for the NET, CS and the CRE components. In particular, there are differences in the LWcs' and SW $\triangle$CRE' components (Table 3). These results clearly suggest that Earth System (ES) interactions, particularly chemistry-aerosol coupling, can strongly affect estimates of climate forcing. Here, these interactions alter the ERF from $O_3$ precursor emissions, while other studies (e.g., Shindell et al., 2009; Karset et al., 2018) show that they also affect estimates of anthropogenic aerosol forcing.

## 4.5 Other Forcings

### 4.5.1 Non-Methane Near-Term Climate Forcers (NTCFs)

The anthropogenic ERF due to $CH_4$, aerosols and $O_3$ abundances was identified as the main source of uncertainty in the total anthropogenic ERF since PI times (Myhre et al., 2013a). This is due to the uncertainty in the individual forcings (e.g., Bellouin et al., 2020) but the interaction between individual forcings, as well as the non-linear response of climate feedbacks due to aci (Feichter et al., 2004; Deng et al., 2016; Collins et al., 2017; Shim et al., 2019) may play a role. In this section, three experiments related to non-$CH_4$ NTCFs are discussed: the combined simulation (*piClim-NTCF*), which is identical to *piClim-control* except that aerosol and $O_3$ precursor emissions are set to PD (year 2014) levels and the single-perturbation runs, which change aerosol and aerosol precursor emissions only (*piClim-aer*), and $O_3$ precursor (VOC, CO, NOx) emissions only (*piClim-O3*) from PI to PD levels. More details are described in Table 1.

The ERF of non-$CH_4$ NTCFs is $-1.03 \pm 0.04$ W m$^{-2}$ (Table 3; Figure 12). The negative ERF results from the combination of a weak negative NETcs' component ($-0.03 \pm 0.03$ W m$^{-2}$) and a strong contribution due to the NET $\triangle$CRE' component ($-1.00 \pm 0.02$ W m$^{-2}$). The weak negative contribution in the CS (NETcs') is due to the negative SWcs' component ($-0.26 \pm 0.02$ W m$^{-2}$) being largely offset by the positive LWcs' component ($0.23 \pm 0.03$ W m$^{-2}$). The negative SWcs' component is correlated with changes in AOD at 550 nm (spatial correlation coefficient of $-0.44$) and is predominantly due to the aerosol IRF ($-0.29 \pm 0.01$ W m$^{-2}$). The LWcs' component is positive ($0.23 \pm 0.03$ W m$^{-2}$) due to both aerosols and $O_3$, but with the aerosol IRF only contributing $0.04 \pm 0.01$ W m$^{-2}$. The spatial variations in the global distribution of the LWcs' component are closely related to

land surface temperature (Ts) changes; the correlation coefficient is -0.78 with a statistical significance well over 99 %. Considering this good correlation, the increased LWcs' component is in response to the Ts change due to NTCFs. The negative SW ΔCRE' component (-0.92 ± 0.03 W m⁻²), however, dominates the ERF and is largely correlated with changes in cloudiness, with the spatial pattern correlation of -0.59 between the SW ΔCRE' component and cloud fraction.

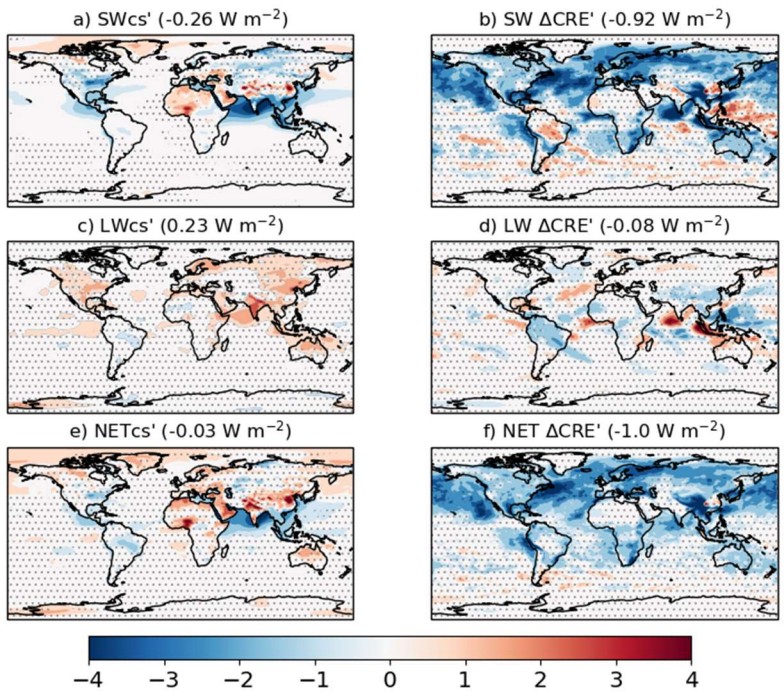

**Figure 12: Global distributions of the PD NTCF ERF components at the top of atmosphere (TOA) relative to the PI period, i.e., *piClim-NTCF* minus *piClim-Control*; (a) SWcs', (b) SW △CRE', (c) LWcs', (d) LW △CRE', (e) NETcs', and (f) NET △CRE', based on Eqn. (8). Global mean values are shown in brackets. Regions where the ERF components are outside the 95 % confidence level are stippled. Units in W m⁻².**

As was the case for other ERFs (e.g., GHGs, aerosols, and $O_3$ precursor gases), the range of perturbation simulations carried out with UKESM1 enables the role of non-linear interactions to be investigated. The total GHG ERF was found to be equal to the sum of the individual GHG ERFs (Section 4.2.5) but non-linearities were evident for the aerosol (Section 4.3) and the $O_3$ precursor (Section 4.4) ERFs. As a result, this study also attempts to estimate the effects of the non-linear interactions between chemistry and aerosols on the combined aerosol and $O_3$ precursor ERF. When combined (aerosol and aerosol and $O_3$ precursor emissions), their interaction may induce an effect that differs from the sum of the individual single ERFs. The ERFs do not add linearly, particularly in the SWcs' and NETcre' components (Table 3). Firstly, we calculate the aerosol IRFs using Eqn.

(7). In *piClim-NTCF,* the net (SW + LW) aerosol IRF is -0.24 ± 0.01 W m$^{-2}$, which is more negative than the sum of the aerosol IRFs in *piClim-aer* (-0.15 ± 0.01 W m$^{-2}$) and *piClim-O3* (-0.02 ± 0.01 W m$^{-2}$). This is due to the sulphate aerosol loading being higher in *piClim-NTCF* relative to *piClim-aer* by up to 3.4 (27) % globally (regionally), and driven by changes in oxidants due

to the PD levels of O$_3$ precursors. Secondly, the NET $\triangle$CRE' component contributes to the non-linearity in the ERFs. The NET $\triangle$CRE' component is more negative in *piClim-NTCF* (-1.00 ± 0.02 W m$^{-2}$) than in the sum of *piClim-aer* (-1.0 ± 0.02 W m$^{-2}$) and *piClim-O3* (0.08 ± 0.02 W m$^{-2}$). This is primarily the result of differences in the SW $\triangle$CRE' component, again driven by the higher sulphate loading in *piClim-NTCF* relative to *piClim-aer*.

Although there is no direct coupling between aerosols and photolysis in UKESM1 (Archibald et al., 2020), aerosol-mediated cloud adjustments result in SW reduction and surface cooling. These changes impact thermal and photochemical reactions leading to reduced photolysis rates (O$_3$ + $hv$ → O($^1$D) + O$_2$) in the lower troposphere, while enhancing the photolysis rate of O$_3$ in the upper troposphere (black and blue dotted lines in Fig. 13), leading to a reduction in surface O$_3$ in *piClim-NTCF* relative to the sum. However, despite differences in O$_3$ (not shown), the LWcs' components of the ERFs appear to add linearly.


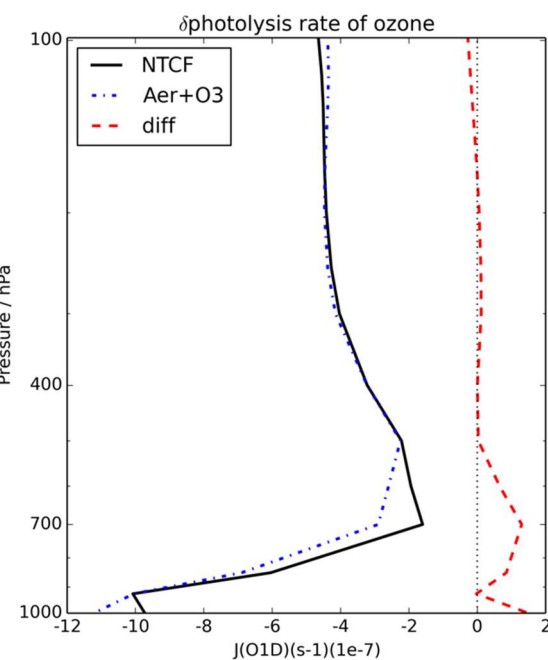

**Figure 13: Vertical distribution of changes in j(O$^1$D) photolysis rates (10$^{-7}$ s$^{-1}$) in *piClim-NTCF* (black solid line) and the sum (blue dotted line) of *piClim-aer* and *piClim-O3* relative to *piClim-control* and the difference between them (red dashed line).**

**4.5.2 Land Use**

Land use change causes radiative forcing primarily by changing surface albedo; croplands and pastures have higher albedos than forests and are less able to mask the high albedo of snow cover. Increased albedo leads to a negative SWcs' component (-0.30 ± 0.02 W m$^{-2}$; Table 3) that is damped by the SW △CRE' (0.09 ± 0.03 W m$^{-2}$; Table 3) and the inclusion of small positive LWcs' (0.02 ± 0.03 W m$^{-2}$; Table 3) and LW △CRE' (0.03 ± 0.01 W m$^{-2}$; Table 3) terms leads to a net ERF of -0.17

± 0.04 W m$^{-2}$. The land use ERF of UKESM1 falls within the 'very likely' range (-0.15 to -0.10 W m$^{-2}$) of AR5 (Myhre et al., 2013a). However, the AR5 estimate is valid for land use change since 1700, not 1850 as in our estimate. Andrews et al. (2017) used HadGEM2-ES, UKESM1's predecessor, to calculate a 1700-1860 land use ERF of -0.1 W m$^{-2}$. If we apply this adjustment to our estimate, we find an ERF of -0.27 W m$^{-2}$, which is outside of the AR5 range. The UKESM1 land use ERF (-0.17 ± 0.04 W m$^{-2}$) is reduced relative to HadGEM2-ES which produced an 1860-2005 land use ERF of -0.40 W m$^{-2}$ (Andrews et al.,

2017). Robertson (2019) showed that, even in the absence of snow cover, the albedo response to land use change in HadGEM2-ES is stronger than observed: at the location of deforestation, the surface SW response to total deforestation was generally 5 W m$^{-2}$ too large. It is likely that the bias found in HadGEM2-ES still exists in UKESM1.

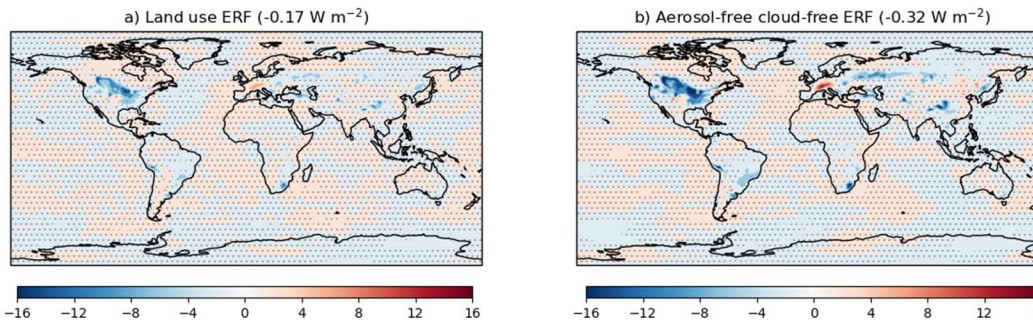


**Figure 14: Global distribution of the present-day (PD; year 2014) ERF relative to the pre-industrial (PI; year 1850) period of a) land use ERF and b) the contribution from the cloud-free and aerosol-free component. Global mean values are included and areas are stippled, where the ERF and/or its component are not statistically significant from zero at the 95 % confidence interval. Units in W m$^{-2}$.**


While the global mean land use ERF is small, regionally it can be the dominant source of the anthropogenic ERF. Figure 14 shows the distribution of the land use ERF and its cloud-free and aerosol-free, component which emphasizes the surface flux contribution. The ERF is mostly confined to regions of land use change, with deforestation in North America and western Eurasia causing a negative ERF and increased tree cover in central Europe causing a positive ERF. Alongside the distribution

of land use change itself, the magnitude of the ERF is larger in the mid-latitudes than the tropics, because the masking of snow cover by trees greatly increases the albedo response to changes in tree cover. The large regions of negative ERF in the northern

mid-latitudes are consistent with previous model studies (e.g., de Noblet-Ducoudre et al., 2012); they are caused by agricultural expansion and the forcing is expected to have gradually increased over the period 1850-1970 in North America and over the period 1900-1980 in Eurasia. The seasonality of leafiness, snow cover and insolation causes the land use ERF to be largest in northern hemisphere spring (March to May mean of -0.28 ± 0.11 W m$^{-2}$) and smallest in autumn (September to November mean of -0.08 ± 0.09 W m$^{-2}$).

The land use ERF is calculated by modifying three land-surface fields: land-cover, leaf area index (LAI) and canopy height. The values of the modified fields are taken from a coupled UKESM1 simulation that is only subject to historical land use change. In the coupled configuration of UKESM1 (Sellar et al., 2019), the three land surface fields are prognostic fields calculated using a dynamic global vegetation model (DGVM), while in the atmosphere-only ERF configuration they are prescribed fields. This choice was made because the land-surface fields can be very slow to respond to changes in land use change and RF. In the coupled simulation of historical climate change including all forcings (*historical*), the land surface fields respond to changes in $CO_2$, climate and land use, but if we use the land surface fields from this simulation, we find no substantial change in the land use ERF (-0.20 W m$^{-2}$).

In addition to the ERF caused by albedo changes, land use change can alter climate via a number of other mechanisms. Land use change alters the land carbon sink. In particular, deforestation emits $CO_2$ to the atmosphere, and so some fraction of the $CO_2$ ERF is attributable to land-use change. Land use change also affects surface climate via changes in roughness length and transpiration. These non-radiative mechanisms usually drive larger temperature changes than the albedo response and both the non-radiative mechanisms and the $CO_2$ forcings tend to oppose the albedo response.

**4.5.3 Total Anthropogenic ERF**

As noted above, historical climate change has been driven by a wide range of anthropogenic activities that act together, alongside natural changes, to perturb the Earth's radiation balance. The total anthropogenic ERF is, therefore, a key metric in understanding observed and modelled changes in the climate system since the PI era. These various anthropogenic drivers are not necessarily independent of each other and it is therefore worthwhile calculating the total anthropogenic ERF from a separate timeslice simulation including all perturbations together (*piClim-anthro*; Pincus et al., 2016). We completed a dedicated timeslice simulation with all concentrations, emissions, and land use set to 2014 levels, including all GHG concentrations ($CO_2$, $N_2O$, $CH_4$, HCs – both ODSs and non-ODSs), $O_3$ precursors (VOC, CO, NOx), land use changes, and anthropogenic aerosol or aerosol precursor emissions ($SO_2$, OC, BC). This experiment was not proposed in AerChemMIP (Collins et al., 2017) but is included here as part of RFMIP (Pincus et al., 2016). The main difference is that atmospheric chemistry in UKESM1 is fully interactive whereas other models participating in RFMIP (e.g., Andrews et al., 2019) use the CMIP6 $O_3$ dataset to represent changes in tropospheric and stratospheric $O_3$ and prescribe oxidants for secondary aerosol formation.

The UKESM1 *piClim-anthro* experiment leads to an ERF of 1.76 ± 0.04 W m$^{-2}$ (Table 3), which is dominated by a positive LWcs' component due to GHGs and partially offset by a negative SWcs' component due to aerosols and negative contributions from the SW ΔCRE' and LW ΔCRE' components (Table 3). This ERF is a little lower than the equivalent estimate from HadGEM3-GA7.1, which was 1.81 W m$^{-2}$ (Andrews et al., 2019). The UKESM1 estimate is also lower than the median estimate from CMIP5 models assessed in AR5, which equates to approximately 1.9 W m$^{-2}$ after adjustment to the reference

period of 1861-1880 to 2010-11 (Andrews and Forster, 2020). AR5 also provided an overall central estimate of 2.2 W m$^{-2}$ and a 5-95 % confidence range of 1.0 to 3.2 W m$^{-2}$ (after adjustment to the same reference period as above in Andrews and Forster, 2020) taking into account multiple streams of evidence. Andrews and Forster (2020) re-evaluated this as 2.3 W m$^{-2}$ with a narrower range of 1.7 - 3.0 W m$^{-2}$ [5 - 95 % confidence] using a combination of atmospheric model outputs and observational constraints. The lower bound of this range was reduced to 1.5 W m$^{-2}$ if larger uncertainties were assumed for the climate

feedback parameter or for the global-mean surface temperatures anomalies used to constrain the forcing. The UKESM1 estimate of 1.76 ± 0.04 W m$^{-2}$ is therefore within the original uncertainty range given by AR5 and just within the range proposed in Andrews and Forster (2020).

    There are several factors that contribute to the relatively low estimate of anthropogenic ERF in UKESM1. Firstly, the

anthropogenic aerosol ERF in UKESM1 (and HadGEM3-GC3.1) is -1.09 ± 0.04 W m$^{-2}$ (well within the uncertainty range of Bellouin et al., 2020) and offsets a major portion of the positive GHG ERF (2.92 ± 0.04 W m$^{-2}$). Secondly, the ERF from *piClim-HC* is negative (-0.18 ± 0.04 W m$^{-2}$) due to a strong O$_3$ response and connected aerosol-mediated cloud adjustments (Morgenstern et al., 2020). Thirdly, adjustments in vegetation lead to an appreciable negative ERF from land use changes (-0.17 ± 0.04 W m$^{-2}$). The stronger negative ERF from *piClim-HC* balanced by a stronger positive ERF from *piClim-CH4* (0.97

± 0.04 W m$^{-2}$) due to indirect effects and the positive tropospheric RA term (Thornhill et al., 2020) largely explain why the UKESM1 estimate is only ~0.05 W m$^{-2}$ lower than HadGEM3-GA7.1. As shown in Fig. 3i, the negative contributions more than offset the positive GHG ERF in certain regions. For instance, the anthropogenic ERF is negative over large parts of North America and Asia (from a combination of land use change and aerosol ERFs; see Figs. 3c and 3g) and at southern high latitudes (from O$_3$ depletion due to ODSs, see Fig. 3a; Section 4.2.3). In contrast, the anthropogenic ERF is strongly positive over the

tropics and southern hemisphere sub-tropics (Fig. 3e) where the direct radiative effect of GHGs dominates.

    The couplings between chemistry, aerosol and land surface processes included in UKESM1 increase the possibilities for non-linear interactions among the various anthropogenic forcing agents. However, these apparently have little net overall effect on the total forcing (1.76 ± 0.04 W m$^{-2}$), which is within the uncertainty of the sum of the forcings (1.87 ± 0.08 W m$^{-2}$) from the

four separate groups that it includes (GHGs, aerosol, O$_3$ precursors, and land use). The two estimates are not statistically different given the standard error on these is around 0.03 - 0.04 W m$^{-2}$ (Table 3). This does not imply that the forcings act independently as it is possible that competing non-linear interactions cancel in this particular case. As shown in Sect. 4.5.1, the aerosol and O$_3$ precursor emission ERFs did not add linearly.

## 5 Conclusions

Quantifying effective radiative forcings (ERFs) from anthropogenic perturbations to the Earth System (ES) is important for understanding changes in climate since the pre-industrial (PI) period. In this study, we have quantified and analysed a wide range of present-day (PD) anthropogenic ERFs with the UK's Earth System Model (ESM), UKESM1 (Sellar et al., 2019). ERFs have been shown to be a more useful metric for evaluating and comparing the relative roles of diverse forcing agents due to the relationship to global-mean temperature and other impacts that scale with it. In particular, by quantifying ERFs

within a full ESM, this study addresses gaps in previous assessments in which rapid adjustments (RAs) were neglected and enables the role of indirect contributions to ERF estimates and various climate-chemistry-aerosol-cloud interactions to be investigated.

We find that the change in carbon dioxide ($CO_2$) concentration since the PI period exerts an ERF of $1.89 \pm 0.04$ W m$^{-2}$,

consistent with previous estimates, making it the single largest contributor to the total anthropogenic ERF. However, UKESM1 appears to have a more pronounced surface warming adjustment associated with the physiological forcing by $CO_2$ than its successor, HadGEM2-ES. The nitrous oxide ($N_2O$) ERF quantified here ($0.25 \pm 0.04$ W m$^{-2}$) is higher than previous estimates (e.g., Hansen et al., 2005; Myhre et al., 2013a) but is consistent with the more recent Aerosol and Chemistry Model Intercomparison Project (AerChemMIP) multi-model ERF assessment (Thornhill et al., 2020).


The PD-PI change in methane ($CH_4$) concentration leads to an ERF of $0.97 \pm 0.04$ W m$^{-2}$, with the majority of the ERF due to the clear-sky longwave (LWcs') component. Given the inclusion of interactive chemistry in UKESM1, the ERF is larger than other estimates of direct $CH_4$ forcing as a result of indirect effects. It is also at the high end of the range of estimates from AerChemMIP (Thornhill et al., 2020), partly due to these indirect effects but partly due to the additional positive contribution

from the tropospheric RA term. O'Connor et al. (2019) apportion the $CH_4$ ERF between direct and indirect contributions as well as considering an emission-based perspective.

The ERF from the change in ozone ($O_3$) depleting substances (ODSs) is $-0.18 \pm 0.04$ W m$^{-2}$. Using a range of AerChemMIP models and observed $O_3$ trends as a constraint, Morgenstern et al. (2020) estimate that the UKESM1 ERF is too strongly

negative; this is the result of a high $O_3$ bias in the PI period and a strong response to increasing ODSs in UKESM1 relative to other models (Keeble et al., 2020; Morgenstern et al., 2020). Considering all greenhouse gases (GHGs) together, we quantify an ERF of $2.92 \pm 0.04$ W m$^{-2}$, less than the 3.09 W m$^{-2}$ estimate from the physical model HadGEM3-GC31, due to indirect effects. There is also no evidence of non-linearity between the combined GHG ERF and the sum of the individual GHG ERFs.

The new GLOMAP-mode aerosol scheme in UKESM1 (Mulcahy et al., 2020) leads to an intermediate sized negative ERF ( -$1.09 \pm 0.04$ W m$^{-2}$) due to strong aerosol-cloud interactions (aci), despite strong absorption by black carbon (BC) and relatively

weak negative aerosol instantaneous radiative forcing (IRF) from aerosol-radiation interactions (ari). Internal mixing and chemical interactions included in the new aerosol scheme mean that neither aerosol IRF nor aci are linear (sulphate, organic carbon (OC), BC interact with one another) making the aerosol ERF less than the sum of the individual speciated aerosol ERFs.

Examining tropospheric $O_3$ stratospherically-adjusted radiative forcing (SARF) alone, results from UKESM1 suggest that the contribution from $CH_4$ dominates (45 %), with nitrogen oxides (NOx) and volatile organic compound (VOC)/carbon monoxide (CO) contributing 37 and 18 %, respectively. These emissions-based contributions are well within the spread of estimates from the Atmospheric Chemistry and Climate Model Intercomparison Project (ACCMIP; Stevenson et al., 2013): $CH_4$ (44 ± 12 %), NOx (31 ± 9 %), and VOC/CO (25 ± 3 %) although there is disagreement with other studies. Changes in oxidants, driven by the changes in $O_3$ precursor emissions, lead to an indirect aerosol ERF from aci, which either supplements or offsets the positive SARF from tropospheric $O_3$, leading to global mean ERFs of 0.33 ± 0.04 and 0.03 ± 0.04 W m$^{-2}$ for VOC/CO and NOx emissions changes, respectively. However, there appears to be disagreement across the AerChemMIP models on the sign and/or magnitude of the $O_3$ precursor ERFs and further analysis to understand what is driving these differences is required.

The aerosol and $O_3$ precursors (called near-term climate forcers (NTCFs) in the context of AerChemMIP) together exert an ERF of -1.03 ± 0.04 W m$^{-2}$, which is mainly due to changes in the cloud radiative effect (CRE). There is also evidence of non-linearity in the ERF between the combined *piClim-NTCF* experiment and the sum of the individual *piClim-aer* and *piClim-O3* experiments; this is mainly evident in the shortwave clear-sky (SWcs') and shortwave cloud radiative effect (SW ΔCRE') components. Land use (LU) change since the PI period has also exerted a negative ERF, estimated to be -0.17 ± 0.04 W m$^{-2}$. However, this estimate is outside the range from previous estimates, and is most likely due to too strong an albedo response.

Historical climate change has been driven by a wide range of anthropogenic activities that act together, alongside natural changes, to perturb the Earth's radiation balance. As a result, the total anthropogenic ERF is a key metric in understanding observed and modelled changes in the climate system since the PI era. The estimate of the total anthropogenic ERF from UKESM1 is 1.76 ± 0.04 W m$^{-2}$, which is relatively low compared to previous assessments; this is mainly due to an intermediate negative aerosol ERF, a modest negative land use ERF and strong stratospheric $O_3$ depletion. Although it may be biased low, that combined with high climate sensitivity (Andrews et al., 2019) means that UKESM1 reproduces well the historical global mean warming over the 1850 – 2014 period (Sellar et al., 2019).

In addition to quantifying anthropogenic ERFs with an ESM, this study and other studies (e.g., Morgenstern et al., 2020; O'Connor et al., 2019) show the importance of indirect contributions to the ERFs and climate-aerosol-chemical interactions. There are substantial interactions between GHGs, stratospheric and tropospheric $O_3$, and aerosols, some of which act non-linearly. These effects demonstrate the importance of including ES interactions when quantifying ERFs. In particular, we

suggest that RAs included in the definition of ERF should include chemical as well as physical adjustments, consistent with Ramaswamy et al. (2019). They concluded in their recent assessment that although the radiative forcing concept is simple, it needs to increasingly account for the complex relevant processes in the Earth System.

**Appendix A**

Table A1 shows the main differences between the atmosphere-only configurations of HadGEM3-GC3.1 (called HadGEM3-GA7.1) and UKESM1 used to calculate present day effective radiative forcings (ERFs) in Andrews et al. (2019) and in this study, respectively. The implementation of the CMIP6 inputs, as applied to both models, is described in detail in Sellar et al. (2020).


| Model Feature | HadGEM3-GC3.1 (Kuhlbrodt et al., 2018; Williams et al., 2018) | UKESM1 (Sellar et al., 2019) |
|---|---|---|
| Atmosphere Resolution | N96L85[a] or N216L85 | N96L85 |
| Vegetation, land use, and dust | Prescribed using 9 surface types, including 5 plant functional types (PFTs) | Prescribed from UKESM1 climatology using 17 surface types, including 9 natural PFTs and 4 crop/pasture PFTs; Dust tuning |
| Biogenic volatile organic compound (VOC) emissions | Prescribed | Monoterpene and isoprene emissions interactive; Other biogenic VOCs prescribed |
| Primary Marine Organic Aerosol (PMOA) emissions | Not included | Interactive, using prescribed surface water chlorophyll climatology from UKESM1 |
| Dimethyl Sulphide (DMS) surface water concentration | Prescribed from Lana et al. (2011) | Prescribed from UKESM1 climatology |
| DMS emissions | Interactive, with scaling of 1.7 | Interactive, with no scaling |
| Sulphur dioxide ($SO_2$) anthropogenic emissions | Split between surface and "high level" (0.5 km) dependent on sector | Added at surface only |
| Oxidants for secondary aerosol formation | Prescribed OH, $HO_2$, $H_2O2$, $NO_3$, and $O_3$ fields | Interactive |
| Long-lived greenhouse gases (LLGHGs) in radiation scheme | Uniform mass mixing ratio prescribed | Differs depending on the LLGHG – see Section 3 |
| $O_3$ in radiation scheme | Prescribed, with vertical re-distribution scheme | Interactive |

**Table A1: Differences between the atmosphere components of the UK's models for CMIP6: HadGEM3-GC3.1 (called HadGEM3-GA7.1) and UKESM1. ªAtmosphere resolution of HadGEM3-GC3.1 used in the Tier 1 RFMIP simulations (Andrews et al., 2019).**

**Acknowledgements**

The development of the UK's Earth System Model, UKESM1, was supported by the Met Office Hadley Centre Climate Programme funded by BEIS and Defra (GA01101) and by the Natural Environment Research Council (NERC) national capability grant for the UK Earth System Modelling project, grant number NE/N017951/1. MD, GAF, CH, BJ, JPM, FMO'C, ER, and AW were funded by the Met Office Hadley Centre Climate Programme funded by BEIS and Defra (GA01101). GF, FMO'C and JCT also acknowledge the EU Horizon 2020 Research Programme CRESCENDO project, grant agreement

number 641816. OM and GZ were supported by the NZ Government's Strategic Science Investment Fund (SSIF) through the NIWA programme CACV. JW acknowledges support by the Deep South National Science Challenge (DSNSC), funded by the New Zealand Ministry for Business, Innovation and Employment (MBIE). BK, JS and SS were supported by the Korea Meteorological Administration Research and Development Program "Development and Assessment of IPCC AR6 Climate Change Scenario", grant agreement number 1365003000.


The authors acknowledge the contribution of NeSI high-performance computing facilities to the results of this research. New Zealand's national facilities are provided by the New Zealand eScience Infrastructure (NeSI) and funded jointly by NeSI's collaborator institutions and through MBIE's Research Infrastructure programme. This work used Monsoon2, a collaborative High Performance Computing facility funded by the Met Office and the Natural Environment Research Council. This work

also used the NEXCS High Performance Computing facility funded by the Natural Environment Research Council and delivered by the Met Office. This work used JASMIN, the UK collaborative data analysis facility; the authors are grateful for the provision of these facilities.

The authors thank T. Andrews for constructive comments on the manuscript. Last but not least, the authors wish to

acknowledge the huge effort from the UKESM1 core group in building and evaluating UKESM1 and making it available for use in RFMIP and AerChemMIP.

**Data Availability**

All of the data from the RFMIP and AerChemMIP simulations analysed in this study have been published on the Earth

System Grid Federation and the model Source ID is UKESM1-0-LL. Below is a table listing all of the corresponding dataset citations. In some cases, data from 2 realisations of the same experiment are available.

| Experiment ID | MIP | Data Citation(s) |
|---|---|---|

| | | |
|---|---|---|
| *piClim-control* | AerChemMIP/RFMIP | doi:10.22033/ESGF/CMIP6.6276 |
| *piClim-4xCO2* | RFMIP | doi:10.22033/ESGF/CMIP6.11061 |
| *piClim-N2O* | AerChemMIP | doi:10.22033/ESGF/CMIP6.9434 |
| *piClim-HC* | AerChemMIP | doi:10.22033/ESGF/CMIP6.9433 |
| *piClim-CH4* | AerChemMIP | doi:10.22033/ESGF/CMIP6.6229 |
| *piClim-GHG* | RFMIP | doi:10.22033/ESGF/CMIP6.11094 |
| *piClim-SO2* | AerChemMIP | Realisation 1: doi:10.22033/ESGF/CMIP6.9440<br>Realisation 2: doi:10.22033/ESGF/CMIP6.6261 |
| *piClim-BC* | AerChemMIP | doi:10.22033/ESGF/CMIP6.6225 |
| *piClim-OC* | AerChemMIP | Realisation 1: doi:10.22033/ESGF/CMIP6.9439<br>Realisation 2: doi:10.22033/ESGF/CMIP6.6257 |
| *piClim-aer* | AerChemMIP | doi:10.22033/ESGF/CMIP6.6270 |
| *piClim-NOx* | AerChemMIP | doi:10.22033/ESGF/CMIP6.6246 |
| *piClim-VOC* | AerChemMIP | doi:10.22033/ESGF/CMIP6.6266 |
| *piClim-O3* | AerChemMIP | doi:10.22033/ESGF/CMIP6.6254 |
| *piClim-NTCF/piClim-aerO3* | AerChemMIP/RFMIP | Realisation 1: doi:10.22033/ESGF/CMIP6.8418<br>Realisation 2: doi:10.22033/ESGF/CMIP6.6249 |
| *piClim-LU* | RFMIP | doi:10.22033/ESGF/CMIP6.11104 |
| *piClim-Anthro* | RFMIP | doi:10.22033/ESGF/CMIP6.11090 |

**Author Contributions**

The model simulations were set up, reviewed, and/or ran by FMO'C, NLA, MD, GF, PG, CH, BTJ, BK, JK, JPM, ER, SS, ST, AW, JW, and GZ. Data processing and upload to the Earth System Grid Federation (ESGF) was carried out by MD, BK, RK, MR, SS, JCT, and JW. Analysis was carried out by FMO'C, PG, BTJ, JK, JPM, ER, JS, SS, JCT, AW, and GZ. The manuscript was prepared by FMO'C, PG, BTJ, JK, OM, ER, JS, SS, JCT, AW, and GZ, with additional contributions from all co-authors.


**Completing Interests**

The authors declare that they have no conflict of interest.

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
