# Peer review of "Assessment of pre-industrial to present-day anthropogenic climate forcing in UKESM1"

_Atmospheric Chemistry and Physics, 2019_

## Referee Comment (RC1) · Anonymous Referee #1 · 27 Feb 2020

This paper is an important documentation of effective radiative forcing in a leading climate model, and should be published following modifications.

For reasons that I understand and appreciate, the paper has a rushed and inconsistent feel to it and is poorly written in some places. I have a large number of comments, some of them rather important for clarification of what has been done, and what the results indicate. Hence, I have recommended the paper needs major modifications before I believe it will be suitable for publication.

Major

There is a lack of clarity about some of the experiments which led to significant confu-

sion. Some of it was dissipated by the text later in the paper but, for some, "mystery" persisted. Although I suspect these labels follow the various MIP protocols, they are unhelpful to a reader less immersed in the world of MIPs and clearer information, early in the paper, of what is and what is not included in various experiments is necessary. While Table 3 goes some way to doing this, it leaves various issues unclear; it, or an accompanying table, needs to be much more specific about what is, and what isn't included in different integrations

1. L33 starts the confusion about what NTCF is and whether it includes (following AR5) methane and the shorter-lived halocarbons. So a clear definition early in the paper is needed (or as indicated below, NTCF is not used at all, as it is incorrectly used as a shorthand for aerosol and ozone precursors).

2. L33: Even after reading the paper, I did not really understand what "tropospheric ozone precursors" means and whether the impact that these have on lower strato-spheric ozone (e.g. following what was shown in the Sovde et al. papers) was included or excluded from the analysis. Was ozone change above the tropopause prevented in piClim-O3 runs (and the associated individual runs such as piClim-NOx)? If not, I would class it as quite misleading to call this "tropospheric ozone" and would suggest an alternate name (such as "ozone reactive precursor gases" to make it clearer). If it was prevented, then it needs to be made clear what the criteria were for prevent-ing ozone change in the stratosphere. There is the parallel issue for pi-Clim-HC, and whether it allows tropospheric ozone change resulting from the ODSs. An additional confusion is whether CH4 is classed as an ozone precursor. Again, it took some time before this became clear (or more precisely, less unclear) and Table 3 didn't help in this regard.

3. L160: piClim-HC gave me even more trouble. I actually thought HC stood for "hy-drocarbon" until quite late in the paper. I then found it meant "halocarbon" but it was being used interchangeably with ODS, which was strange until I learnt that non-ODS halocarbons were excluded, for reasons that are not clear to me given the name of the

experiment. And it wasn't until later in the paper that it became clear that the impact of HCs on ozone was included in these experiments, but still whether this influence remains stratospheric is unclear. This needed more detail in Table 3 or a new table making clear what is what.

4. L160: To find that piClim-VOC really meant VOC+CO was also confusing.

5. L160: Added clarity would be given if Table 3 could make clear the distinction between experiments that impose changes in emissions and those that apply changes in concentrations (or in lower boundary conditions). Even after reading the paper, I am not entirely clear on some of these (e.g. methane). For this table, HC should be spelt out (or replaced by ODS) and it should be explicit what gases are included in the Trop O3 precursors. The authors should also reconsider whether "aerO3" is a better descriptor than NTCF, or removed entirely.

6. An additional and quite prevalent issue is that many of the important references are to papers that are submitted (but unavailable) and in some cases only "in preparation" – some key results was in these papers and I felt a bit teased by the allusion to these, without being given either sufficient information or the underlying punchline. I am particularly concerned about the overlap between this paper and Andrews et al. (2019 - submitted) which, from the title of that paper includes forcings in UKESM1, and the Morgenstern et al (2019 - in preparation) paper that contains a discussion of a key result (HC-driven net forcing is strongly negative) when no details at all are presented here, beyond a cursory (4 line) section. This also leads to a very mixed feeling to the paper. Tropospheric ozone is handled in depth, while stratospheric ozone is dealt with in a cursory way, with a reference to the in preparation paper.

7. I am concerned that it is hard to understand what features in the geographical plots are signal and which result from unforced variability. This needs to be discussed in more detail, especially for forcings such as N2O that are inherently small. This is one clear disadvantage of the ERF and partly the reason for my comment at 104/127,

below, about ERF being the metric of choice. I do not believe it is the metric of choice in all circumstances, especially for small forcings.

Other (but note that these are not exhaustive)

25: "altering the sign" – it would help readers to know what the sign has been altered from and to.

49: I would say the change of concentrations resulting from changing emissions is the first part of the cause-effect change

65-94: The intro, with its focus on LLGHGs and tropospheric ozone, feels unbalanced. Why are other forcings not discussed here (or is this detail inappropriate here)? The split between tropospheric and stratospheric ozone forcing is to some extent artificial and a little dated (see e.g. Sovde et al.), because the mapping of the drivers of ozone change does not map exclusively on to ozone change in the troposphere and stratosphere, and that could be clearly acknowledged

104 and 127: "preferred" and "metric of choice" – by who?

117: Xia et al – although it would be tiresome to say "found in their model" on all occasions in this paper, I think it is important in cases, as it is a result of a single model study

119-121: This needs clarification. For RF it is a requirement that tropopause and TOA forcings are identical (not "nearly identical") following SARF, so in a sense the TOA/trop distinction doesn't matter. However, where it does matter is that RF requires the specification of the tropopause, which is always to some extent arbitrary. whereas ERF does not – the model just does its own thing. This is one advantage of ERF.

154: "addressed the strong negative" – the nuance here (and in other places in the paper) is that it was too strong negative (i.e. unrealistic). Is that the intention?

268: "from energy budget constraints" – I have no idea what this means and hence

whether the Andrews and Forster (2019 - submitted) paper is in some ways a superior assessment to that given here. Should we (or Andrews and Forster) be concerned that the UKESM forcing falls outside their stated range? This is another problem with the heavy reliance on unpublished papers.

282: "some evidence" – I did not know what this meant. It either is, or it isn't? Or is this meant to imply that the derived forcing is no larger than the unforced variability in this region, and hence is not a robust signal? There is no indication of statistical significance in any of the plots, or discussion of it and this should be rectified in some way. See Major Comment 7

282: This paragraph is the first place where it is made clear that the HC calculations include the ozone forcing from ODSs (and indeed that HC=ODS) and leaves the reader unclear whether this is the case for (e.g.) methane and CO2.

287: The GHG forcing is rather inhomogeneous. Is this statement that the aerosol forcing is more so based on an objective measure of inhomogeneity or a subjective assessment?

288-290: Is this conjecture or the result of model analysis?

292: Here it is unclear whether ozone change from methane is in the GHG or the trop O3 plot

296-297: Is the land use forcing over the ocean real (resulting from downstream rapid adjustments) or just indicating unforced variability in the model?

298: This negative forcing over the mid-lat continents is a striking result. Is this the first paper to indicate it? Also, presumably the Land Use forcing in these areas is mostly "historical". This paper understandably does not address the time dimension of the evolution of the forcing, but perhaps some added text could be added to indicate when this forcing would have been most active.

303: I wonder if adding an additional frame with the zonal-mean forcings of each com-

ponent would be useful for bringing out the overall larger-scale structures of the forcing.

303: It is striking that the geographical plots in each figure differ, presumably because different coauthors were responsible for them, with different conventions in the headers, and in the case of Figure 8, a different projection. Could these be made more consistent? Personally, I find the addition of global mean values on each frame, as in Figure 4, quite helpful.

323: Here and elsewhere it should be made clear if the concentrations are surface values or mass-weighted means over the whole atmosphere (and whether a single global mean values is used everywhere in the model for each LLGHGs).

328: 121 ppm. Presumably the expressions in Etminan et al. could be used to derive their forcing for the same $CO_2$ change?

331: Some of the $CO_2$ SW forcing likely comes from the SW bands of $CO_2$, but this is implicitly discounted here. Or are these bands not included in the radiation code?

352: These two estimates overlap within the uncertainties.

358: "noise" – here and throughout, it is important to guide the reader as to which features are noise and which are robust. Could this be done via masking or stippling? For example, is the marked negative feature in clear-sky longwave in Figure 4c over Russia and its surroundings, robust?

363-365 and Figure 5: 364 talks about "increased ozone" (in the UTLS) but no hint is given as to what drives this feature. Is it a "self-healing" effect of the overlying depletion? The caption to Fig 5 needs to make clear this is for the N2O experiment.

396-398: There is no hint of what is going on here, except a reference to a paper that is "in preparation". This is very frustrating and leads me to question whether the HC/ODS results should be shown here at all, if there is no accompanying analysis of the striking result. It became a bit unclear to me why, for example, the tropospheric ozone is discussed in such detail in this paper, but not stratospheric ozone. Also note

that the assumption that HC=ODS is made here without elaboration, although later there is some hint at this.

414-415: Again, this allusion to a different submitted paper, with no hint of what is found, is frustrating

423: The frames in Figure 6 are in a different order to those in Figure 4, and there are a different number. Commonality of presentation would help the reader. Again, the caption to this Figure does not make it clear it is for methane. (All captions should be checked for this – e.g. Figure 12 suffers the same lack of specificity.)

440: This is the first place the reader learns that the non-ODS halocarbons are not included in HC.

444: "warrant further investigation" – I agree. this difference is interesting and one factor not considered (spectral overlap) would likely push the difference in the wrong direction (i.e. make GHG even lower than the sum of individual components)

473: There is a minor ambiguity in using "sum", "all" and "total". As I understand the Total=All? This also applies to the text from 513-521.

484: "many other models" – but aren't versions of HadGEM included in the Stjern et al. 2017 paper?

494-496: This sentence seems out of place, especially without any supporting reference.

550: I had a lot of comments on Section 4.4 as it was particularly unclear in my view. One underlying issue, raised above, is whether the tropospheric precursor gases are allowed to change lower stratospheric ozone, and whether that change is included here. Without a clear statement, it is very hard to understand this section and place it in the context of prior work. Similarly, the lack of clarity on whether CH4 is included or not, as an emitted species, is unhelpful.

559: This figure purports to show "tropospheric column" but the panels are labelled in ppbv. Is this the mean (mass weighted) ppbv over the depth of the tropophere? How is the troposphere defined for this purpose? Would Dobson Units be a better measure? Figure 5a included a latitude-height plot of ozone change due to N2O, and a similar plot would be helpful here.

565-566: I wasn't sure why the "South Pacific" was highlighted and here and when "Western Pacific" is mentioned, I presume in both cases it means tropical Pacific?

588: By applying this mask, how much forcing (due to the precursors' impact on stratospheric ozone) is excluded from the analysis?

594: I found this paragraph hard to follow, and in particular whether differences in the literature were due to radiation code differences (or the effect of applying a kernel) or process level differences in determining the ozone change. I ended up confused as to whether the new result was higher or lower than Stevenson et al. and for what (dominant) reason it was so.

607: "Section 3.2.3" – I presume this means Section 4.2.3 but, as noted above, there is essentially no detail in 4.2.3, and certainly nothing on the tropospheric ozone change in piClim-HC. This is frustrating when trying to make sense of this section.

616: It would be useful to have the global-mean values, perhaps in the same way that they are shown on Figure 4 in each frame. Also, some comment about how different Fig 10d and Fig 3c are (i.e. RF versus ERF) would seem useful.

666: Are the methane concentrations, the change (i.e. NOx – control, as indicated in Column 1) or the absolute values for NOx. It might be better to show the change?

680: Make clear that this is the OH at 1 km.

686: These do not appear large (e.g. compared to NOx)

695: "eastern Pacific" – does this mean the eastern tropical Pacific, as I see changes

of both signs in the E Pacific?

699: AOD is meaningless without specifying a wavelength. I'm curious why the VOC/CO run has an increase in AOD but a decrease in CDNC (e.g. over the W Pacific). Is this because one compares column integrated amounts and the other is at 1 km?

712: "negligible" – is this because of cancellation of regions of positive and negative, but the local values can be large?

726: Are the NOx sources just from the surface, or are aviation emissions also included? I know it is beyond the scope of this work, but it is of interest whether this net negative applies to all sources or likely just surface sources.

739: Again, a reference to an "in preparation" paper is unhelpful here.

750: I think this would be better labelled as Aerosol/Ozone, rather than NTCF since it clearly isn't all NTCFs (which would avoid the "excluding methane" repetition). Since both aerosol and ozone are treated separately anyway, I am not quite sure of the need for this section, especially as they, together, only form a subset of NTCFs.

769-771: These lines seem to contradict each other about whether it is cloud fraction or cloud optical depth, or maybe I misunderstand the point being made.

776: Units are missing for the temperature and cloud parameters. Are temperature changes really in K?

810-811 "too strong" and "bias still exists" – I haven't read the Robertson paper, but could it be said what the too strong albedo response is relative to (other models? observational constraints?), and whether this indicates that UKESM is definitely incorrect (as this text implies) rather than being a plausible outlier?

Typos etc (but note that these are not exhaustive)

25: If "forcing" means "ERF" it should say this.

83-89: Long sentence

64 and elsewhere: "it's … doesn't … weren't" – it doesn't worry me much, but such contractions are not usually used in formal scientific writing

216: machine's

236: These equations use the subscripts cs and clear – do they mean the same thing?

552: refer to Figure 3 here?

592: "Garcia"?

608: I think it important to also refer to the Sovde corrigendum https://doi.org/10.5194/acp-12-7725-2012. .

---

## Referee Comment (RC2) · Anonymous Referee #3 · 15 Apr 2020

To the editor,

I think this work serves two main purposes, i.e., (1) documenting (and disentangling) the ERF from short-lived emissions, long-lived GHGs, land-use, and ODSs in UKESM1, and (2) indicating the relevance of ESM-interactions for estimating the anthropogenic forcing.

I think the work is solid, the subject is well explained, comparisons with earlier studies are included, and the information shared in figures and tables is appropriate.

Presenting estimates of ERF is important to understand the evolution of climate (modeled or observed). Models participating in CMIP6 were motivated to perform specific

experiments in AerChemMIP and RFMIP, which would allow a better characterization of the forcings felt by ESMs, e.g., over the historical period. It is important that those ERF estimates are well-documented, as they facilitate the interpretation of the fully-coupled behaviour of models (and climate). I think this study achieves presenting ERF estimates for the UKESM1 model in a nice, coherent, and attractive way. In general, the study is well written and attractive to read.

In addition, as UKESM1 contains a relatively extensive description of atmospheric chemistry, aerosols, and aerosol-cloud processes, the interactions between different species and processes are included. This makes this model an interesting tool to estimate the ERF of various short-lived emissions or longer-lived GHGs.

I think this work is valuable and of sufficient quality to be published in ACP. However, I think the manuscript could be considerably improved. I have listed several points on which I think the manuscript should be changed.

I have grouped my comments in three categories, i.e., main remarks, technical remarks, and additional detailed remarks.

Main remarks

1. TABLE 3

Table 3 is very interesting and plays to my opinion a rather central role in this study. A lot of the values in the table are being referred to in the text. However, some parts of Table 3 are not discussed. This is the case for two large and two small groups of experiments : SO2-BC-OC-Aer, NOx-VOC-O3, LU and anthro. For these groups, only the NET ERF is discussed, but not the individual contributions. I assume this is related to the fact that for those 4 groups another split than the one in Eqs. (6)-(8) is followed, which is probably assumed to be more appropriate.

To make the study more consistent, I would at least mention that the non-discussed values exist in the Table 3, and are given for comparison/completeness/consistency.

Explicitly presenting the numbers from the other approach followed for some experiments in a table would be interesting. Now these numbers are only mentioned in the text.

**2. EXPLANATION OF RADIATIVE FORCING**

Although radiative forcing and effectve radiative forcings are very useful concepts, it is not always so easy to explain them. I think the authors have done a nice effort in trying to explain it. However, an extra effort should be done to make it more precise, fully coherent, and more illustrative. The final link why it is a useful concept should be elaborated more. I list here below also some specific places where I think the description should be improved.

- page 2, line 61-63 : Including ... . As a result ... : I do not think that "As a result" is appropriate, as I do not think there is a causal relationship.

- page 2, line 63-64 : RF is a framework : this is a rather vague explanation.

- page 4, line 115-117 : I think that the sentence on line 115-117 illustrates that ERF can be very different from IRF, however, it does not illustrate that it is better (as suggested in the former sentence). It might be an option to reverse the order of these sentences.

- page 4, line 113-115 : although it is correct what is written here, I think it should be explained better.

**3. FEEDBACKS, INTERACTIONS, NON-LINEARITY, INDIRECT EFFECTS**

I think that feedbacks, interactions and indirect effects are an important part of this paper. In the abstract, it is stated that "... by quantifying ..., it enables the role of various climate-chemistry-aerosol-cloud feedbacks to be quantified." However, despite this sentence in the abstract, not all feedbacks have been quantified (it would be a very large task to quantify them all). The fact that they are active in UKESM1, and impact therefore ERF estimates, is already an important step. I think the text will benefit from

more precisely describing what is quantified, and what one is not able to quantify.

I do not ask the authors to do additional analysis, or perform more comparisons with a model which contains less interactions (e.g., HadGEM-GC3.1). For quite some of the forcings, the authors did a very nice job in unraveling several contributions. However, it would be nice if the authors could be more careful and precise in some of the general expressions about feedbacks and interactions.

I have in addition the following comments.

1) I think the authors should explain what according to them is the difference between a feedback and an interaction.

2) It is a pitty that the studies of Morgenstern et al. [2019, in preparation] on ozone and O'Connor et al. [2019, submitted] on methane have not been published yet.

3) Some of the formulations, referring to feedbacks and interactions, are often rather vague (less firm than the abstract suggests). Examples of these are :

- page 15, line 354 : ... "likely" ... the effect of adjustments ... including O3 depletion and fast cloud adjustments

- page 15, line 359 : which "might" be due to an aerosol effect

4) the abstract stresses well the non-additivity (or non-linearity) of several ERF estimates. However, the study sometimes remains vague in reasons for it, and just uses expressions like "the non-linearity between the GHG", or "the internal mixing of aerosols". Other examples of not so clear usage of the concepts :

- page 2, line 36 : due to the inclusion of non-linear feedbacks and ES interactions : vague

- page 2, line 38 : By including feedbacks between GHGs (isn't it interactions?)

- page 2, line 38-39 : some of which act non-linearly

[Figure]

**4. UNCERTAINTIES**

The ERF estimates in the manuscript are accompanied by an uncertainty range in large parts of the main text and in Table 3, but not in the abstract, the conclusions, or some parts of the main text. This should be made consistent.

Also, one should maybe say something about the real estimate of the uncertainty. There is something on aerosol ERF uncertainty on page 20, line 453-455, and maybe this can be expanded. In general, UKESM1 shows uncertainties of around 0.02 to 0.04 W/m2 on global 30-year mean averages of ERF – it is the uncertainty on the ERF in the model. However, if one wants to see these numbers as best guesses for the the ERF as experienced by the Earth, then the uncertainty is probably much larger. Emission uncertainty, lack of process understanding, too low spatial resolution, biases in the mean state of UKESM1 and others factors might contribute to the uncertainty. It would be interesting if the authors could also shed some light on this.

**5. VALIDATION OF THE MODEL**

There is very little information on the validation of the model, except via references to Mulcahy et al. [2018, 2019] and Archibald et al. [2019]. There are a few paragraphs (e.g., page 20, line 454-457 : aerosol comparison; page 21, line 491-492 : AOD comparison), mentioning some comparisons with observations. We do not ask for a detailed comparison, but some qualitative main findings on the performance of the model might be mentioned, both related to distributions of forcing agents (aerosols, ozone, ...) as to the general behaviour of the model (mean model state).

Technical remarks

**1. POSITIVE NUMBERS**

Some positive numbers have a "+", some not. This should be made consistent.

**2. REFERENCES TO THE PHYSICAL MODEL**

It might be relevant to explain or highlight the relevant differences between UKESM1 and HadGEM3-GC3.1 better (both models are very close and have both participated in CMIP6 I assume). In the text quite often comparisons with HadGEM3-GC3.1 are made, and it would illustrate the role of having more/less interactions in a coupled model. I would think that possible differences are related to fixed ozone, fixed oxidants for secondary aerosol formation, fixed methane profile, fixed CO2 profile, ...

3. REFERENCE TO FORSTER ET AL. [2016] FOR STANDARD ERROR

The way to calculate the standard error is mentioned three times. It should be mentioned when the uncertainty is met for the first time (both in the text and in the tables). The standard error and its reference are now mentioned in :

- page 12, line 272 (Table 3).

- page 18, line 402 (related to CH4) : where the 0.04 W m-2 is the standard error following Forster et al. (2016) : should be mentioned once (the first time).

- page 31, line 760-761.

4. MULTI-ANNUAL MEAN / ANNUAL MEAN / MEAN

It is mentioned in the beginning of the manuscript that almost all values and maps will be 30-year averages. Some sentences and figure captions in the text treating those values or maps once more mention explicitly that the averages are "multi-annual", whereas other figures captions or sentences do not. Please describe it in a consistent ways. An option might be to describe it clearly at some point in the, and say that it is valid for all the remaining text. Examples of differences in the description :

- page 18, line 395 : multi-annual mean (while page 14, line 323 : global mean)

- page 24, line 562 : The multi-annual mean ...

- page 25, line 599 : on a global annual mean basis

[Figure]

- page 29, line 699 : Global distributions of the multi-annual distributions

- page 33, line 795 : multi-annual global mean

5. WRITING OF LAND USE

Both "land-use" and "land use" are used in the text.

6. CONSISTENCY

- page 10, line 261-265 : why is a difference of 0.2 W/m2 for GHG called "consistent", but for the aerosol mentioned "less then the estimate from HadGEM3-GC3.1"?

- page 22, line 502 : -0.12 W m-2 (-0.4 to +0.1), whereas on page 10, line 267 : 2.3 (+1.7 to +3.0) [5-95%] W m-2 : the position of Wm-2 is different.

- page 22, line 515-519 : only here "(i)" and "(ii)" are used. I would not use it just for this single occasion

- page 23, line 536 : only here "&" is used. I would not use it just for this single occasion.

7. NAMING of SWcs, Lwcs, SWcre and LWcre COMPONENT OF ERF

There are different abbreviations used to express the same physical quantity. One should make these coherent. An example of this is the LW clear-sky component of ERF, which appears in different ways in the text, tables, and equations :

- page 11, Table 3 : LWcs

- page 10, Eq. 8 : ERFcs

- in the text : a CS LW component

- page 16, Fig. 4 : (a) clear sky SW, (c) clear sky LW, and (f) net (SW+LW) CRE : it sounds as something is missing after SW and LW

- page 31, line 762-770 : NETcs, LWcs, SWcre

- page 32, Fig. 12, caption : SWcs, LWcs, SWcre, and LWcre

**8. STATISTICAL SIGNIFICANCE OF DIFFERENCE BETWEEN IDENTICAL SIMULATIONS**

Several experiments have been performed on different machines. Related to this I have the following remarks :

- Table 4 : I assume there are three piClim-SO2 experiments, so one could show the difference R1-R2 but also R1-R3.

- Table 4 : Some differences are a bit large compared to their uncertainty for SO2 : NET ERF, SW CRE, NET CRE. Is there an explanation for this?

- Table 4 : Why is everything 0 for the NTCF experiment?

**9. NOx EMISSIONS AND SO2**

Figure 11 : is the impact of NOx emissions on AOD not mainly in regions with SO2 emissions from volcanoes? Indonesia, west coast of South America, Etna region, existing ship-lanes in 1850 over North Atlantic? Might probably NOx have then also a large effect if it is co-emitted with anthropogenic SO2?

**10. FORMULAS**

The "." or "," at the end of formulas can be closer to the actual equations (currently there is a 1.5 cm large gap). There should not always be a ":" at the end of the text just before a formula. This depends on the context. One can also have a ",", or nothing.

**11. REFERENCES TO SECTIONS**

The references to specific sections are not always correct. Please check and correct them. A list of incorrect section references :

- page 8, line 192 : Sect 3.1 -> 4.1

- page 12, line 287 : Sect. 3.2 -> Sect 4.2

- page 13, line 291 : Sect. 3.3 -> Sect 4.3

- page 13, line 296 : Sect. 3.4 -> Sect4.4

- page 14 ,line 307 : Sect. 2 -> Sect 3.

- page 25, line 602 : Sect. 3.2.3 -> Sect 4.2.3

- page 29, line 705 : Sect. 3.4.2 -> Sect 4.4.2

12. USE OF e.g.

There should be a "," before and after "e.g.".

13. REFERENCE TO EQUATIONS

The way one refers to equations is not homogeneous in the text (sometimes capital letters, small letters, abbreviations, or no abbreviations). Please make it coherent. A list of the differences found in the text :

- page 4, line 111 : Eqn. 1

- page 10, line 251-252 : in equation (5)

- page 15, line 348 : following the equations (6)-(8) described above

- page 19, line 425 : using Equations (6)-(8)

- page 27, line 643 : From Eqns. (10) and (11)

14. USE OF THE WORD "FORCINGS"

I have the impression that the word "forcing" (outside the context of radiative forcing and effective radiative forcing) is possibly not used in a completely coherent way. I have listed below a few locations where it is used, and it seems not to always have the same meaning. I think the text should be more careful and precise in how and where it is used.

- page 1, line 16 : climate forcings

- page 1, line 31 : a positive forcing due to ozone

- page 3, line 65 : various mechanisms, both anthropogenic and natural. Here I would have used the word forcing and not "mechanism".

- page 3, line 65 : ... use of RF ... it is often used inconsistently ... : is RF meant in the first part of the sentence, because that is well-defined (page 2, line 54-56)?

- page 3, line 67-69 : This sentence mentions three things, and it is not clear whether the reader should see a causal relationship between "the inconsistent calculation of forcing between drivers", and "the large differences in forcing between CMIP5 models". For the last part of the sentence, it is not clear whether the authors mean that models are very different or that the methods of estimating forcing can be very different.

- page 3, line 82 : the resulting RF : is not the same as meant by the definition on page 2 line 54-56.

- page 7, line 171 : in which SSTs and SI and all forcings

- page 33, line 782 : between the aerosol and O3 forcings

- page 35, line 840-845 : "forcing" used four times. Also "anthropogenic" forcings and "natural" forcings, whereas it is not so clear what to understand here under natural forcing. In the text there is a reference "As summarized above", but I think there was not a large focus on "natural forcings".

15. THE USE OF ABBREVIATIONS

The definition of an abbreviation is often given several times, whereas that should be limited to :

- defined once in the abstract,

- defined once in the conclusions,

- defined once in the figure caption, and

- defined once in the main text (but not more).

Once a definition is given in the main text, it should not be defined again.

16. PD AND PI ARE ADJECTIVES

- page 1, line 21 : at the PD, or relative to the pre-industrial (PI) : PD and PI are both adjectives, and not substantives. So they cannot be used on their own. E.g., on page 1, line 15 it is used correctly as an adjective : "... a wide range of present-day (PD) anthropogenic climate forcings ...".

17. PD-PI DIFFERENCE

The way the ERF is described as a difference between TOA fluxes in PD and PI times, is not coherent in the text. Sometimes one finds a "PD forcing" type of expression, sometimes a "PD to PI forcing" type of expression, and sometimes neither PD nor PI are mentioned. Some examples of the varying usage are given below. It would be nice if the description could be more coherent. An option would be to write in the text that all values shown in text from that point onwards are always PD-PI differences, and then PD and PI should not be repeated every time.

- page 3, line 71 : between the pre-industrial (PI) and the present day (PD)

- page 3, line 76 : at the PD relative to the PI

- page 5, line 130 : the PI to PD effective radiative forcings (ERFs)

- page 5, line 132 and 133 : PD forcings will be quantified relative to the PI [in addition is "the" strange before PI which is just an adjective]

- page 5, line 139 : PD anthropogenic forcings relative to PI

- page 6, line 157 : To calculate the pre-industrial to present-day effective radiative forcings ... due to a PI-to-PD perturbation

- page 10, line 260 : the global mean PD ERF [here without any reference to PI]
- page 12, line 271 : PD effective radiative forcings (ERFs) of climate relative to PI
- page 12, Fig. 2 : y axis PI-to-PD ERF
- page 12, line 280-281 : PD ERFs from changes in ... since PI
- page 12, line 281 : PD ERF from GHG [but no reference to PI]
- page 13, line 303-304 : effective radiative forcing (ERF) relative to the pre-industrial
- page 14, line 319 : cloud radiative effect (CRE) at the present day [no reference to PI]
- page 15, line 338 : this forcing is small for the PD [no reference to PI]
- page 15, line 347 : due to changes in N2O from PI to PD
- page 16, line 370 : for the present relative to the pre-industrial
- page 18, line 400 : resulting in a PD ERF [no reference to PI]
- page 19, line 423 : of the PD methane ERF relative to PI
- page 19, line 430 : This PD GHG ERF
- page 20, line 462-463 : of PI to PD aerosol ERF
- page 21, line 472 (Fig. 7) : Aerosol ERF at TOA ... [no PD or PI mentioned]
- page 22, line 501 : The PD OC ERF relative to PI
- page 23, line 551 : The global mean ERF from ... [no PD or PI mentioned]
- page 25, line 591 : the PI-to-PD change in tropospheric O3
- page 27, line 651-652 : the tropospheric O3 RF between PI and PD
- page 28, line 679 : and the PD aerosol ERF
- page 30, line 723 : ERF from PI-to-PD changes

- page 31, line 744 : the PD ERF from piClim-O3

- page 33, line 795 : of changes (PD-PI) in

18. TIMESLICE EXPERIMENTS

Maybe it would be nice to better define what a timeslice experiment is. I would possibly describe it as an experiment with fixed boundary conditions (possibly having seasonal cycles), which one runs for several years to reduce the noise-to-signal ratio (the noise is caused by inter-annual variability). The longer one runs, the better estimate for the mean one can obtain. The locations were it is used are :

- page 6, line 158 : maybe explain timeslice here (forcings are kept constant)

- page 7, line 171 : time slice

- page 15, line 604 : timeslices : in ACCMIP they were different from here ...

19. FIGURE 4

Fig. 4, panel (a) : Why is there only some spatial variability north of 30N?

20. CONSISTENCY OF THE FIGURES

The figures and their captions should be more coherent throughout the manuscript. Below I list some places where improvements should be made.

- page 16, Fig. 4 : at the top of the atmosphere (TOA) : this is however not mentioned in Fig. 3.

- page 17, Fig. 5 : why mentioning "annual mean" or "multi-annual mean", whereas it is not mentioned in Fig. 4.

- page 19, Fig. 6 : it would be nice to have the global mean values given in the figure heading.

- page 19, Fig. 6 : mentions "according to Ghan (2013)". Why not in Fig. 4?

- units in figures : Fig 7 uses "(W/m2)", whereas other figures use "/ W m-2".

- page 24, Fig. 9 : of the "multi-annual" distributions, whereas other figures do not mention "mult-annual".

- page 26, Fig. 10 : it would be nice to add the global mean RF.

- page 32, Fig. 12 : it would be nice to add the global mean values

Additional detailed remarks

ABSTRACT :

- page 1, line 23 and 28 : "larger than the sum of the individual GHG ERFs" and "less than the sum of the individual speciated aerosol ERFs" : I don't know whether these aspects should be mentioned in the abstract.

- page 1, line 15 : "In this paper" : I would not mention "In this paper" in the abstract.

- page 1, line 18-19 : by quantifying ..., it enables ... : this sentence seems not completely coherent. There is also twice "quantify" in the same sentence (in addition there was already "quantify" on line 15).

- page 1, line 21-22 : I would put the numbers at the end of the sentence, and "carbon dioxide, nitrous oxide, ..." at the beginning of the sentence. Now, one first reads numbers, but one does not know what their meaning is.

- page 1, line 19 : by this sentence, one suggests that climate feedbacks can be quantified by fixed-SST simulations. However, some of them are strongly suppressed in fixed-SST simulations.

- page 1, line 25-26 : is the "BC absorption" not part of the "instantaneous forcing from aerosol-radiation interactions"?

- page 1, line 27 : mean -> "imply" or "cause".

INTRODUCTION

- page 2, line 42-45 : necessary ... detailed ... all aspects : this is probably exaggerated. I suggest to formulate it differently.

- page 2, line 42 : attribute it and its impacts : it (refers to climate change) and its impacts. It is not clear whether the authors mean something different with "climate change" and "impacts".

- page 2, line 44 : climate response and its impacts : (same comment as above).

- page 2, line 44 : a key mechanism : I would not call CMIP6 a "mechanism". Also "key" is possibly exaggerated - other initiatives (if CMIP would not have existed) might have also had good outcome.

- page 2, line 45-46 : which designs and distributes data : "designs data" sounds strange. Possibly one could say that "experiments are designed".

- page 2, line 47 : "these important climate science questions" : it is not clear which questions one refers to.

- page 2, line 50 : quantifying changes to the Earth's radiation budget, often termed radiative forcing : maybe radiative forcing needs a bit more explanation.

- page 2, line 64 : It's been -> It has been.

- page 3, line 65 : the strength of the various mechanisms : of various mechanisms

- page 3, line 65 : mechanisms, both anthropogenic and natural. I do not think that "mechanism" is the most appropriate word to be used here, especially in the context of "anthropogenic" and "natural".

- page 3, line 70-71 : is typically based on .. and using -> uses.

- page 3, line 72 : I suggest to put "e.g." before "based on Myhre et al. (1998) and Ramaswamy et al. (2001)", as there might exist other expressions.

- page 3, line 75 : Skeie et al. [2011] : I think that study is not so much about

**ACPD**

observational-based estimates of forcing. Is this paper very relevant in the discussion of forcing strength of GHGs?

- page 3, line 81 : including -> therefore allowing.

- page 3, line 82 : meaning -> implying.

- page 3, line 82 : doesn't -> does not.

- page 3, line 84-85 : of a robust ... constraint -> of robust ... constraints.

- page 3, line 84 and 89 : twice "additional uncertainties".

- page 3, line 85-86 : across multi-model ensemble : is that really what the authors want to stress? Might "across models" be sufficient?

- page 3, line 87 : chemistry models : does one mean CTMs or CCMs?

- page 3, line 91-93 : three times the word "uncertainty" in one sentence. Maybe it can be reduced to two.

- page 3, line 91-93 : it looks like aerosols get only very limited text attributed.

- page 4, line 96 : is "schematic" the correct wording?

- page 4, line 101 : Although -> Because/As.

- page 4, line 102 : maybe "also" can be skipped. I don't know if it really reflects well the meaning of the sentence.

- page 4, line 109 : andAi -> and Ai (blanco needed).

- page 4, line 110 : or over the land : vague.

- page 4, line 112-113 : but global mean surface temperatures or global ocean conditions remain unchanged. However, in reality with fixed-SST simulations, land surface temperature (and thus global mean surface temperatures) can still change a bit.

- page 4, line 113 : error -> uncertainty (I assume the authors mean "uncertainty").

- page 5, line 129 : I would skip "including" because the list mentioned seems rather complete.

- page 5, line 130 : forcings from anthropogenic drivers -> forcing from anthropogenic drivers.

- page 5, line 131 : with a fully coupled : the "with" gives the impression that one uses the model here in its "fully-coupled" configuration. But here it is not used in its fully-coupled configuration.

SECTION 2

- page 5, line 142 : "is" the atmospheric and land components -> consists of.

- page 5, line 150 : "is determined by prescribed oxidant fields" : maybe describe differently, as oxidants are not the only determining factor.

SECTION 3

- page 7, table 1 : piClim-VOC : add that also CO is perturbed.

- page 7, line 166-167 : twice ERF : I think mentioning "fixed-SST" is enough to describe the experiments. Obtaining the ERF is the result of such an experiment.

- page 7, line 171 : Effectively, this involves ... : it is a bit a strange way to mention that also a reference simulation is needed.

- page 7, line 171 : SSTs and SI and all forcings : one should not have two "and"s in a row.

- page 7, line 171 : the abbreviations SST and SI have not been defined.

- page 7, line 173-176 : One uses two different expressions, i.e., "monthly time-varying climatologies derived from 30 years of output" and "30-year monthly mean climatologies", to describe the same thing.

- page 7, line 177-184 : is it not in agreement with RFMIP and AerChemMIP?

- page 8, line 194 : emissions and/or GHG concentrations : I think this can just be "and".

- page 8, line 199 : fixed SST ERF experiments : I think fixed-SST is enough to describe the experiments.

- page 8, line 200 : ammonium nitrate : but other forms of nitrate are probably also not present (e.g., nitrate on dust and seasalt).

- page 8, line 208 : fixed SST timeslice ERF experiments : I think fixed-SST is enough to characterize the experiments.

- page 8, line 210 : This makes ... to changes in platform that cannot guarantee bit-reproducible results : this can probably be expressed more precisely.

- page 8, line 211-213 : was scientifically consistent with each other : one should be more clear in what is meant by "scientifically consistent".

- page 8, line 220 : Further to this -> in addition to this.

SECTION 4

- page 9, line 234 : Cloud-Radiative Effect (CRE): in the rest of the text, no capital letters are used when defining an abbreviation.

- page 10, line 240 : either ... and/or : I do not think that it is common to combine "either" with "and/or".

- page 10, line 241 : what is meant by "dynamical feedbacks" : changing meteorology which changes lifetime and thus burden of aerosols?

- page 10, line 251-252 : and any non-aerosol changes in CS flux : is, e.g., the deposition on snow of BC included in this term?

- page 10, line 252 : "The effective radiative forcing (ERF), clear-sky CS), and cloud

[Figure]

Interactive
comment

radiative (CRE) contributions" : as I assume that "contributions" also relates to "clear-sky", I would write : "The effective radiative forcing (ERF), and its clear-sky CS) and cloud radiative (CRE) contributions".

- page 10, line 257-258 : following equations (6) to (8) : maybe only (8)? (As that is the final split which is presented in Table 3).

- page 10, line 262 : HadGEM3 GC3.1 -> HadGEM3-GC3.1

- page 12, line 271 : effective radiative forcings (ERFs) of climate : I do not think "of climate" is needed here.

- page 12, line 273 : use Realisations 2. -> use realisation 2.

- page 12, Fig. 2, caption : "diagnosed from paired fixed SST timeslice simulations with an atmosphere-only configuration of UKESM1". It is not clear why the fact that paired simulations are needed to estimate ERFs is mentioned here. It is, e.g., not mentioned in Table 3 (although also paired simulations are the bases for the results of Table 3).

- page 12, line 283-284 : ERF from the piClim-HC perturbation experiment ... positive forcing from the other LLGHGs : in the first part of the sentence one talks about the ERF from experiments, and in the second part about the ERF of physical things (in this case LLGHGs). The sentence should be improved.

- page 12, line 287 : The aerosol forcing is ... due to their ... : "aerosol" is singular, but "their" refers to something plural. So it sounds a bit strange.

- page 13, line 294 : "weakly positive in comparison with other forcings" : this sounds a bit strange.

- page 13, line 298 : their combined : sounds strange. I would suggest "the combined".

- page 13, Fig. 3 : maybe add the global mean values in the figure headings.

- page 14, line 308-309 : Despite ..., ... produce slightly different results. Isn't it what

one should expect? As it is expected, I would not use "despite".

- page 14, line 324 : 1.83 Wm-2 (but 1.82 Wm-2 in Table 3).

- page 14, line 325-328 : It is informative to stress the absolute difference in CO2 ppm. However, maybe one could add that the relative change in CO2 concentration is more relevant for the forcing.

- page 15, line 337 : low-clouds -> low clouds.

- page 15, line 338 : piClim-CO2phys : the "2" should not be an index.

- page 15, line 340-341 : low-level cloud -> low-level clouds.

- page 15, line 337-341 : The first sentence gives the impression that the balance comes from two terms. However, the second sentence adds that the balance (or closure) comes from other terms.

- page 15, line 342 : found a much larger effect : this looks like a dramatic message, but it is just because the forcing is stronger (it is a 4xCO2 experiment). Therefore the reader is a bit in doubt whether he captures what the authors want to say.

- page 15, line 354 : weren't -> were not.

- page 16, line 366 : correlated to -> correlated with.

- page 16, line 378 : SAM : this abbreviation has not been defined.

- page 17, line 380-381 : "and a reduction of associated ... " : this sentence is slightly confusing, as through the reduction in high clouds, the outgoing LW radiation can be stronger again.

- page 17, line 386 : near surface wind : maybe one can specify the altitude. Is it at 10 m?

- page 18, line 408 : are of the order -> are in the order.

- page 18, line 417 : The major driver ... is greenhouse gases (GHGs) -> are [although I am not sure].

- page 18, line 417 : which is offset by aerosol : This is a slightly unlogical construction : I would think that forcings can be offset, but not GHGs.

- page 18, line 418 : key metrics -> key values.

- page 19, line 424 : "in e)" -> I would advance that slightly.

- page 19, line 433-437 : is the value 2.82 Wm-2 representing the 1850-2011 estimate? (does it already include the correction for going from 1750 to 1850?)

- page 19, line 436 : e.g. CH4 : I assume this is also valid for CO2. Why not mentioning CO2?

- page 19, line 440 : However, there is a discrepancy in ERF of 0.35 W m-2 ... which cannot ... -> However, the discrepancy of 0.35 Wm-2 ... cannot ...

- page 20, line 443 : is this non-linearity similar to (or larger/smaller than) the one which one sees in the RF formulas of AR3 for N2O and CH4?

- page 20, line 450 : The rapid adjustments (RA) ... includes -> The rapid adjustments (RAs) ... include.

- page 20, line 453-454 : are there no other sources of uncertainty : the lifetime of aerosols? Their vertical profile?

- page 20, line 454-455 : twice "sources" in this sentence.

- page 21, line480 : AEROCOM II -> AEROCOM Phase II.

- page 21, line 481 : The BC ERF was +0.32 W m-2 -> The BC ERF is +0.32 W/m2.

- page 21, line 481 : and small negative offset -> and a small negative offset.

- page 21, line 484 : in upper-level cloud -> in upper-level clouds.

- page 21, Fig 7b : should the orange bar represent -0.14 Wm-2? It looks larger.

- page 23, line 525-526 : Are there two experiments with prescribed CDNC : piClim-control-fixedCDNC and piClim-aer-fixedCDNC? "By comparison with the main piClim-aer" : shouldn't be added "and piClim-control"?

- page 23, line 532-533 : To complete the breakdown, ... : I assume this was on the main piClim-control and piClim-aer simulations, and not on the ones with fixed CDNC. This is maybe not so clear from the text.

- page 24, line 566 : "over the South Pacific" : looking at the figure, it is not so clear that the South Pacific stands out more than other regions.

- page 24, line 573 : is increased -> are increased.

- page 25, line 596 : tropics -> subtropics.

- page 25, line 602-603 : due to experimental setup -> due to the experimental setup.

- page 25, line 609 : 15 % -> 15% (there is a blanco space between "15" and "%" in the text).

- page 26, Eq. 9, and line 630 : CH should not be written in italic.

- page 26, line 630 : where ... is ... the concentrations -> concentration.

- page 26, line 631 : there is apparently no blanco space after "piClim-control,".

- page 28, line 669-670 : I suggest to write "hydroxyl radical" and "nitrate radical".

- page 28, line 674: it doesn't -> it does not.

- page 28, line 680-681 : maybe add at which altitude. It is mentioned in the caption of the figure, but it is maybe informative to mention it also in the text.

- page 29, line 699 : Global distributions of the multi-annual distributions : twice "distributions".

- page 29, line 707 : is changes to OH -> are changes to OH.

- page 30, line 710-714 : how is the ari from NOx calculated? How in general are the ari/aci from NOx and VOC calculated?

- page 30, line 731 : as was the case with NOx : is meant here that the same mechanisms are active related to OH? As the change of OH is however opposite, the forcing is also opposite.

- page 30, line 735 : and CO2 -> and CO2 response.

- page 30, line 735-739 : maybe one can mention explicitly that part of this message was already given earlier (on page 30, line 718-720).

- page 31, line 753-754 : Is this true : are these the two main reasons (the fact that there are interactions, and the fact that there is non-linearity)?

- page 31, line 764 : closely correlated : the correlation seems not that high (-0.44). In addition, from Figs. 12a and 12b it is not so easy to see that there is an anti-correlation. So I would not write "closely" correlated.

- page 31, line 767 : "may be" : can this not be said with more certainty? It seems like a sound explanation.

- page 31, line 770 : with good correlation with -> correlating well with.

- page 31, line 771-772 : cloud -> clouds (twice).

- page 32, line 778 : ", (h)" -> ", and (h)".

- page 33, line 782 : This study also attempts ... : it seems to be a bit a sudden introduction. Maybe one can first introduce the topic in general, and then say that it is also a focus of this study.

- page 33, line 782 : interaction between the aerosol and O3 forcings : is it really an interaction between the forcings which causes this?

- page 33, line 784 : particularly in the net CS components : it is a bit strange to focus on the the net CS component, as the difference in SE CRE is even bigger.

- page 33, line 784 : Firstly we calculate the aerosol IRFs. How are the IRFs calculated?

- page 33, line 792 (not shown) and Fig. 13: it might be interesting to add a figure panel with the differences in O3 profiles.

- page 34, line 805-807 : when reading this sentence, is seems that more attention is given to the +0.07 Wm-2 effect (i.e., the change from -0.39 to -0.32 Wm-2), than on the value of -0.32 Wm-2 itself.

- page 34, line 817 : and its cloud-free, aerosol-free, component -> and its cloud-free and aerosol-free component.

- page 35, line 822 : The seasonality of ... cause -> causes.

- page 35, line 830 : hist -> historical (the official name of this CMIP6 experiment).

- page 35, line 840 : As summarized above ... : maybe this is not a very good introduction - I don't think that "summarized" is the best way to refer to earlier text.

- page 35, line 848-850 : What is the motivation for this sentence? Why is it mentioned that ozone is prescribed, but not, e.g., that oxidants like OH, NO3, and H2O2 are sometimes prescribed in models?

- page 36, line 857 : AF19 : this abbreviation is used only three times - I would think that it does not make so much sense to define it.

- page 36, line 866 : what is GC3.1? Should probably be HadGEM-GC3.1.

- page 36, line 866 : (well within the uncertainty range) : maybe one can add which uncertainty range is meant.

SECTION 5

- page 37, line 887 : paper -> study.

- page 37, line 900 : may result : cannot it be expressed more firmly?

- page 37, line 900 : coming from cloud top -> coming from cloud tops.

- page 38, line 942 : reproduces -> reproduces well.

- page 38, line 949 : we consider -> we suggest.
* * *

---

## Author Comment (AC1) · 31 Jul 2020

Author Responses to the Reviews Received from acp-2019-1152: "Assessment of pre-industrial to present-day anthropogenic climate forcing in UKESM1" by F. M. O'Connor

On behalf of my co-authors, I would also like to thank the two anonymous reviewers for their positive and constructive reviews of the submitted manuscript.

During the discussion phase, we discovered a bug in the experimental set of UKESM1 in the preindustrial atmosphere-only timeslice experiment that acts as the control experiment (*piClim-control*) for all of the other perturbation experiments (*piClim-X*) discussed in the paper. Essentially, in the atmosphere-only configuration of UKESM1, we prescribe the land surface characteristics using output for vegetation fractions, leaf area index (LAI), and canopy heights from the coupled model. We found that the seasonal cycle in the pre-industrial climatology for LAI was 6 months out of phase, affecting surface albedo and hence, the top-of-atmosphere radiative fluxes.

As a result, we have re-run the simulations and re-evaluated all of the present-day effective radiative forcings discussed in the manuscript. All of the original data has been retracted from the Earth System Grid Federation (ESGF) and the updated data and an erratum have now been published: <a href="https://errata.es-doc.org/static/view.html?uid=5e70c479-9b19-ca91-a9ba-d11febe15377">https://errata.es-doc.org/static/view.html?uid=5e70c479-9b19-ca91-a9ba-d11febe15377</a>

As a result of the bug, and as agreed by email with the topical editor, we will make two sets of modifications to the revised manuscript from that submitted. The first set of changes will be to update the manuscript to take account of the LAI bugfix – these changes will be shown in the marked up manuscript in blue, e.g., the methane ERF is old value new value.

The second set of modifications will address the reviewers' comments. Below, we iterate the reviewers' comment in black with grey shading and our response to their comment will be *black italics*. Any new text will be shown in the marked up manuscript in red, and any deleted text will be like this: deleted text.

I thank you again for your comments.

Regards,

Fiona O'Connor (on behalf of all co-authors)

**First Set of Modifications:**

As indicated above, the LAI bugfix meant that all simulations were re-run, and all the ERFs were reevaluated. As a result, figures, some section text, and some tables all required updating. The following tables were updated:

Table 3 (Page 14-16 of the marked-up manuscript)

Table 4 (Page 21 of the marked-up manuscript)

Table 5 (Page 41 of the marked-up manuscript)

The following figures were replaced:

Figure 2 (Page 17 of the marked-up manuscript)

Figure 3 (Page 20 of the marked-up manuscript)

Figure 4 (Page 24-25 of the marked-up manuscript)

Figure 5 no longer on piClim-N2O but on piClim-HC (Page 27 of the marked-up manuscript)

Figure 6 (Page 30 of the marked-up manuscript)

Figure 7 (Page 32-33 of the marked-up manuscript)

Figure 8 (Page 34 of the marked-up manuscript)

Figure 9 (Page 36 of the marked-up manuscript)

Figure 10 (Page 39 of the marked-up manuscript)

Figure 11 (Page 43 of the marked-up manuscript)

Figure 12 (Page 48 of the marked-up manuscript)

Figure 13 (Page 50 of the marked-up manuscript)

Figure 14 – now removed

Figure 15 - now re-labelled as Figure 14 (Page 52 of the marked-up manuscript)

The following sections were updated:

Abstract - Pages 1-2 of the marked-up manuscript

Section 4.1 (Overview) - Pages 13-21 of the marked-up manuscript

Section 4.2.1 (CO2) – Pages 21-22 of the marked-up manuscript

Section 4.2.2 (N2O) – Pages 22-26 of the marked-up manuscript

Section 4.2.3 (now re-named as Ozone Depleting Substances) – Pages 26-27 of the marked-up manuscript

Section 4.2.4 (Methane) - Pages 28-31 of the marked-up manuscript

Section 4.2.5 (GHG) – Pages 28-31 of the marked-up manuscript

Section 4.3 (Aerosols) – Pages 31-35 of the marked-up manuscript

Section 4.4 (Ozone precursors) - Pages 35-45 of the marked-up manuscript

Section 4.5.1 (NTCF) – Pages 45-50 of the marked-up manuscript

Section 4.5.2 (Land use) – Pages 51-53 of the marked-up manuscript

Section 4.5.3 (Total anthropogenic) – Pages 53-54 of the marked-up manuscript

Section 5 (Conclusions) – Pages 54-57 of the marked-up manuscript

**Response to Reviewer #1:**

This paper is an important documentation of effective radiative forcing in a leading climate model, and should be published following modifications.

For reasons that I understand and appreciate, the paper has a rushed and inconsistent feel to it and is poorly written in some places. I have a large number of comments, some of them rather important for clarification of what has been done, and what the results indicate. Hence, I have recommended the paper needs major modifications before I believe it will be suitable for publication.

Response: Thank you for your positive comments and for your detailed constructive feedback on the submitted manuscript. We aim to address your detailed comments/suggestions in our responses below and in the changes applied to the manuscript.

**Major Comments:**

There is a lack of clarity about some of the experiments which led to significant confusion. Some of it was dissipated by the text later in the paper but, for some, "mystery" persisted. Although I suspect these labels follow the various MIP protocols, they are unhelpful to a reader less immersed in the world of MIPs and clearer information, early in the paper, of what is and what is not included in various experiments is necessary. While Table 3 goes some way to doing this, it leaves various issues unclear; it, or an accompanying table, needs to be much more specific about what is, and what isn't included in different integrations

Response: Thank you for this comment. Unfortunately, we had no control over the experiment names used in the manuscript – these were pre-defined by the protocols for the Radiative Forcing Model Intercomparison Project (RFMIP; Pincus et al., 2016) and the Aerosol and Chemistry Model Intercomparison Project (AerChemMIP; Collins et al., 2017). However, we appreciate that the experiment names and/or the experimental set up for the different simulations may not have been clear enough for those not directly involved in the MIPs. In order to address this, we have done the following:

1. In Table 1, we added an additional column to distinguish between those halocarbons that are ozone depleting substances (ODS) and those that are not. It clarifys that the piClim-HC simulation only perturbs ODSs and makes the difference in the present-day perturbations applied to the halocarbons between the experiments piClim-HC and piClim-GHG clearer.

See Table1; pages 8-9 in the marked-up document

2. We also re-named Section 4.2.3 to further emphasise that piClim-HC is only considering ozone-depleting halocarbons as follows:

4.2.3 Ozone (O3) Depleting Substances (ODSs)

See Page 26 in the marked-up document

3. In Table 1, we expanded the column titles to indicate that for greenhouse gases including methane, concentrations are prescribed and that for the aerosol, aerosol precursors and ozone precursors, emissions are used. Table 1 has a separate column already for methane and by expanding the column title names, we label the tropospheric ozone precursors as "O3 (VOC, CO, NOx) precursor emissions". We have also moved the aerosol column so that methane and the other ozone precursors are now side by side, but considered separately. This separation between methane and the other ozone precursors was inherited from

AerChemMIP and so was retained here for consistency. In addition, instead of referring to non-VOC or non-NOx, we have made them more explicit, e.g., "non-VOC" is now replaced with "NOX" and "non-NOX" is replaced with "VOC & CO". And we have also added text to Section 3 to clarify the experimental setup of piClim-VOC, piClim-NOx, and piClim-O3 as follows:

"In all the experiments in Table 1, the emissions of primary aerosol and aerosol precursors (BC, Organic carbon (OC), and sulphur dioxide (SO2)) and O3 precursors (VOC, CO, and NOx) excluding CH4 for 1850 and/or 2014 were taken from Hoesly et al. (2018) and van Marle et al. (2017). In the case of the NOx emissions perturbation experiment (piClim-NOx), both surface and aircraft emissions were changed from PI to PD levels. In piClim-VOC, both VOC and CO emissions were changed from PI to PD levels while the experiment piClim-O3 perturbs emissions of VOC, CO, and NOx only, with the CH4 concentration remaining at PI levels. "

See Section 3: Lines 274-279 in the marked-up document and Table 1: pages 8-9 in the marked-up document

And we changed the title of Section 4.4 to "4.4 Ozone (O3) precursor (VOC, CO, and NOx) gases"

See page 35 in the marked-up document

- 4. There was one experiment missing from Table 1: piClim-4xCO2phys. This has now been added See Table 1: pages 8-9 of marked-up document.
- 5. The way the greenhouse gas concentration in UKESM1 is prescribed differs for the different greenhouse gases. As a result, we added more detail on how these experiments were set up as follows:

"In all cases, the GHG concentrations for 1850 and/or 2014 were taken from Hoesly et al. (2018), van Marle et al. (2017), and Meinhausen et al. (2017). However, the recommended concentrations for the different GHGs in UKESM1 are implemented differently. In the case of CO2, the prescribed concentration is uniform in mass mixing ratio throughout the model domain. For CH4 and nitrous oxide (N2O), the recommended concentrations are treated as lower boundary conditions (LBCs); their 3D distributions are modelled interactively by the UKCA chemistry scheme (Archibald et al., 2020) and coupled to radiation. For ozone depleting (OD) halocarbons in piClim-HC, their concentrations are prescribed separately (and consistently) in UKCA and in the radiation scheme. For the UKCA chemistry scheme, LBCs are prescribed for trichlorofluoromethane (CFC11), dichlorodifluoromethane (CFC12), and methyl bromide (CH3Br), all of which include contributions from other chlorine- and brominecontaining source gases not explicitly treated in UKCA. This approach ensures that the correct stratospheric chlorine and bromine loadings are used for the PD period. Further details on the species included in the CFC11, CFC12, and CH3Br LBCs can be found in Archibald et al. (2020). For the radiation scheme, the radiative effects of ODSs are handled by prescribing the mass mixing ratio of a lumped species (CFC12-eq) uniformly throughout the atmosphere, consistent with the UKCA LBCs, Finally, the piClim-GHG experiment collectively perturbs all the GHGs from PI to PD levels, including non-OD halocarbons. For these latter GHGs, a uniform mass mixing ratio of a lumped species (HFC134a-eq), provided by Meinhausen et al. (2017), is prescribed in the radiation scheme."

See Section 3 (page 10; Lines 258-272) in the marked-up manuscript.

1. L33 starts the confusion about what NTCF is and whether it includes (following AR5) methane and the shorter-lived halocarbons. So a clear definition early in the paper is needed (or as indicated below, NTCF is not used at all, as it is incorrectly used as a shorthand for aerosol and ozone precursors).

Response: Although near-term climate forcers (NTCFs) include methane and short-lived halocarbons (e.g., Myhre et al., 2013), in the context of the AerChemMIP experimental protocol (Collins et al., 2017), the experiment piClim-NTCF only alters emissions of aerosol and aerosol precursors (SO2, BC, OC) and the ozone precursors (VOC, CO, and NOx). It does not include perturbations to methane and/or other short-lived climate forcers. To make this clear, we have modified the abstract as follows:

"Together, aerosol and  $O_3$  precursors (called near-term climate forcers (NTCFs) in the context of AerChemMIP) exert an ERF of -1.03 ± 0.04 W m-2, mainly due to changes in the cloud radiative effect (CRE). " – See lines 38-40 of the marked-up document

And a sentence was added to the section on the Experimental Set up (Section 3) to make this clearer as follows:

"Finally, although near-term climate forcers (NTCFs) include CH4 and short-lived halocarbons (e.g., Myhre et al., 2013), in the context of the AerChemMIP protocol (Collins et al., 2017), the experiment piClim-NTCF does not perturb concentrations of CH4 or other short-lived GHGs. It only changes emissions of aerosol and aerosol precursors (SO2, BC, OC) and O3 precursors (VOC, CO, NOx) from PI to PD levels; it is also referred to as piClim-aerO3 in the RFMIP protocol (Table 1)."

**(See Lines 279-283 of revised manuscript)**

2. L33: Even after reading the paper, I did not really understand what "tropospheric ozone precursors" means and whether the impact that these have on lower stratospheric ozone (e.g. following what was shown in the Sovde et al. papers) was included or excluded from the analysis. Was ozone change above the tropopause prevented in piClim-O3 runs (and the associated individual runs such as piClim-NOx)? If not, I would class it as quite misleading to call this "tropospheric ozone" and would suggest an alternate name (such as "ozone reactive precursor gases" to make it clearer). If it was prevented, then it needs to be made clear what the criteria were for preventing ozone change in the stratosphere. There is the parallel issue for pi-Clim-HC, and whether it allows tropospheric ozone change resulting from the ODSs. An additional confusion is whether CH4 is classed as an ozone precursor. Again, it took some time before this became clear (or more precisely, less unclear) and Table 3 didn't help in this regard.

Response: In response to previous comments, we have modified the manuscript to make clear that the perturbation experiments involving what were previously called "tropospheric ozone precursors" only consider perturbations to VOC, CO, and NOx emissions; methane is considered separately. As indicated above, these have been re-named as "O3 precursors" (Table 1; Pages 8-9 of marked-up document). And in response to previous comments, we added the following to Section 3:

"In all the experiments in Table 1, the emissions of primary aerosol and aerosol precursors (BC, Organic carbon (OC), and sulphur dioxide (SO2)) and O3 precursors (VOC, CO, and NOx) excluding CH4 for 1850 and/or 2014 were taken from Hoesly et al. (2018) and van Marle et al. (2017). In the case of the NOx emissions perturbation experiment (piClim-NOx), both surface and aircraft emissions were changed from PI to PD levels. In piClim-VOC, both VOC and CO emissions were changed from PI to PD levels while the experiment piClim-O3 perturbs emissions of VOC, CO, and NOx only, with the CH4 concentration remaining at PI levels. "

**(See Section 3; Lines 274-279 in the marked-up document)**

For the ERF estimates, there is no masking applied. Atmospheric composition and the atmospheric state are both allowed to fully adjust in the perturbation experiments such that the ERF encompasses all changes to the TOA radiative fluxes due to whole-atmosphere changes in composition (including ozone) and/or other rapid adjustments (e.g. temperature, humidity etc..) as well as land surface rapid adjustments. The following is added to Section 4:

"ERF =  $\Delta F$ , where  $\Delta F$  is in response to whole-atmosphere changes in composition and/or other rapid adjustments; no masking of the response is applied. For example, the ERF quantified from piClim-HC minus piClim-control includes the direct radiative effect from the increase in ODS concentrations, the indirect radiative effect of the whole-atmosphere O3 response as well as other changes to the TOA radiative fluxes due to whole-atmosphere and land surface rapid adjustments."

(See Section 4; lines 324-328 of the marked-up manuscript)

It is only when we consider the radiative forcing by tropospheric ozone – in order to compare against Stevenson et al. (2013), the IPCC 5th assessment report (Myhre et al., 2013), and other estimates – that we mask off the stratosphere and only consider the tropospheric O3 response. This was done solely to make the comparisons with Stevenson et al. and other estimates as comparable as possible. However, we recognise that methane and the other ozone precursors may affect ozone in the lower stratosphere and likewise, that ozone depleting substances may affect tropospheric ozone – we estimate the offset to tropospheric ozone radiative forcing by ODSs in Section 4.4.2. We add a statement to Section 4.4.2 to make clear that masking has been applied in this case:

*"While the ERF captures changes in radiative fluxes at the TOA due to whole-atmosphere responses, here, we mask off the stratosphere and focus solely on the tropospheric O3 response."*

(Section 4.2.2; Lines 794-795 of marked-up document).

We do not assess the RF due to stratospheric ozone changes or total ozone changes. Stratospheric ozone depletion is discussed in the Morgenstern et al. paper in the context of the multi-model assessment of the ERF from piClim-HC. There is a paper accepted for publication by R. Skeie, which is a comprehensive assessment of the historical evolution of total ozone forcing from the CMIP6 models. It includes a detailed discussion of the results from UKESM1. For this reason, we do not explicitly discuss stratospheric ozone RF or total ozone RF here but refer to the relevant papers.

3. L160: piClim-HC gave me even more trouble. I actually thought HC stood for "hydrocarbon" until quite late in the paper. I then found it meant "halocarbon" but it was being used interchangeably with ODS, which was strange until I learnt that non-ODS halocarbons were excluded, for reasons that are not clear to me given the name of the experiment. And it wasn't until later in the paper that it became clear that the impact of HCs on ozone was included in these experiments, but still whether this influence remains stratospheric is unclear. This needed more detail in Table 3 or a new table making clear what is what.

Response: As indicated above, the experiment name piClim-HC was inherited from AerChemMIP (Collins et al., 2017). However, we acknowledge the potential confusion that arose because this was not made clear. As a result, we have added further details to Table 1 and to the manuscript (as

indicated above) to clarify what greenhouse gases are perturbed in both piClim-HC and piClim-GHG and how the perturbation to halocarbons differ between the two (See Table 1 on pages 8-9 of marked-up manuscript; Lines 256-265 of marked-up manuscript).

We also specify that the ERF derived from piClim-HC minus piClim-control is the result of the direct radiative effect of ozone-depleting halocarbons, the ozone depletion that they cause, and any other rapid adjustments in the following:

"ERF =  $\Delta F$ , where  $\Delta F$  is in response to whole-atmosphere changes in composition and/or other rapid adjustments; no masking of the response is applied. For example, the ERF quantified from piClim-HC minus piClim-control includes the direct radiative effect from the increase in ODS concentrations, the indirect radiative effect of the whole-atmosphere O3 response as well as other changes to the TOA radiative fluxes due to whole-atmosphere and land surface rapid adjustments."

(See Lines 324-328 of the marked-up document).

4. L160: To find that piClim-VOC really meant VOC+CO was also confusing.

Response: As indicated previously, we do not have the option of changing this experiment name. However, as indicated previously, we updated Table 1 and added clarity on what was perturbed in piClim-VOC - See Section 3; Lines 278-279 and Table 1: pages 8-9 in the marked-up document

5. L160: Added clarity would be given if Table 3 could make clear the distinction between experiments that impose changes in emissions and those that apply changes in concentrations (or in lower boundary conditions). Even after reading the paper, I am not entirely clear on some of these (e.g. methane). For this table, HC should be spelt out (or replaced by ODS) and it should be explicit what gases are included in the Trop O3 precursors. The authors should also reconsider whether "aerO3" is a better descriptor than NTCF, or removed entirely.

Response: Thank you for this suggestion. We have modified the column titles in Table 1, as indicated previously. And we have altered the titles of the rows in Table 3 to make clear whether a change in concentration or emissions is responsible for the quoted ERFs – See Table 1 on pages 8-9 and Table 3 on pages 14-16 of marked-up manuscript.

6. An additional and quite prevalent issue is that many of the important references are to papers that are submitted (but unavailable) and in some cases only "in preparation" – some key results was in these papers and I felt a bit teased by the allusion to these, without being given either sufficient information or the underlying punchline. I am particularly concerned about the overlap between this paper and Andrews et al. (2019 - submitted) which, from the title of that paper includes forcings in UKESM1, and the Morgenstern et al (2019 - in preparation) paper that contains a discussion of a key result (HC-driven net forcing is strongly negative) when no details at all are presented here, beyond a cursory (4 line) section. This also leads to a very mixed feeling to the paper. Tropospheric ozone is handled in depth, while stratospheric ozone is dealt with in a cursory way, with a reference to the in preparation paper.

Response: We acknowledge that this was problematic and resulted from a high number of papers being submitted ahead of the submission deadline for papers to be considered in the 6th assessment report of the IPCC (December 2019). We were definitely reticent about what to include here on the results from piClim-HC because, at the time, the Morgenstern et al. manuscript was in review.

*Nevertheless, in direct response to your concern about the overlap with the Andrews et al. (2019) paper, we can now confirm that Andrews et al. has been published – see https://aqupubs.onlinelibrary.wiley.com/doi/10.1029/2019MS001866*

Despite the title, the only forcings discussed in that paper are based on the UK's physical climate model, HadGEM3-GC3.1 from the RFMIP experiments. Our paper is quite distinct - it quantifies ERFs in UKESM1 and includes the RFMIP experiments as well as the more comprehensive set of experiments from AerChemMIP. As a result, there is a greater range of ERFs investigated in this manuscript and for those RFMIP experiments that are common between the two papers, there may be differences in ERFs due to the additional Earth System interactions included in UKESM1. As a result, we include comparisons with the estimates from Andrews et al., (2019), where appropriate.

In the case of Morgenstern et al., that paper has subsequently been re-scoped, includes additional models, and has been re-submitted and hence, still under review. However, the leaf area index bugfix outlined above has substantially reduced the magnitude of the present-day ERF from piClim-HC (from -0.33 in the submitted manuscript to -0.18 W/m2 in the revised manuscript). It is still an interesting result and we now include a short section on the ERF from piClim-HC for completeness and a new figure (Section 4.2.3, page 26-27 of the marked-up manuscript).

In relation to the balance between the treatment of stratospheric and tropospheric ozone forcing, this was in large part due to the Morgenstern et al. and Skeie et al. papers. Neither of these papers discuss tropospheric ozone RF in detail or attribute it to different factors. Hence, why we focussed more on tropospheric ozone RF in this paper. The impact of stratospheric ozone depletion (from piClim-HC) on the present-day forcing by tropospheric ozone was quantified in this study. And we now acknowledge that by masking off the stratosphere, we are not fully accounting for the RF resulting from methane and the other precursor gases (e.g., Sovde et al., 2011 & corrigendum):

See Section 4.4.2 - lines 815-818 and lines 823-825 of the marked-up manuscript

*Of the other papers that were cited but were in review or in preparation at the time of submission, we can now confirm that the following are available:*

Andrews et al., https://aqupubs.onlinelibrary.wiley.com/doi/10.1029/2019MS001866, 2019.

Andrews and Forster, https://www.nature.com/articles/s41558-020-0696-1?proof=t, 2020.

Bellouin et al., https://aqupubs.onlinelibrary.wiley.com/doi/full/10.1029/2019RG000660, 2020.

Mulcahy et al., https://www.geosci-model-dev-discuss.net/gmd-2019-357/.

Sellar et al., https://agupubs.onlinelibrary.wiley.com/doi/full/10.1029/2019MS001739, 2019.

Thornhill et al., https://www.atmos-chem-phys-discuss.net/acp-2019-1205/.

These citations have been updated – See Lines 1316-1318; 1320-1321; 1344-1348; 1559-1563; 1644-1650; 1731-1735 in the marked-up manuscript.

7. I am concerned that it is hard to understand what features in the geographical plots are signal and which result from unforced variability. This needs to be discussed in more detail, especially for forcings such as N2O that are inherently small. This is one clear disadvantage of the ERF and partly the reason for my comment at 104/127, below, about ERF being the metric of choice. I do not believe it is the metric of choice in all circumstances, especially for small forcings.

Response: We thank the reviewer for this comment. We have now adapted Figures 3-6 (Page 20, Page 24, Page 27, and Page 30) and Figures 8 (Page 34), 12 (Page 48), and 14 (Page 52), such that areas of the globe, where the unforced variability is outside of the 95 % confidence level, are masked out in white. Indeed, taking N2O as an example, the ERF over most of the globe is not statistically significant at the 95 % confidence interval, except for the longwave clear-sky (LWcs') component, which has a clear signal mainly over the oceans. This is made clear in the Figure captions and when referring to the contributions of the different components to the total ERF (See Lines 386-412 of revised manuscript).

**Other Comments (but note that these are not exhaustive)**

25: "altering the sign" – it would help readers to know what the sign has been altered from and to.

Response: This has changed due to the updated simulations following the LAI bugfix. However, we have been more explicit with the updated results:

"Ozone (O3) precursors gases consisting of volatile organic compounds (VOCs), carbon monoxide (CO), and nitrogen oxides (NOx) but excluding CH4, in addition to exert a positive radiative forcing due to increases in  $O_3$ . However, they also lead to oxidant changes, which in turn cause an indirect aerosol ERF. The net effect is that the ERF from PD-PI changes in NOx emissions is negligible at 0.03 ± 0.04 W m-2, while the ERF from changes in VOC and CO emissions is 0.33 ± 0.04 W m-2."

See Abstract Lines 34-38 of marked-up manuscript

49: I would say the change of concentrations resulting from changing emissions is the first part of the cause-effect change

Response: This has now been altered as follows:

"An important part of this cause-effect chain from activity to climate response, mediated through the atmosphere and the land surface, is quantifying changes to the Earth's radiation budget, often termed radiative forcing."

See line 59 of revised manuscript

65-94: The intro, with its focus on LLGHGs and tropospheric ozone, feels unbalanced. Why are other forcings not discussed here (or is this detail inappropriate here)? The split between tropospheric and stratospheric ozone forcing is to some extent artificial and a little dated (see e.g. Sovde et al.), because the mapping of the drivers of ozone change does not map exclusively on to ozone change in the troposphere and stratosphere, and that could be clearly acknowledged 104 and 127: "preferred" and "metric of choice" – by who?

Response: We agree that the introduction to the study was unbalanced. In response, we have added some background on aerosol forcing (Lines 108-120 of revised manuscript) as follows:

"Aerosol forcing involves a wide range of physical processes. These include (i) direct changes to the radiation budget through scattering and absorption of both SW and LW radiation (e.g. Haywood and Boucher, 2000), (ii) indirect impacts on the radiation budget by changing the microphysical properties of clouds (Twomey et al., 1977), and (iii) changes in the distribution of cloud cover or condensate that follow on from perturbations in cloud microphysics (Albrecht 1989) or radiative heating by aerosol (Hansen et al., 1997). Direct aerosol radiative forcing can be calculated using offline radiative transfer models in a similar manner to GHG and O3 forcing, whereas assessing impacts of aerosols on clouds requires simulations in atmospheric models. The 5th assessment report

(AR5) of the IPCC recommended the ERF framework as an effective way of assessing the overall aerosol forcing as it enables the more complex cloud impacts to be evaluated as part of the climate's rapid adjustments (Myhre et al., 2013). To simplify terminology, AR5 also made a clear distinction between components of the forcing driven by Aerosol-Radiation-Interactions (i.e. the direct or instantaneous forcing) and Aerosol-Cloud-Interactions (that include all indirect or semi-direct cloudrelated forcings). Despite wide-ranging and on-going research, the role of aerosols remains the leading source of uncertainty in PI to PD climate forcing, due to the difficulty in constraining the sensitivity of clouds to changing microphysical processes (Bellouin et al., 2019)"

And we have added a section on land-use RF estimates (See lines 122-131 of revised manuscript) as follows:

"In the case of land use, RF estimates have been made using a single GCM simulation with a doublecall to the radiation scheme (e.g. Betts et al., 2007) or by comparing paired simulations that include rapid adjustments (e.g. Andrews et al., 2016). However, the choice of RF calculation is not the major source of differences in RF estimates. Similar to O3, uncertainty in PI land-cover is a major source of uncertainty in land-use RF (e.g. de Noblet-Ducoudré et al., 2012). Historically, deforestation has been the dominant type of land-use change, and this causes a positive RF due to increased CO2 emissions and a negative RF due to increased surface albedo. Here, we include the effects of land-use CO2 emissions in the CO2 RF estimates and the land-use RF is due to biophysical changes, predominately albedo. Deforestation has a much larger effect on albedo in snowy regions and model biases in snow cover also contribute to uncertainty in land-use RF (Pitman et al., 2011). Land-use RF estimates also vary due to different time periods being considered (Myhre et al., 2013a), because, unlike many other forcings, there was substantial land-use change before the industrial revolution."

In response to the comment on ozone and the strat/trop split, we agree with the reviewer's comment. In the manuscript, we acknowledge it in the Introduction and add the Søvde et al. (2011) reference:

"Other uncertainties in Stevenson et al. (2013) arise from neglecting the change in O3 in the lower stratosphere attributable to changes in O3 precursors and the contribution from stratospheric O3 depletion on the modelled changes in tropospheric O3 (e.g., Søvde et al., 2011a; 2011b)."

See Lines 102-104 of revised manuscript

In relation to the ERF being the metric of choice, we have now moved the first statement (and removed the duplicate statement) and provided an additional reference to support it as follows:

"Although the uncertainty associated with an ERF tends to be larger than that of RF (or SARF), it is more representative of the climate response than the traditional RF (Hansen et al., 2005; Forster et al., 2016). As a result, it is now the preferred metric of choice for ranking the drivers of climate change (Boucher et al., 2013; Forster et al., 2016)."

See Lines 158-161 of revised manuscript.

117: Xia et al – although it would be tiresome to say "found in their model" on all occasions in this paper, I think it is important in cases, as it is a result of a single model study

Response: Added (Line 154 of marked-up manuscript)

119-121: This needs clarification. For RF it is a requirement that tropopause and TOA forcings are identical (not "nearly identical") following SARF, so in a sense the TOA/trop distinction doesn't matter. However, where it does matter is that RF requires the specification of the tropopause, which

is always to some extent arbitrary. whereas ERF does not – the model just does its own thing. This is one advantage of ERF.

*Response: Thank you for this comment. We have now added more clarity following the suggestion as follows:*

"The SARF and ERF differ in where the change in radiative fluxes is diagnosed. The SARF, diagnosed at the tropopause, requires the tropopause to be defined but the ERF has the advantage of being diagnosed at the TOA, with no need for a tropopause. While the SARF at the tropopause and TOA are, by definition, identical, this is not the case for the ERF; climate forcings can lead to adjustments in stratospheric circulation and therefore changes to dynamical heating above the tropopause. Changes in dynamical heating are thus balanced by radiative divergence across the stratosphere, explaining the change in net fluxes between the tropopause and TOA."

See Lines 167-172 of marked-up manuscript

154: "addressed the strong negative" – the nuance here (and in other places in the paper) is that it was too strong negative (i.e. unrealistic). Is that the intention?

Response: We have altered the statement to state explicitly that the aerosol ERF in the GA7.0 configuration of HadGEM3 was -2.75 W m-2, leading to an unrealistic negative total anthropogenic forcing for the 20th century. The improvements implemented in Mulcahy et al. reduced the magnitude of the aerosol ERF to -1.45 W m-2 (Lines 209-213 of marked-up manuscript)

268: "from energy budget constraints" – I have no idea what this means and hence whether the Andrews and Forster (2019 - submitted) paper is in some ways a superior assessment to that given here. Should we (or Andrews and Forster) be concerned that the UKESM forcing falls outside their stated range? This is another problem with the heavy reliance on unpublished papers.

Response: The Andrews and Forster study has now been published and the citation updated (Lines 1320-1321 of marked-up document) – see https://www.nature.com/articles/s41558-020-0696-1?proof=t. We have now added more detail on the approach used by the Andrews and Forster study to constrain the total anthropogenic ERF at the present day (Lines 370-372 of marked-up document). In addition, the updated simulations with the corrected seasonal cycle in the prescribed leaf area index result in the total anthropogenic ERF from UKESM1 changing from 1.61 to 1.76 W m-2; this corrected estimate now falls just within the range of estimates from Andrews and Forster.

282: "some evidence" – I did not know what this meant. It either is, or it isn't? Or is this meant to imply that the derived forcing is no larger than the unforced variability in this region, and hence is not a robust signal? There is no indication of statistical significance in any of the plots, or discussion of it and this should be rectified in some way. See Major Comment 7 282: This paragraph is the first place where it is made clear that the HC calculations include the ozone forcing from ODSs (and indeed that HC=ODS) and leaves the reader unclear whether this is the case for (e.g.) methane and CO2.

Response: We have removed any reference to "some evidence" and made the statement more explicit that the GHG ERF is strongly positive everywhere except for the southern hemisphere high latitudes, where it is negative as a result of the contribution from the piClim-HC experiment. As outlined in previous responses, the plot of the top-level forcings (Figure 3: Page 20 of marked-up document) masks out those areas where the signal is not statistically different from zero at the 95 % confidence interval. It means that the negative ERF in the southern high latitudes from the piClim-HC and piClim-GHG experiments is a robust signal and not due to variability. 287: The GHG forcing is rather inhomogeneous. Is this statement that the aerosol forcing is more so based on an objective measure of inhomogeneity or a subjective assessment?

Response: Thank you for this comment. We have since carried out a chi-squared test for homogeneity and found that the global distributions of the ERFs from the top-level experiments (piClim-GHG, piClim-aer, piClim-O3, piClim-LU, and piClim-anthro) are all inhomogeneous. However, it does find that the aerosol ERF distribution plot is more heterogeneous than that for GHGs. We also quantify the percentage of the global area giving an ERF statistically significant from zero and outside the variability at the 95 % confidence level. (Lines 386-412 of marked-up manuscript)

**288-290: Is this conjecture or the result of model analysis?**

Response: This is a statement based on our model analysis. Over bright surfaces, the aerosol IRF can become positive if the aerosol mixture is sufficiently absorbing (e.g. Haywood and Boucher, 2000) and this happens in our model over parts of the Sahara, Arabia and Southern Asia as shown in Fig 3c. To ascertain this, we looked at IRF and ERF from the individual aerosol components (BC, OC, sulphate) as well as the total aerosol ERF. The BC IRF was particularly high in those regions as shown in the figure below.

**292: Here it is unclear whether ozone change from methane is in the GHG or the trop O3 plot**

Response: We have now added in a number of places in the document that methane is considered in the piClim-CH4 experiment, while the other ozone precursor gases (VOC, CO, and NOx) are considered in the piClim-VOC (VOC and CO), piClim-NOx, and piClim-O3 (VOC, CO, and NOx) experiments. We add explicit reference to VOC, CO, and NOx here to directly address this (Lines 402-405 of marked-up manuscript).

296-297: Is the land use forcing over the ocean real (resulting from downstream rapid adjustments) or just indicating unforced variability in the model?

Response: The land use ERF over the ocean was predominantly due to unforced variability and is not a statistically significant signal at the 95 % confidence interval. These areas have now been masked out in white from the new Figure 14 (page 52 of the marked-up manuscript).

298: This negative forcing over the mid-lat continents is a striking result. Is this the first paper to indicate it? Also, presumably the Land Use forcing in these areas is mostly "historical". This paper understandably does not address the time dimension of the evolution of the forcing, but perhaps some added text could be added to indicate when this forcing would have been most active.

Response: The strong negative forcing over mid-latitude continents is a common result in GCM studies of historical land use change (e.g. de Noblet-Ducoudre et al., 2012). In the large regions with a strong negative ERF, forests have been replaced by agriculture, which has remained upto the present day, and the associated SW forcing has also persisted. Comparing transient coupled UKESM1 simulations with and without historical land use change, shows the albedo response in North America gradually grows from the 1850s to the 1960s, with a northward expansion starting after about 1900 (not shown). The albedo response in Eurasia gradually increases from 1900 until the 1980s.

The following text has been added at Lines 1089-1091 of the revised manuscript.

"The large regions of negative ERF in the northern mid-latitudes are consistent with previous model studies (e.g. de Noblet-Ducoudre et al., 2012); they are caused by agricultural expansion and the forcing is expected to have gradually increased from about 1850-1970 in North America and from 1900-1980 in Eurasia."

303: I wonder if adding an additional frame with the zonal-mean forcings of each component would be useful for bringing out the overall larger-scale structures of the forcing.

Response: Figure 3 (Page 20 of the marked-up manuscript) which plots total GHG, aerosol, ozone precursors, land use and total anthropogenic ERFs now includes zonal mean plots. This is a very informative addition and we thank for the reviewer for the suggestion.

303: It is striking that the geographical plots in each figure differ, presumably because different coauthors were responsible for them, with different conventions in the headers, and in the case of Figure 8, a different projection. Could these be made more consistent? Personally, I find the addition of global mean values on each frame, as in Figure 4, quite helpful.

Response: Thank you for this observation and your assumption that different co-authors produced the different figures was indeed correct! We have now made the global distribution plots as consistent as possible, e.g., masking out regions (with white) where the signal is not statistically different from zero at the 95 % confidence interval, adding in global mean values, using the same order of subplots, using the same projection, figure captions, etc..

See Figures 3 (Page 20), Figure 4 (Page 24), Figure 5 (Page 27), Figure 6 (Page 30), Figure 8 (Page 34), Figure 12 (Page 48), and Figure 14 (Page 52).

323: Here and elsewhere it should be made clear if the concentrations are surface values or massweighted means over the whole atmosphere (and whether a single global mean values is used everywhere in the model for each LLGHGs).

*Response: This has previously been addressed in response to another comment - See Section 3 (page 10; Lines 258-272) in the marked-up manuscript.*

328: 121 ppm. Presumably the expressions in Etminan et al. could be used to derive their forcing for the same CO2 change?

Response: Yes, indeed. The equivalent from Etminan et al. is 1.80 W m-2. The revised ERF is, however, 1.89 W m-2. The text has been updated accordingly (Lines 446-447 of the marked-up manuscript).

331: Some of the CO2 SW forcing likely comes from the SW bands of CO2, but this is implicitly discounted here. Or are these bands not included in the radiation code?

Response: Yes, the radiation scheme does include SW forcing from the CO2 bands in the near-IR. However, as you say, the text implicitly discounts this. We have updated the text to note that the direct effect is included but is small compared to the LW based on results from the radiation codes in RFMIP (Pincus et al., 2020) – Lines 453-448 in the marked-up manuscript. Using this assumption, we estimate the clear-sky component of the ERF in the SW is dominated by the rapid albedo response in our model.

**352: These two estimates overlap within the uncertainties.**

*Response: Given the leaf area index bugfix and the new simulations, our updated ERFs are no longer overlapping. The new ERF is now substantially larger than the cited value – See line 480 in the marked-up manuscript.*

358: "noise" – here and throughout, it is important to guide the reader as to which features are noise and which are robust. Could this be done via masking or stippling? For example, is the marked negative feature in clear-sky longwave in Figure 4c over Russia and its surroundings, robust?

Response: Figure 4 (and other figures) have been modified to show only regions where ERF components have significant signals at the 95 % confidence interval; all other regions are masked with white. With updated ERFs shown in the figure, the negative contribution to the ERF from the clear-sky longwave (LWcs') component over Russia and its surroundings are no longer significant (and now appear masked white).

Other global distribution plots also now only show regions where ERFs (and/or its components) have a signal statistically different from zero at the 95 % confidence interval

363-365 and Figure 5: 364 talks about "increased ozone" (in the UTLS) but no hint is given as to what drives this feature. Is it a "self-healing" effect of the overlying depletion? The caption to Fig 5 needs to make clear this is for the N2O experiment.

Response: The updated ERF due to N2O following the leaf area index bugfix now shows a marked difference from the original ERFs. We have now dropped figure 5 and the associated discussion. The reviewer may well be right about the self-healing but we have not assessed that. The finding is consistent with the literature (e.g., Morgenstern et al., ACP, 2018).

396-398: There is no hint of what is going on here, except a reference to a paper that is "in preparation". This is very frustrating and leads me to question whether the HC/ODS results should be shown here at all, if there is no accompanying analysis of the striking result. It became a bit unclear to me why, for example, the tropospheric ozone is discussed in such detail in this paper, but not stratospheric ozone. Also note that the assumption that HC=ODS is made here without elaboration, although later there is some hint at this.

Response: We appreciate the reviewer's comments and have now included a figure and some discussion of the ERF of ozone-depleting HCs. The forcing from stratospheric and/or total ozone forcing are discussed in two other papers: Morgenstern et al. (In review) and Skeie et al. (Accepted) manuscripts. As a result, it would not be appropriate to include here.

414-415: Again, this allusion to a different submitted paper, with no hint of what is found, is frustrating

Response: The O'Connor et al. paper has received reviews but requires some additional analysis before publication. So, rather than expanding on their findings, we include some comparisons of the

methane ERF with the AerChemMIP multi-model ensemble, and cite the discussion paper of Thornhill et al., which is available at:

**https://www.atmos-chem-phys-discuss.net/acp-2019-1205/**

**as follows:**

"The UKESM1 ERF quantified here is at the upper end of estimates from the recent study of AerChemMIP multi-model ERFs by Thornhill et al. (2019). They found that the multi-model mean ERF was 0.70 W m-2, with a standard deviation of 0.22 W m-2. They attributed part of the inter-model spread to different complexities in the representation of interactive chemistry in the respective models i.e. some models only captured the direct radiative effect of CH4 (e.g., NorESM2) while others (e.g., UKESM1) also included indirect contributions from CH4-driven changes in O3 and SWV. However, the contribution to the ERF from tropospheric adjustments differed in both magnitude and sign, with UKESM1 being the only model with a positive rapid adjustment. The relative contributions of the direct and indirect forcings to the total CH4 ERF quantified here and the mechanism behind the positive rapid adjustment can be found in O'Connor et al. (2020)."

See lines 580-588 of revised manuscript

423: The frames in Figure 6 are in a different order to those in Figure 4, and there are a different number. Commonality of presentation would help the reader. Again, the caption to this Figure does not make it clear it is for methane. (All captions should be checked for this – e.g. Figure 12 suffers the same lack of specificity.)

Response: As indicated above, the figures have now been more consistent, both in terms of including global mean values, order of sub-plots, map projection, figure captions, etc. - See Figures 3 (Page 20, Figure 4 (Page 24), Figure 5 (Page 27), Figure 6 (Page 30), Figure 8 (Page 34), Figure 12 (Page 48), and Figure 14 (Page 52) in the marked-up manuscript

440: This is the first place the reader learns that the non-ODS halocarbons are not included in HC.

Response: This has now been made clearer both in Table 1 (Page 8-9), Table 3 (Page 14-15), and in the description of the experimental setup in Section 3 (Lines 253-266 of marked up manuscript). We have also re-named the section (Section 4.2.3 – Page 26) presenting the results of piClim-HC as "Ozone (O3) Depleting Substances (ODSs)" to again make clear that the experiment piClim-HC only considers ozone-depleting halocarbons.

444: "warrant further investigation" – I agree. this difference is interesting and one factor not considered (spectral overlap) would likely push the difference in the wrong direction (i.e. make GHG even lower than the sum of individual components)

Response: As the reviewer points out, this was indeed potentially very interesting. We had quantified IRFs (using an offline radiative transfer code) and found that the sum of the individual IRFs was higher than the total GHG IRF, suggesting that spectral overlap and/or saturation effects were occurring but were in the opposite sense to what the ERFs quantified from the UKESM1 simulations were indicating. We chose not to include this at the time of writing, because we were considering doing a separate study on the non-linearity of the GHG ERF.

However, the apparent non-linearity has somewhat disappeared with the updated simulations, in which we have corrected the seasonal cycle in the prescribed leaf area index! We now find that the sum of the ERFs from piClim-CO2, piClim-CH4, piClim-N2O and piClim-HC is  $2.93 \pm 0.08$  W m-2,

indicating that the ERFs add linearly and agree with that from piClim-GHG (2.92  $\pm$  0.04 W m-2). See the end of Section 4.2 (Lines 626-629 of the marked-up manuscript).

473: There is a minor ambiguity in using "sum", "all" and "total". As I understand the Total=All? This also applies to the text from 513-521.

Response: Yes, the reviewer is correct – we had used "total" and "all" interchangeably. We have removed all references to "total" to avoid any doubt. See lines 649-659 and the caption of Figure 7 (Page 33 of the marked-up manuscript)

484: "many other models" – but aren't versions of HadGEM included in the Stjern et al. 2017 paper?

Response: Well, not quite. Stjern et al. (2017) included HadGEM2 (the UK's CMIP5 model) and an early development version of HadGEM3 that still had the mass-based aerosol scheme (CLASSIC) from HadGEM2. UKESM1 now has a new and totally different aerosol scheme (GLOMAP-mode; Mann et al., 2010; Mulcahy et al., 2019) with quite a different representation for black carbon. As a result, the UKESM1 results presented here are quite separate from those included in the Stjern et al. study.

494-496: This sentence seems out of place, especially without any supporting reference.

Response: Agreed, this has now been removed (Lines 686-687 in the marked-up manuscript)

550: I had a lot of comments on Section 4.4 as it was particularly unclear in my view. One underlying issue, raised above, is whether the tropospheric precursor gases are allowed to change lower stratospheric ozone, and whether that change is included here. Without a clear statement, it is very hard to understand this section and place it in the context of prior work. Similarly, the lack of clarity on whether CH4 is included or not, as an emitted species, is unhelpful.

*Response: We have addressed the lack of clarity with the experimental set up in our previous responses above (Section 3, Table 1, Table 3).*

For the ERF estimates, no masking of the response is applied. However, when estimating the tropospheric ozone RF in order to compare with previous estimates, we mask off the stratosphere. We add the following statement to make this clear: "While the ERF captures changes in radiative fluxes at the TOA due to whole-atmosphere responses, here, we mask off the stratosphere and focus solely on the tropospheric ozone response" to Section 4.4.2 (See Lines 794-795 of the marked-up document).

559: This figure purports to show "tropospheric column" but the panels are labelled in ppbv. Is this the mean (mass weighted) ppbv over the depth of the tropophere? How is the troposphere defined for this purpose? Would Dobson Units be a better measure? Figure 5a included a latitude-height plot of ozone change due to N2O, and a similar plot would be helpful here.

Response: This was a typo, as the data are in Dobson Units. The tropospheric column was calculated by applying a monthly-mean tropospheric mask, derived from the PV+380K level, to the total ozone column. (Figure 9, page 36 of marked-up document)

565-566: I wasn't sure why the "South Pacific" was highlighted and here and when "Western Pacific" is mentioned, I presume in both cases it means tropical Pacific?

*Response: This has been amended to Eastern tropical pacific. (See line 767 of the marked-up document)*

588: By applying this mask, how much forcing (due to the precursors' impact on stratospheric ozone) is excluded from the analysis?

Response: We used a tropospheric  $O_3$  radiative kernel and hence, masked off the stratosphere. However, we acknowledge that this may have the effect of underestimating the resulting tropospheric  $O_3$  RF from methane and the other  $O_3$  precursor gases. For example, Sovde et al. (2011a, b) estimate that of the  $O_3$  RF from methane and the other precursors, approximately 85 % is due to changes in tropospheric  $O_3$ , with the remaining 15 % due to changes in the stratosphere.

We have now added the following statement to the discussion:

"However, in masking off the stratosphere, the UKESM1 estimate for the  $O_3$  RF attributable to  $O_3$  precursors is likely to be underestimated. For example, Sovde et al. (2011a; b) estimates that approximately 15 % of the  $O_3$  response from changes in CH4 and other  $O_3$  precursors may be in the stratosphere, and hence not considered here." (See lines 823-825 of the marked-up document)

594: I found this paragraph hard to follow, and in particular whether differences in the literature were due to radiation code differences (or the effect of applying a kernel) or process level differences in determining the ozone change. I ended up confused as to whether the new result was higher or lower than Stevenson et al. and for what (dominant) reason it was so.

Response: We apologise for any lack of clarity. We have now removed any discussion on the radiation code. Although it contributes to the uncertainty of the tropospheric  $O_3$  RF in Stevenson et al. (2013), it is only one of the contributing factors. Other factors include inter-model spread in  $O_3$  response, emissions uncertainty, definition of tropopause etc. Stevenson et al. estimate an overall uncertainty of  $\pm$  30 % on their central estimate. As a result, our estimate is within the uncertainty of the best 1850-2010 estimate from Stevenson et al. (2013) and is consistent with other estimates. The updated paragraph is as follows:

"Figure 10 shows the global distribution of the PD tropospheric O3 RF from piClim-CH4, piClim-NOx, and piClim-VOC experiments and their sum using the kernel approach. It shows that the tropospheric  $O_3$  RF is strongest over the northern hemisphere tropics and weakest over the southern hemisphere high latitudes. The strongest RF occurs in regions of warm surface temperatures and high albedo, coinciding with the largest tropospheric  $O_3$  change (Shindell et al., 2013a). As was the case in Stevenson et al. (2013), the forcing is weaker over regions of high altitude (e.g., Tibetan Plateau) due to there being less  $O_3$  column aloft to absorb in the LW. The tropospheric  $O_3$  RF from the kernel approach is 414 mW m-2. However, the increase in tropospheric  $O_3$  at the PD (and its resulting RF) are offset by decreases due to ODSs (e.g., Søvde et al., 2011a; Søvde et al., 2011b; Shindell et al., 2013a). Applying the kernel method to the diagnosed decrease in tropospheric O3 from the piClim-HC experiment (Sect. 4.2.3), we find a RF offset of -101 mW m-2. Although larger in magnitude by nearly a factor of 2 than the estimates from Søvde et al. (2011a; 2011b) and Shindell et al. (2013a) due to the strong  $O_3$  depletion in UKESM1 (Keeble et al., 2019; Morgenstern et al., 2020), it reduces our original estimate to 313 mW m-2. This revised estimate is within the 30 % uncertainty of the Stevenson et al. (2013) best estimate, albeit lower than their central estimate by 13 %. It is also consistent with a number of other estimates: the CMIP6 historical  $O_3$  dataset (312 mW m-2 from Checa-Garcia et al., 2018), a recent study in which observational isotopic data was used as a constraint on historical increases in tropospheric  $O_3$  (330 mW m-2 derived from the GEOSChem model in Yeung et al., 2019), and a parametric model based on multi-model source-receptor relationships  $(290 \pm 3 \text{ mW m}^{-2} \text{ from Turnock et al., 2019})$ . However, in masking off the stratosphere, the UKESM1 estimate for the PD O3 RF attributable to O3 precursors is likely to be underestimated. For example,

Søvde et al. (2011a; b) estimates that approximately 15 % of the  $O_3$  response from changes in  $CH_4$  and other  $O_3$  precursors may be in the stratosphere, and hence not considered here. "

See lines 794-818 of the marked-up manuscript

607: "Section 3.2.3" – I presume this means Section 4.2.3 but, as noted above, there is essentially no detail in 4.2.3, and certainly nothing on the tropospheric ozone change in piClim-HC. This is frustrating when trying to make sense of this section.

*Response: Thank you for spotting this error – now corrected. Any discussion on the tropospheric O3 change from ODSs is calculated and discussed within this section – see our response above.*

616: It would be useful to have the global-mean values, perhaps in the same way that they are shown on Figure 4 in each frame. Also, some comment about how different Fig 10d and Fig 3c are (i.e. RF versus ERF) would seem useful.

Response: In updating Figure 10 due to the updated simulations following the LAI bugfix, we now include global mean values – See page 39 of marked-up manuscript. However, we do not add any discussion of the differences between Figure 10d and Figure 3c for a number of reasons:

- 1. Figure 10d includes the impact of CH₄ on the tropospheric O3 RF whereas methane is part of the GHG ERF in Figure 3
- 2. We find that the ozone precursors perturb aerosols and hence, are responsible for significant cloud adjustments, which are included in the ERF plotted in Figure 3c. These adjustments are excluded from the tropospheric O3 SARF plotted in Figure 10d. These adjustments have a considerable impact on the ERF see the CRE component in Table 3 (page 14 of the marked-up document).

666: Are the methane concentrations, the change (i.e. NOx – control, as indicated in Column 1) or the absolute values for NOx. It might be better to show the change?

Response: The concentrations shown for methane were the equilibrium concentration that would have occurred had the model been emissions-driven rather than concentration-driven. However, we have now changed this table to include this as a change in methane concentration, i.e., equilibrium concentration minus prescribed concentration. See Table 5 (page 41 of the marked-up manuscript).

680: Make clear that this is the OH at 1 km.

Response: This has been done - See line 899 of the marked-up document

686: These do not appear large (e.g. compared to NOx)

Response: The word large has been removed – see line 904 of the marked-up document

695: "eastern Pacific" – does this mean the eastern tropical Pacific, as I see changes of both signs in the E Pacific?

Response; Reference to the eastern Pacific has been removed so that the sentence now reads "The impact of NOx emissions appears to dominate, given the similarity between piClim-O3 and piClim-NOx" - see lines 913-915 of the marked-up document

699: AOD is meaningless without specifying a wavelength. I'm curious why the VOC/CO run has an increase in AOD but a decrease in CDNC (e.g. over the W Pacific). Is this because one compares column integrated amounts and the other is at 1 km?

Response: The AOD was calculated for 550nm. The discussion of the second part is somewhat moot as we now believe that many of the AOD changes to not pass significance testing. We agree that column-integrated AOD is not necessarily correlated with CDNC changes at 1km, having chosen this altitude for its relevance to cloud formation processes, particularly in marine environments.

712: "negligible" – is this because of cancellation of regions of positive and negative, but the local values can be large?

Response: On a global mean basis, the aerosol IRF is less than 0.01  $\pm$  0.01 W m-2. Although regionally, the IRF varies between -0.31 and 0.25 W m-2, it is only statistically significant over about 5 % of the globe.

726: Are the NOx sources just from the surface, or are aviation emissions also included? I know it is beyond the scope of this work, but it is of interest whether this net negative applies to all sources or likely just surface sources.

Response: The perturbation experiment piClim-NOx perturbs all anthropogenic emission sources of NOx – both at the surface and from aircraft. A sentence to this effect has now been added to the Section on experimental set up – see line 277 of marked-up manuscript

739: Again, a reference to an "in preparation" paper is unhelpful here.

Response: This paper has been published as a discussion paper and can be found at:

https://www.atmos-chem-phys-discuss.net/acp-2019-1205/

The corresponding citation has been updated (Lines 1731-1735 of marked-up manuscript).

750: I think this would be better labelled as Aerosol/Ozone, rather than NTCF since it clearly isn't all NTCFs (which would avoid the "excluding methane" repetition). Since both aerosol and ozone are treated separately anyway, I am not quite sure of the need for this section, especially as they, together, only form a subset of NTCFs.

Response: As indicated previously, we have chosen to retain the name "Near Term Climate Forcers" to be consistent with the experiment name from the AerChemMIP protocol. However, we note in Table 1 that piClim-NTCF is also known as piClim-AerO3 in RFMIP and we re-iterate that in Section 3 (Lines 283 of marked-up manuscript). We have also chosen to retain this section to indicate that the aerosol and ozone precursor ERFs do not add linearly.

769-771: These lines seem to contradict each other about whether it is cloud fraction or cloud optical depth, or maybe I misunderstand the point being made.

*Response: Any discussion about the LWcre' component and high-level cloud fraction has been removed. The SWcre' component is correlated with the cloud fraction.*

776: Units are missing for the temperature and cloud parameters. Are temperature changes really in K?

*Response: The updated Figure 12 now only shows the different components of the ERF – See Page 48 of the marked-up document*

810-811 "too strong" and "bias still exists" – I haven't read the Robertson paper, but could it be said what the too strong albedo response is relative to (other models? observational constraints?), and

whether this indicates that UKESM is definitely incorrect (as this text implies) rather than being a plausible outlier?

Response: Robertson (2019) evaluates the HadGEM2-ES response to land-use change against observational estimates and shows that the model has an incorrect albedo response. This bias was not addressed in the development of UKESM1 although it may have inadvertently been fixed.

Lines 1069-1072 have been changed to:

"Robertson (2019) showed that, even in the absence of snow cover, the albedo response to land-use change in HadGEM2-ES is stronger than observed and it is likely that this bias still exists in UKESM1."

**Typos etc (but note that these are not exhaustive)**

25: If "forcing" means "ERF" it should say this.

Response: Done (See examples on lines 187, 194, etc. of marked-up manuscript)

83-89: Long sentence 64 and elsewhere: "it's . . . doesn't . . . weren't" – it doesn't worry me much, but such contractions are not usually used in formal scientific writing

Response: These have now been removed (e.g., Lines 95, 893, etc. of revised manuscript)

**216: machine's**

Response: Corrected (Line 307 of marked-up manuscript)

236: These equations use the subscripts cs and clear – do they mean the same thing?

*Response: Yes, they do, and we have now replaced "clear" with "cs" – See lines 332 and 341 in marked-up manuscript*

**552: refer to Figure 3 here?**

Response: Done - See line 751 in marked-up manuscript

592: "Garcia"?

Response: Thanks for spotting this error – Now corrected (See line 798 in marked-up manuscript)

608: I think it important to also refer to the Sovde corrigendum <a href="https://doi.org/10.5194/acp-12-7725-2012">https://doi.org/10.5194/acp-12-7725-2012</a>.

*Response: Agreed; it has now been added – See lines 104 and 823-825 in marked-up manuscript. Reference now added – See line 1721 of the marked-up manuscript*

**Anonymous Reviewer #3:**

To the editor,

I think this work serves two main purposes, i.e., (1) documenting (and disentangling) the ERF from short-lived emissions, long-lived GHGs, land-use, and ODSs in UKESM1, and (2) indicating the relevance of ESM-interactions for estimating the anthropogenic forcing.

I think the work is solid, the subject is well explained, comparisons with earlier studies are included, and the information shared in figures and tables is appropriate.

Presenting estimates of ERF is important to understand the evolution of climate (modeled or observed). Models participating in CMIP6 were motivated to perform specific experiments in AerChemMIP and RFMIP, which would allow a better characterization of the forcings felt by ESMs, e.g., over the historical period. It is important that those ERF estimates are well-documented, as they facilitate the interpretation of the fully coupled behaviour of models (and climate). I think this study achieves presenting ERF estimates for the UKESM1 model in a nice, coherent, and attractive way. In general, the study is well written and attractive to read.

In addition, as UKESM1 contains a relatively extensive description of atmospheric chemistry, aerosols, and aerosol-cloud processes, the interactions between different species and processes are included. This makes this model an interesting tool to estimate the ERF of various short-lived emissions or longer-lived GHGs.

I think this work is valuable and of sufficient quality to be published in ACP. However, I think the manuscript could be considerably improved. I have listed several points on which I think the manuscript should be changed.

I have grouped my comments in three categories, i.e., main remarks, technical remarks, and additional detailed remarks.

The authors acknowledge Anonymous Reviewer #3 for their positive response to the manuscript and for the very detailed and comprehensive suggestions for improvements. Thank you! We aim to address them in our responses below and in the changes applied to the manuscript.

**Main remarks**

**1. TABLE 3**

Table 3 is very interesting and plays to my opinion a rather central role in this study. A lot of the values in the table are being referred to in the text. However, some parts of Table 3 are not discussed. This is the case for two large and two small groups of experiments : SO2-BC-OC-Aer, NOX-VOC-O3, LU and anthro. For these groups, only the NET ERF is discussed, but not the individual contributions. I assume this is related to the fact that for those 4 groups another split than the one in Eqs. (6)-(8) is followed, which is probably assumed to be more appropriate.

To make the study more consistent, I would at least mention that the non-discussed values exist in the Table 3, and are given for comparison/completeness/consistency. Explicitly presenting the numbers from the other approach followed for some experiments in a table would be interesting. Now these numbers are only mentioned in the text.

Response: Thank you for this constructive comment. Firstly, we would like to confirm that we have used Eqn. (8) for estimating all of the components of the anthropogenic ERFs for consistency. We state this in Section 4.1 (Line 353 of marked-up manuscript).

There are a lot of data values in Table 3 and the text draws out the most interesting and important points from these. Given the size of the paper, we do not think it would help to duplicate that information even further with additional tables or discussions. However, we do refer to Table 3 in the section on land use (Lines 1058-1061 of marked-up manuscript), the total anthropogenic ERF (Line 1128 of marked-up manuscript, and we also include a statement that the clear-sky and cloud radiative effect components can be found in Table 3 in the aerosol section (Lines 637-638 of markedup manuscript). And it is mentioned in the context of non-linearity of the ERFs for the ozone precursors (Lines 925-936 of the marked-up document).

**2. EXPLANATION OF RADIATIVE FORCING**

Although radiative forcing and effectve radiative forcings are very useful concepts, it is not always so easy to explain them. I think the authors have done a nice effort in trying to explain it. However, an extra effort should be done to make it more precise, fully coherent, and more illustrative. The final link why it is a useful concept should be elaborated more. I list here below also some specific places where I think the description should be improved.

- page 2, line 61-63 : Including ... . As a result ... : I do not think that "As a result" is appropriate, as I do not think there is a causal relationship.

Response: In response to Reviewer #1, we have altered that section as follows:

"Although the error uncertainty associated with an ERF tends to be larger than that of RF (or SARF), climate sensitivity parameters (i.e. the degree of warming per unit forcing) are less dependent on the forcing agent and it is more representative of the climate response than the traditional RF (Hansen et al., 2005; Forster et al., 2016). As a result, it is now the preferred metric of choice for ranking the drivers of climate change (Boucher et al., 2013; Forster et al., 2016)."

*This removes the incorrect causal link – See lines 158-162 of marked-up manuscript.*

- page 2, line 63-64 : RF is a framework : this is a rather vague explanation.

Response: The reference to "framework" has been removed and replaced with "concept", i.e., that forcing is the "rapid" radiative response to a change in activity (e.g., emissions change) which occurs before the longer-term climate response occurs.

See lines 74 and 95 of the marked-up manuscript

- page 4, line 115-117 : I think that the sentence on line 115-117 illustrates that ERF can be very different from IRF, however, it does not illustrate that it is better (as suggested in the former sentence). It might be an option to reverse the order of these sentences.

Response: Now done – see lines 153-162 of marked-up manuscript.

 page 4, line 113-115 : although it is correct what is written here, I think it should be explained better.

Response: We have now included a reference to the seminal work by Hansen et al. (2005), in which they found that the efficacies of climate forcings were less dependent on the forcing agent and closer to 1 for ERFs than for SARFs - See lines 156-158 of marked-up manuscript.

**3. FEEDBACKS, INTERACTIONS, NON-LINEARITY, INDIRECT EFFECTS**

I think that feedbacks, interactions and indirect effects are an important part of this paper. In the abstract, it is stated that "... by quantifying ..., it enables the role of various climate-chemistryaerosol-cloud feedbacks to be quantified." However, despite this sentence in the abstract, not all feedbacks have been quantified (it would be a very large task to quantify them all). The fact that they are active in UKESM1, and impact therefore ERF estimates, is already an important step. I think the text will benefit from more precisely describing what is quantified, and what one is not able to quantify.

Response: We agree that feedbacks, interactions and indirect effects are an important part of the paper and have a large impact on estimates of climate forcing. As the reviewer rightly points out, we

cannot quantify all of them and hence, "quantified" was perhaps a poor choice of word. We have now replaced this with "investigated".

I do not ask the authors to do additional analysis, or perform more comparisons with a model which contains less interactions (e.g., HadGEM-GC3.1). For quite some of the forcings, the authors did a very nice job in unraveling several contributions. However, it would be nice if the authors could be more careful and precise in some of the general expressions about feedbacks and interactions.

Response: In the Introduction, we state that "feedbacks are related to global mean temperature change and forcings are not (Sherwood et al., 2015)". However, in some cases, the term "feedback" was used where it was not associated with a global mean temperature change. In these cases, we have replaced "feedbacks" with "interactions", where the term "interaction" refers to a coupling (e.g., chemistry-aerosol-cloud coupling) in the ESM that results in an indirect contribution to the quantified ERF, and is considered part of the forcing. See examples on lines 20, 46, 336, 1157, 1172, and 1239 of the marked-up manuscript.`

Another example is the term "aerosol-mediated cloud feedback". This term has been replaced with "aerosol-mediated cloud adjustment" and is more consistent with the ERF definition. See, for example, lines 1038 and 1145 of the marked-up manuscript

I have in addition the following comments.

 I think the authors should explain what according to them is the difference between a feedback and an interaction.

Response: See response above.

 It is a pitty that the studies of Morgenstern et al. [2019, in preparation] on ozone and O'Connor et al. [2019, submitted] on methane have not been published yet.

Response: As indicated above in response to Reviewer #1, this is because these papers were part of a large cohort of papers submitted ahead of the paper submission deadline to be considered for inclusion in the 6th assessment report of the Intergovernmental Panel on Climate Change. Unfortunately, neither paper has been published yet but we chose to keep them as separate papers to avoid this manuscript becoming excessively long. However, we now include a longer paragraph on the ERF from piClim-HC (Section 4.2.3: Page 26-27 of the marked-up manuscript) and include some discussion on the comparison between the ERF estimated from piClim-CH4 here with estimates from the AerChemMIP multi-model ensemble (Thornhill et al. 2019) – See lines 580-588 of marked-up manuscript.

3) Some of the formulations, referring to feedbacks and interactions, are often rather vague (less firm than the abstract suggests). Examples of these are :

 page 15, line 354 : ... "likely" ... the effect of adjustments ... including O3 depletion and fast cloud adjustments

Response: The discussion has now been removed. The updated simulations no longer show the big effect discussed here previously. The leading effect now clearly is the direct RF due to the GHG property of N2O.

page 15, line 359 : which "might" be due to an aerosol effect

Response: As for previous response directly above

4) the abstract stresses well the non-additivity (or non-linearity) of several ERF estimates. However, the study sometimes remains vague in reasons for it, and just uses expressions like "the non-linearity between the GHG", or "the internal mixing of aerosols". Other examples of not so clear usage of the concepts :

Response: As indicated in a previous response, we had to re-run all the experiments due to a bug in the seasonal cycle of the prescribed leaf area index. The non-linearity previously seen in the GHG ERFs is no longer evident and any reference to it has been removed (e.g. Lines 613-618 of marked-up manuscript).

In the case of aerosols, Section 4.3 includes a few sentences that explain why aerosol forcings do not add up linearly:

"The main reasons for this lack of linearity are: (i) Cloud droplet numbers do not increase linearly ...". (ii) The absorption of upwelling shortwave radiation...

However, it didn't explain why internal mixing contributes to non-linearity. This additional sentence has been added as a point (iii) into that paragraph:

"Internal mixing creates an interdependency between different aerosol sources, meaning that the aerosol size distributions, optical scattering efficiency and hygroscopicity evolve differently depending on the absolute and relative abundance of different mass components (sulphate, OC, BC and sea salt mass) and differing rates of new particle production via primary emission or nucleation. "– See lines 713-716 of marked-up manuscript

- page 2, line 36 : due to the inclusion of non-linear feedbacks and ES interactions : vague

Response: Now removed – See Lines 42-44 of marked-up manuscript

- page 2, line 38 : By including feedbacks between GHGs (isn't it interactions?)

Response: Changed to "interactions" - See line 46 of marked-up manuscript

- page 2, line 38-39 : some of which act non-linearly

Response: Now removed – See Line 47 of marked-up manuscript

**4. UNCERTAINTIES**

The ERF estimates in the manuscript are accompanied by an uncertainty range in large parts of the main text and in Table 3, but not in the abstract, the conclusions, or some parts of the main text. This should be made consistent.

Response: Thank you for this comment. We have now added the model estimates for the ERF uncertainties in the Abstract (See Pages 1-2 of marked-up manuscript) and the conclusions (See Pages 54-57 of marked-up manuscript) and in other relevant sections, e.g., CO2 – See lines 440-454, N2O – See lines 473-485, ODSs – See lines 529-541, CH4 – See lines 562-578, GHG – See lines 606-610, Aerosol – See lines 649, 658-659, 670, 673, and 684. For ozone precursors – See line 750, NTCFs – See lines 986-1000, and land use – See lines 1058-1061, and total anthropogenic – See line 1126 of the marked-up manuscript.

Also, one should maybe say something about the real estimate of the uncertainty. There is something on aerosol ERF uncertainty on page 20, line 453-455, and maybe this can be expanded. In general, UKESM1 shows uncertainties of around 0.02 to 0.04 W/m2 on global 30-year mean

averages of ERF – it is the uncertainty on the ERF in the model. However, if one wants to see these numbers as best guesses for the ERF as experienced by the Earth, then the uncertainty is probably much larger. Emission uncertainty, lack of process understanding, too low spatial resolution, biases in the mean state of UKESM1 and others factors might contribute to the uncertainty. It would be interesting if the authors could also shed some light on this.

Response: Thank you for this comment. You are indeed correct that the quoted uncertainties are solely based on the signal to noise of the modelled ERF and do not represent the true uncertainty. As you say, the true uncertainty will be larger due to a number of factors, e.g., uncertainties in emissions, systematic biases and/or lack of sensitivity to emissions due to representation of chemistry, pre-industrial background aerosol state, missing and/or unresolved processes, etc..

Although a comprehensive assessment of the true uncertainty in ERF estimates is beyond the scope of the paper, it is indeed an interesting question and we acknowledge this in Section 4.1 (Lines 354-359 of marked-up manuscript).

**5. VALIDATION OF THE MODEL**

There is very little information on the validation of the model, except via references to Mulcahy et al. [2018, 2019] and Archibald et al. [2019]. There are a few paragraphs (e.g., page 20, line 454-457 : aerosol comparison; page 21, line 491-492 : AOD comparison), mentioning some comparisons with observations. We do not ask for a detailed comparison, but some qualitative main findings on the performance of the model might be mentioned, both related to distributions of forcing agents (aerosols, ozone, ...) as to the general behaviour of the model (mean model state).

Response: In relation to aerosols, we do indeed make use of the findings from Johnson et al. (2019) and Mulcahy et al. (2020) in the discussion of the aerosol forcing. We also now include more discussion on the ERF from ODSs (in response to a reviewer comment) and comment on how the performance of stratospheric ozone in UKESM1 may be impacting on that ERF – this is further discussion in Morgenstern et al. (2019) and Skeie et al. (2020). We also include some discussion on the impact of strong O3 depletion on the tropospheric O3 RF. And how the performance of the model may be affecting estimates of the CO2 ERF and the land use ERF is also discussed.

See Lines 549-560, 815-819, 456-472, 1068-1071 of the marked-up manuscript.

**Technical remarks**

**1. POSITIVE NUMBERS**

Some positive numbers have a "+", some not. This should be made consistent.

Response: All those numbers with a "+" have had it removed for consistency – See, for example, Table 3 (Page 14-16) and lines 362, 366, 367, 370, and 405, etc.

**2. REFERENCES TO THE PHYSICAL MODEL**

It might be relevant to explain or highlight the relevant differences between UKESM1 and HadGEM3-GC3.1 better (both models are very close and have both participated in CMIP6 I assume). In the text quite often comparisons with HadGEM3-GC3.1 are made, and it would illustrate the role of having more/less interactions in a coupled model. I would think that possible differences are related to fixed ozone, fixed oxidants for secondary aerosol formation, fixed methane profile, fixed CO2 profile, ...

Response: A table describing the main differences between the atmosphere-only configurations of HadGEM3-GC3.1 (called HadGEM3-GA7.1) and UKESM1 has been added to the manuscript as an appendix – See page 58 of the marked-up manuscript.

**3. REFERENCE TO FORSTER ET AL. [2016] FOR STANDARD ERROR**

The way to calculate the standard error is mentioned three times. It should be mentioned when the uncertainty is met for the first time (both in the text and in the tables). The standard error and its reference are now mentioned in :

- page 12, line 272 (Table 3).

Response: Now removed in multiple places and it is now only mentioned once in Section 4.1 (See line 354-355 in the *marked-up document*

 page 18, line 402 (related to CH4) : where the 0.04 W m-2 is the standard error following Forster et al. (2016) : should be mentioned once (the first time).

Response: Now removed (line 564 of marked-up document)

- page 31, line 760-761.

Response: Now removed (line 987 of marked-up document)

**4. MULTI-ANNUAL MEAN / ANNUAL MEAN / MEAN**

It is mentioned in the beginning of the manuscript that almost all values and maps will be 30-year averages. Some sentences and figure captions in the text treating those values or maps once more mention explicitly that the averages are "multi-annual", whereas other figures captions or sentences do not. Please describe it in a consistent ways. An option might be to describe it clearly at some point in the, and say that it is valid for all the remaining text. Examples of differences in the description :

- page 18, line 395 : multi-annual mean (while page 14, line 323 : global mean)

- page 24, line 562 : The multi-annual mean ...

- page 25, line 599 : on a global annual mean basis

page 29, line 699 : Global distributions of the multi-annual distributions

page 33, line 795 : multi-annual global mean

Response: The manuscript already states the following: "Therefore, all simulations were 45 years in length. Using the latter 30 years of the paired simulations, the ERFs from the PI-to-PD perturbations were diagnosed as the time-mean global-mean difference in the TOA net radiative fluxes. " in Section 3 (Lines 252-254 of manuscript).

We removed all reference to "multi-annual mean", "global annual mean basis", "multi-annual distributions", "multi-annual global mean" etc.., as suggested. See examples at lines 386, 417, 529, 690, 759, 763, 806, 919, 1047, 1174, 1184, and 1191 of marked-up document.

**5. WRITING OF LAND USE**

Both "land-use" and "land use" are used in the text.

*Response: Changed all occurrences of "land-use" to "land use" for consistency (e.g. Section 4.5.2, lines 1057, 1062, 1068, and 1070, as examples, in the marked-up document).*

**6. CONSISTENCY**

- page 10, line 261-265 : why is a difference of 0.2 W/m2 for GHG called "consistent", but for the aerosol mentioned "less then the estimate from HadGEM3-GC3.1"?

Response: This has now been altered. With the updated experiments, we now state that the aerosol forcing is consistent with HadGEM3-GC3.1 but the GHG ERF is lower i.e.,  $2.92 \pm 0.04$  cf. 3.09 W m-2. See lines 361--364 of marked-up manuscript.

- page 22, line 502 : -0.12 W m-2 (-0.4 to +0.1), whereas on page 10, line 267 : 2.3 (+1.7 to +3.0) [5-95%] W m-2 : the position of Wm-2 is different.

Response: These have been made consistent with each other. See lines 368-370 of revised manuscript

 page 22, line 515-519 : only here "(i)" and "(ii)" are used. I would not use it just for this single occasion

Response: In response to your previous comment, we have now added a third point in relation to internal mixtures. As a result, we have opted to retain the numbering (i) (ii), and (iii) – See lines 705-712 of marked-up manuscript.

- page 23, line 536 : only here "&" is used. I would not use it just for this single occasion.

Response: Replaced with "and" - see line 734 of marked-up manuscript

7. NAMING of SWcs, Lwcs, SWcre and LWcre COMPONENT OF ERF

There are different abbreviations used to express the same physical quantity. One should make these coherent. An example of this is the LW clear-sky component of ERF, which appears in different ways in the text, tables, and equations :

- page 11, Table 3 : LWcs

- page 10, Eq. 8 : ERFcs

- in the text : a CS LW component

- page 16, Fig. 4 : (a) clear sky SW, (c) clear sky LW, and (f) net (SW+LW) CRE : it sounds as something is missing after SW and LW

- page 31, line 762-770 : NETcs, LWcs, SWcre

- page 32, Fig. 12, caption : SWcs, LWcs, SWcre, and LWcre

Response: In all cases, we have adopted consistent names for the different components of the ERF: SWcs', LWcs', SWcre', LWcre', NETcs', and NETcre', as given in Eqn. (8). These have been used throughout the manuscript, e.g., Lines 449 onwards for CO2, Lines 475 onwards for N2O, Lines 527-531 onwards for ODSs, Lines 564 onwards for CH4, etc. Figure captions have also been updated, e.g., for Figures 4, 5, 6, etc.

8. STATISTICAL SIGNIFICANCE OF DIFFERENCE BETWEEN IDENTICAL SIMULATIONS

Several experiments have been performed on different machines. Related to this I have the following remarks :

**- Table 4 : I assume there are three piClim-SO2 experiments, so one could show the difference R1-R2 but also R1-R3.**

Response: As detailed in Table 2 (Page 12 of the marked-up manuscript), there are only two piClim-SO2 experiments, realisation 1 (R1) was performed on the NIWA XC50 and realisation 2 (R2) was performed on the Met Office Cray XC40. To make this clearer, we have changed the title of the last column in Table 2 from "Realisation" to "Realisation ID" and placed "R" in front of each number to indicate more clearly that there are only 2 realisations of the piClim-SO2 experiment i.e. R1 and R2. The Realisation ID is unique for each realisation of a particular simulation and forms part of the CMIP6 metadata.

**- Table 4 : Some differences are a bit large compared to their uncertainty for SO2 : NET ERF, SW CRE, NET CRE. Is there an explanation for this?**

Response: The non-linearity of the equations being solved in Earth System Models makes them sensitive to the propagation of small perturbations making the models sensitive to change in HPC platform. As the model integrates (through time), these perturbations grow randomly changing the climate state of the experiment (e.g. cloud development, precipitation, temperature, etc.). The results in Table 4 show the impact in terms of ERF from the change in HPC platform and their associated standard error (as a measure of uncertainty). Even though some values in the original manuscript appeared large (as pointed out by the reviewer), the Kolmogorov-Smirnov statistical test applied showed that these differences in the time series were not statistically significant in terms of their distribution (Please refer to lines 424-432 of the marked-up manuscript). The updated results in Table 4 (Page 21 of the marked-up manuscript) from correcting the seasonal cycle of leaf area index no longer appear to be large and any difference in ERF is within the uncertainty.

**- Table 4 : Why is everything 0 for the NTCF experiment?**

Response: The 2 piClim-NTCF realisations were performed on similar machines, a Cray XC40 using similar compilers: Cray compiling environment 8.3.7 for R1 and 8.3.4 for R2. Because of this, the results of experiments produced on these machines were similar and no differences were found between these realisation pairs. The updated results in Table 4 (Page 21 of the marked-up manuscript) from correcting the seasonal cycle of leaf area index no longer appear as zero, and are now similar to what was found for other realisations.

**9. NOx EMISSIONS AND SO2**

Figure 11 : is the impact of NOx emissions on AOD not mainly in regions with SO2 emissions from volcanoes? Indonesia, west coast of South America, Etna region, existing ship-lanes in 1850 over North Atlantic? Might probably NOx have then also a large effect if it is co-emitted with anthropogenic SO2?

Response: This is an interesting observation, and it likely is the case that coupling between NOx and SO2 emissions drives stronger AOD changes. The UKESM1 model includes climatological SO2 emissions from non-explosive volcanic degassing (following Dentener et al., 2006; https://www.atmos-chem-phys.net/6/4321/2006/acp-6-4321-2006.pdf). However, while the AOD changes are largest in regions where these SO2 emissions occur, they are not truly co-located – NOx emissions occur at the surface while SO2 emissions are released at higher levels. We feel that exploring this coupling is beyond the scope of this study, and so do not specifically refer to this point in the manuscript.

**10. FORMULAS**

The "." or "," at the end of formulas can be closer to the actual equations (currently there is a 1.5 cm large gap). There should not always be a ":" at the end of the text just before a formula. This depends on the context. One can also have a ",", or nothing.

Response: Done - see lines 68, 146, 324, 333, 341, and 343 marked-up manuscript

**11. REFERENCES TO SECTIONS**

The references to specific sections are not always correct. Please check and correct them. A list of incorrect section references :

- page 8, line 192 : Sect 3.1 -> 4.1

- page 12, line 287 : Sect. 3.2 -> Sect 4.2

- page 13, line 291 : Sect. 3.3 -> Sect 4.3

- page 13, line 296 : Sect. 3.4 -> Sect4.4

- page 14 ,line 307 : Sect. 2 -> Sect 3.

- page 25, line 602 : Sect. 3.2.3 -> Sect 4.2.3

- page 29, line 705 : Sect. 3.4.2 -> Sect 4.4.2

*Response: Thank you for spotting these errors. These have been corrected. See lines 255, 395, 401, 407, 424, 815, and 926 of marked-up manuscript.*

12. USE OF e.g.

There should be a "," before and after "e.g.".

Response: Now corrected – See lines 72, 78, 86, 92, and 98, etc. We also corrected similar errors with "i.e." – See lines 117, 157, 211, 236, 247, etc.

**13. REFERENCE TO EQUATIONS**

The way one refers to equations is not homogeneous in the text (sometimes capital letters, small letters, abbreviations, or no abbreviations). Please make it coherent. A list of the differences found in the text :

- page 4, line 111 : Eqn. 1

- page 10, line 251-252 : in equation (5)

- page 15, line 348 : following the equations (6)-(8) described above

- page 19, line 425 : using Equations (6)-(8)

- page 27, line 643 : From Eqns. (10) and (11)

*Response: All references to equations are now consistent – See lines 347, 353, 376, 383, 475, etc. of marked-up manuscript.*

**14. USE OF THE WORD "FORCINGS"**

I have the impression that the word "forcing" (outside the context of radiative forcing and effective radiative forcing) is possibly not used in a completely coherent way. I have listed below a few

locations where it is used, and it seems not to always have the same meaning. I think the text should be more careful and precise in how and where it is used.

page 1, line 16 : climate forcings

Response: Changed to "effective radiative forcings" – See line 16 of the marked-up manuscript

page 1, line 31 : a positive forcing due to ozone

*Response: Changed to "a positive radiative forcing due to increases in O3" – See line 35 of the marked-up manuscript*

- page 3, line 65 : various mechanisms, both anthropogenic and natural. Here I would have used the word forcing and not "mechanism".

Response: Changed – See line 77

- page 3, line 65 : ... use of RF ... it is often used inconsistently ... : is RF meant in the first part of the sentence, because that is well-defined (page 2, line 54-56)?

Response: We agree that RF is well defined but here, we are saying that it is often calculated inconsistently, e.g., GHGs vs ozone. There are different methods for calculating RF, e.g., line-by-line radiative transfer calculations, formulae, GCM radiative transfer calculations, Gregory regression from coupled models, timeslice atmos-only experiments, different inputs (e.g., observed concentrations, modelled concentrations) etc.

- page 3, line 67-69 : This sentence mentions three things, and it is not clear whether the reader should see a causal relationship between "the inconsistent calculation of forcing between drivers", and "the large differences in forcing between CMIP5 models". For the last part of the sentence, it is not clear whether the authors mean that models are very different or that the methods of estimating forcing can be very different.

Response: There is no causality intended. We have now added that the large differences in forcing between CMIP5 models for CO2 can be due to either calculation method and/or model diversity – See line 81 of the marked-up manuscript.

- page 3, line 82 : the resulting RF : is not the same as meant by the definition on page 2 line 54-56.

*Response: Agreed, changed as follows: "*The simulations used the corresponding sea surface temperatures (SSTs) and sea ice (SI) conditions for the time periods of interest (PI and PD), therefore allowing some climate response and feedbacks at the PD, implying that the resulting estimate is not consistent with the RF definition; it does not fit into the simple forcing-feedback concept, whereby feedbacks are related to global mean temperature change and forcings are not (Sherwood et al., 2015). " – See lines 92-96 of the marked-up manuscript

- page 7, line 171 : in which SSTs and SI and all forcings

Response: Changed "forcings" to "boundary conditions" - See line 235

- page 33, line 782 : between the aerosol and O3 forcings

Response: Changed to "As a result, this study also attempts to estimate the effects of the non-linear interactions between chemistry and aerosols on the combined aerosol and  $O_3$  precursor ERF" – See lines 1017-1018

- page 35, line 840-845 : "forcing" used four times. Also "anthropogenic" forcings and "natural" forcings, whereas it is not so clear what to understand here under natural forcing. In the text there is a reference "As summarized above", but I think there was not a large focus on "natural forcings".

Response: Changed to "As noted above, historical climate change has been driven by a wide range of anthropogenic activities that act together, alongside natural changes, to perturb the Earth's radiation balance. The total anthropogenic ERF is, therefore, a key metric in understanding observed and modelled changes in the climate system since the PI era. These various anthropogenic drivers are not necessarily independent of each other and it is therefore worthwhile calculating the total anthropogenic ERF from a separate timeslice simulation including all PD perturbations together (piClim-anthro; Pincus et al., 2016). " – See lines 1113 onwards of the marked-up manuscript.

**15. THE USE OF ABBREVIATIONS**

The definition of an abbreviation is often given several times, whereas that should be

limited to :

- defined once in the abstract,

- defined once in the conclusions,

defined once in the figure caption, and

- defined once in the main text (but not more).

Once a definition is given in the main text, it should not be defined again.

Response: Done for Abstract – See Page 1-2 of the marked-up manuscript; Done for Conclusions – See Pages 54-56 of the marked-up manuscript; We could find no occurrence of where an abbreviation was defined twice within a figure caption – no action taken. Done for main text – See lines 64, 141, 143, 148, 149, 190, 218, 220, 236, 241, 288, 289, 298, 339, and 400, for examples.

16. PD AND PI ARE ADJECTIVES

- page 1, line 21 : at the PD, or relative to the pre-industrial (PI) : PD and PI are both adjectives, and not substantives. So they cannot be used on their own. E.g., on page

1, line 15 it is used correctly as an adjective : "... a wide range of present-day (PD) anthropogenic climate forcings ...".

Response: Any reference to PI and PD have been corrected. See Line 375, 383, 388, 418 etc. in the marked-up manuscript.

**17. PD-PI DIFFERENCE**

The way the ERF is described as a difference between TOA fluxes in PD and PI times, is not coherent in the text. Sometimes one finds a "PD forcing" type of expression, sometimes a "PD to PI forcing" type of expression, and sometimes neither PD nor PI are mentioned. Some examples of the varying usage are given below. It would be nice if the description could be more coherent. An option would be to write in the text that all values shown in text from that point onwards are always PD-PI differences, and then PD and PI should not be repeated every time.

- page 3, line 71 : between the pre-industrial (PI) and the present day (PD)

- page 3, line 76 : at the PD relative to the PI

- page 5, line 130 : the PI to PD effective radiative forcings (ERFs)

 page 5, line 132 and 133 : PD forcings will be quantified relative to the PI [in addition is "the" strange before PI which is just an adjective]

- page 5, line 139 : PD anthropogenic forcings relative to PI

- page 6, line 157 : To calculate the pre-industrial to present-day effective radiative forcings ... due to a PI-to-PD perturbation

- page 10, line 260 : the global mean PD ERF [here without any reference to PI]

- page 12, line 271 : PD effective radiative forcings (ERFs) of climate relative to PI

- page 12, Fig. 2 : y axis PI-to-PD ERF

- page 12, line 280-281 : PD ERFs from changes in ... since PI

- page 12, line 281 : PD ERF from GHG [but no reference to PI]

- page 13, line 303-304 : effective radiative forcing (ERF) relative to the pre-industrial

- page 14, line 319 : cloud radiative effect (CRE) at the present day [no reference to PI]

- page 15, line 338 : this forcing is small for the PD [no reference to PI]

- page 15, line 347 : due to changes in N2O from PI to PD

- page 16, line 370 : for the present relative to the pre-industrial

- page 18, line 400 : resulting in a PD ERF [no reference to PI]

- page 19, line 423 : of the PD methane ERF relative to PI
- page 19, line 430 : This PD GHG ERF
- page 20, line 462-463 : of PI to PD aerosol ERF

- page 21, line 472 (Fig. 7) : Aerosol ERF at TOA ... [no PD or PI mentioned]

- page 22, line 501 : The PD OC ERF relative to PI

- page 23, line 551 : The global mean ERF from ... [no PD or PI mentioned]

- page 25, line 591 : the PI-to-PD change in tropospheric O3
- page 27, line 651-652 : the tropospheric O3 RF between PI and PD
- page 28, line 679 : and the PD aerosol ERF
- page 30, line 723 : ERF from PI-to-PD changes
- page 31, line 744 : the PD ERF from piClim-O3

- page 33, line 795 : of changes (PD-PI) in

Response: Thank you for pointing this out. In Section 4, we have the following statement: "Therefore, all simulations were 45 years in length. Using the latter 30 years of the paired simulations, the ERFs were diagnosed as the time-mean global-mean PD-PI difference in the TOA net radiative fluxes.

Details on how the ERF was further decomposed can be found in Sect. 4.1." – See lines 252-254 of marked-up manuscript.

*We also write "*The aim of the current study is to quantify PD (Year 2014) ERFs from anthropogenic drivers of climate change with an atmosphere-only configuration of an ESM. Using the experimental protocol recommended for the Radiative Forcing Model Intercomparison Project (RFMIP; Pincus et al., 2016), PD ERFs will be quantified relative to the PI period from PD-PI changes in emissions, concentrations, and/or land use due to anthropogenic activities." – See lines 184-188 of the marked-up manuscript.

In response to the lack of consistency, we have therefore removed most references to PI and PD when referring to the quantified ERFs in the subsequent sections – See lines 25, 40, 89, 194, 361, etc. of the marked-up manuscript for examples.

**18. TIMESLICE EXPERIMENTS**

Maybe it would be nice to better define what a timeslice experiment is. I would possibly describe it as an experiment with fixed boundary conditions (possibly having seasonal cycles), which one runs for several years to reduce the noise-to-signal ratio (the noise is caused by inter-annual variability). The longer one runs, the better estimate for the mean one can obtain. The locations were it is used are :

- page 6, line 158 : maybe explain timeslice here (forcings are kept constant)

- page 7, line 171 : time slice

- page 15, line 604 : timeslices : in ACCMIP they were different from here ...

Response: In Section 3, we introduce the term "timeslice" and we have added further details as suggested. For example, we now state that "SSTs, SI and all other boundary conditions were fixed at year-1850 levels" for the pre-industrial control simulation – See lines 234 onwards of marked-up manuscript. We also note that "Running for 30 years when the model has reached steady state reduces the uncertainty associated with meteorological variability (e.g., Shindell et al., 2013a) and improves the estimate of the ERF. " – See lines 254 onwards of marked-up manuscript.

**19. FIGURE 4**

Fig. 4, panel (a) : Why is there only some spatial variability north of 30N?

*Response: This figure has now been updated following the leaf area index bugfix and the new figure no longer shows this feature.*

**20. CONSISTENCY OF THE FIGURES**

The figures and their captions should be more coherent throughout the manuscript. Below I list some places where improvements should be made.

- page 16, Fig. 4 : at the top of the atmosphere (TOA) : this is however not mentioned in Fig. 3.

 page 17, Fig. 5 : why mentioning "annual mean" or "multi-annual mean", whereas it is not mentioned in Fig. 4.

- page 19, Fig. 6 : it would be nice to have the global mean values given in the figure heading.

- page 19, Fig. 6 : mentions "according to Ghan (2013)". Why not in Fig. 4?

- units in figures : Fig 7 uses "(W/m2)", whereas other figures use "/ W m-2".

 page 24, Fig. 9 : of the "multi-annual" distributions, whereas other figures do not mention "multannual".

- page 26, Fig. 10 : it would be nice to add the global mean RF.

- page 32, Fig. 12 : it would be nice to add the global mean values

Response: Figures and Figure captions have been made more consistent, as indicated above to a previous comment. Figures also include global mean values, where appropriate (e.g., Figure 3, Figure 4, Figure 5, Figure 6, Figure 10, Figure 12). We have also removed all references to "multi-annual annual mean", "multi-annual mean", etc., in response to a previous comment. We have also made the units in Figure 7 more consistent (i.e., W m-2) with the other figures.

**Additional detailed remarks**

**ABSTRACT :**

- page 1, line 23 and 28 : "larger than the sum of the individual GHG ERFs" and "less than the sum of the individual speciated aerosol ERFs" : I don't know whether these aspects should be mentioned in the abstract.

Response: Now removed - See line 26

- page 1, line 15 : "In this paper" : I would not mention "In this paper" in the abstract.

Response: Removed - See line 15

page 1, line 18-19 : by quantifying ..., it enables ... : this sentence seems not completely coherent.
There is also twice "quantify" in the same sentence (in addition there was already "quantify" on line 15).

Response: Replaced with "investigated" – See line 20

- page 1, line 21-22 : I would put the numbers at the end of the sentence, and "carbon dioxide, nitrous oxide, ..." at the beginning of the sentence. Now, one first reads numbers, but one does not know what their meaning is.

Response: Good idea – Done! Line 22

- page 1, line 19 : by this sentence, one suggests that climate feedbacks can be quantified by fixed-SST simulations. However, some of them are strongly suppressed in fixed-SST simulations.

Response: We change "feedbacks" to "interactions" and say that they can be investigated.

- page 1, line 25-26 : is the "BC absorption" not part of the "instantaneous forcing from aerosolradiation interactions"?

Response: Yes, it is. In Section 4.4, we state that "the IRF is rather small and negative ( $-0.15 \pm 0.01 W m^{-2}$ ) (Fig. 7a) due to scattering by sulphate and OC that is partially offset by absorption from BC." For the abstract, we modify the sentence to make it clearer: A relatively strong negative forcing from aerosol-cloud interactions and a small negative instantaneous forcing from aerosol-radiation interactions from sulphate and organic carbon are partially offset by a substantial forcing from black carbon absorption."

- page 1, line 27 : mean -> "imply" or "cause".

Response; Done - See line 31

**INTRODUCTION**

- page 2, line 42-45 : necessary ... detailed ... all aspects : this is probably exaggerated. I suggest to formulate it differently.

*Response: Done – See line 52*

- page 2, line 42 : attribute it and its impacts : it (refers to climate change) and its impacts. It is not clear whether the authors mean something different with "climate change" and "impacts".

Response: Modified as suggested – See line 51

- page 2, line 44 : climate response and its impacts : (same comment as above).

Response: Modified as suggested – See line 53

 page 2, line 44 : a key mechanism : I would not call CMIP6 a "mechanism". Also "key" is possibly exaggerated - other initiatives (if CMIP would not have existed) might have also had good outcome.

Response: Agreed, modified accordingly. - See line 54

- page 2, line 45-46 : which designs and distributes data : "designs data" sounds strange. Possibly one could say that "experiments are designed".

Response: Done – See line 55

- page 2, line 47 : "these important climate science questions" : it is not clear which questions one refers to.

Response: Removed - See line 57

page 2, line 50 : quantifying changes to the Earth's radiation budget, often termed radiative forcing
maybe radiative forcing needs a bit more explanation.

Response: Added an explanatory sentence. - See lines 60-61

- page 2, line 64 : It's been -> It has been.

Response: Corrected – See line 76

- page 3, line 65 : the strength of the various mechanisms : of various mechanisms

Response: Corrected – See line 77

 page 3, line 65 : mechanisms, both anthropogenic and natural. I do not think that "mechanism" is the most appropriate word to be used here, especially in the context of "anthropogenic" and "natural".

Response: Changed to "forcings" – See line 77

- page 3, line 70-71 : is typically based on .. and using -> uses.

Response: Corrected – See line 83

- page 3, line 72 : I suggest to put "e.g." before "based on Myhre et al. (1998) and Ramaswamy et al. (2001)", as there might exist other expressions.

*Response: Done – See response above*

- page 3, line 75 : Skeie et al. [2011] : I think that study is not so much about observational-based estimates of forcing. Is this paper very relevant in the discussion of forcing strength of GHGs?

Response: Removed – See line 86

- page 3, line 81 : including -> therefore allowing.

Response: Done – See line 94

- page 3, line 82 : meaning -> implying.

Response: Done – See line 94

- page 3, line 82 : doesn't -> does not.

Response: Done – See line 95

- page 3, line 84-85 : of a robust ... constraint -> of robust ... constraints.

Response: Done – See line 97

- page 3, line 84 and 89 : twice "additional uncertainties".

Response: Removed duplicate - see line 102

- page 3, line 85-86 : across multi-model ensemble : is that really what the authors want to stress? Might "across models" be sufficient?

Response: Changed as suggested – See line 99

- page 3, line 87 : chemistry models : does one mean CTMs or CCMs?

Response: Changed to "chemistry-climate models" – See line 100

- page 3, line 91-93 : three times the word "uncertainty" in one sentence. Maybe it can be reduced to two.

Response: Done! See line 105

- page 3, line 91-93 : it looks like aerosols get only very limited text attributed.

*Response: A section on aerosol forcing has now been added to the Introduction – See lines 108 onwards of marked-up manuscript*

- page 4, line 96 : is "schematic" the correct wording?

Response: Changed to "Vertical profiles of temperature" - See line 135

- page 4, line 101 : Although -> Because/As.

Response: Done! See line 140

 page 4, line 102 : maybe "also" can be skipped. I don't know if it really reflects well the meaning of the sentence.
Response: Done - See line 141

- page 4, line 109 : andAi -> and Ai (blanco needed).

Response: Done – See line 148

- page 4, line 110 : or over the land : vague.

Response: Changed to "land surface" and added an extra example of a land surface adjustment, i.e., land surface temperature – See line 151

 page 4, line 112-113 : but global mean surface temperatures or global ocean conditions remain unchanged. However, in reality with fixed-SST simulations, land surface temperature (and thus global mean surface temperatures) can still change a bit.

*Response:* Correct – have now removed any reference to global mean surface temperatures – See line 152

- page 4, line 113 : error -> uncertainty (I assume the authors mean "uncertainty").

Response: Changed – See line 159

- page 5, line 129 : I would skip "including" because the list mentioned seems rather complete.

Response: Removed – See line 180

- page 5, line 130 : forcings from anthropogenic drivers -> forcing from anthropogenic drivers.

Response: Done! See line 184

- page 5, line 131 : with a fully coupled : the "with" gives the impression that one uses the model here in its "fully-coupled" configuration. But here it is not used in its fully-coupled configuration.

*Response: Corrected – changed to "an atmosphere-only configuration of an Earth System Model" – See line 185*

**SECTION 2**

- page 5, line 142 : "is" the atmospheric and land components -> consists of.

Response: Done – See line 197

 page 5, line 150 : "is determined by prescribed oxidant fields" : maybe describe differently, as oxidants are not the only determining factor.

Response: Now changed – See line 205

SECTION 3

- page 7, table 1 : piClim-VOC : add that also CO is perturbed.

Response: Done – See Table 1 (page 8-9)

- page 7, line 166-167 : twice ERF : I think mentioning "fixed-SST" is enough to describe the experiments. Obtaining the ERF is the result of such an experiment.

Response: Caption now changed – See line 227

- page 7, line 171 : Effectively, this involves ... : it is a bit a strange way to mention that also a reference simulation is needed.

*Response: I would respectfully argue that it is important to explain the experiment setup of piClimcontrol, as this underpins all of the other perturbation experiments.*

- page 7, line 171 : SSTs and SI and all forcings : one should not have two "and"s in a row.

Response: Corrected – See line 234

- page 7, line 171 : the abbreviations SST and SI have not been defined.

Response: They were defined at the very beginning of Section 3 – See line 93

 page 7, line 173-176 : One uses two different expressions, i.e., "monthly time-varying climatologies derived from 30 years of output" and "30-year monthly mean climatologies", to describe the same thing.

Response: Dropped the "monthly time-varying climatologies" - See line 235

- page 7, line 177-184 : is it not in agreement with RFMIP and AerChemMIP?

*Response: Interactive vegetation was specified as part of the RFMIP protocol. However, I could find no discussion about vegetation in the AerChemMIP protocol.*

- page 8, line 194 : emissions and/or GHG concentrations : I think this can just be "and".

Response: Now removed

- page 8, line 199 : fixed SST ERF experiments : I think fixed-SST is enough to describe the experiments.

Response: Now removed - line 258

 page 8, line 200 : ammonium nitrate : but other forms of nitrate are probably also not present (e.g., nitrate on dust and seasalt).

Response: Replaced with "nitrate aerosol" - See line 291

- page 8, line 208 : fixed SST timeslice ERF experiments : I think fixed-SST is enough to characterize the experiments.

Response: Replaced with "fSST" which has been previously defined - See line 298

- page 8, line 210 : This makes ... to changes in platform that cannot guarantee bitreproducible results : this can probably be expressed more precisely.

Response: Now changed - See lines 300 onwards

 page 8, line 211-213 : was scientifically consistent with each other : one should be more clear in what is meant by "scientifically consistent".

Response: Now edited – See line 3030

- page 8, line 220 : Further to this -> in addition to this.

Response: Done – See line 311

**SECTION 4**

 page 9, line 234 : Cloud-Radiative Effect (CRE): in the rest of the text, no capital letters are used when defining an abbreviation.

Respone: Done – See line 329

 page 10, line 240 : either ... and/or : I do not think that it is common to combine "either" with "and/or".

*Response: Changed sentence to: "However, many of the experiments in this study either directly perturb aerosol emissions or indirectly alter aerosol concentrations via chemical and dynamical feedbacks." – See line 355*

- page 10, line 241 : what is meant by "dynamical feedbacks" : changing meteorology which changes lifetime and thus burden of aerosols?

Response: Yes

- page 10, line 251-252 : and any non-aerosol changes in CS flux : is, e.g., the deposition on snow of BC included in this term?

Response: In UKESM1, deposition of BC acts as a sink for BC and it does not influence the radiation budget once deposited. The non-aerosol changes in CS flux is referring to absorption by gas-phase constituents, e.g., ozone. We have now included this example.

- page 10, line 252 : "The effective radiative forcing (ERF), clear-sky CS), and cloud radiative (CRE) contributions" : as I assume that "contributions" also relates to "clear sky", I would write : "The effective radiative forcing (ERF), and its clear-sky CS) and cloud radiative (CRE) contributions".

Response: Corrected as suggested – See line 352

- page 10, line 257-258 : following equations (6) to (8) : maybe only (8)? (As that is the final split which is presented in Table 3).

Response: Agreed. Now changed to just say Eqn. (8). - See line 353

- page 10, line 262 : HadGEM3 GC3.1 -> HadGEM3-GC3.1

*Response: This had been previously removed due to the updated results from the simulations with the LAI bugfix included. See line 368*

 page 12, line 271 : effective radiative forcings (ERFs) of climate : I do not think "of climate" is needed here.

Response: Removed, as suggested. See line 375

- page 12, line 273 : use Realisations 2. -> use realisation 2.

Response: Corrected – See line 377

- page 12, Fig. 2, caption : "diagnosed from paired fixed SST timeslice simulations with an atmosphere-only configuration of UKESM1". It is not clear why the fact that paired simulations are needed to estimate ERFs is mentioned here. It is, e.g., not mentioned in Table 3 (although also paired simulations are the bases for the results of Table 3).

Response: Now removed from Figure 2 caption – See line 382.

- page 12, line 283-284 : ERF from the piClim-HC perturbation experiment ... positive forcing from the other LLGHGs : in the first part of the sentence one talks about the ERF from experiments, and in the second part about the ERF of physical things (in this case LLGHGs). The sentence should be improved.

*Response: One of those sentences has already been removed and the other was made more consistent. See line 391 onwards*

- page 12, line 287 : The aerosol forcing is ... due to their ... : "aerosol" is singular, but "their" refers to something plural. So it sounds a bit strange.

Response: Replaced with "the shorter aerosol lifetime" - See line 397

- page 13, line 294 : "weakly positive in comparison with other forcings" : this sounds a bit strange.

Response: Removed "in comparison with other forcings" – See line 406

- page 13, line 298 : their combined : sounds strange. I would suggest "the combined".

Response: Done – See line 410

- page 13, Fig. 3 : maybe add the global mean values in the figure headings.

Response: Done in response to a previous comment

- page 14, line 308-309 : Despite ..., ... produce slightly different results. Isn't it what one should expect? As it is expected, I would not use "despite".

Response: Changed sentence as follows: "Statistical methods ensure that the model is not scientifically different on the different HPC platforms, but such duplicate experiments still produce slightly different results." – See line 425

- page 14, line 324 : 1.83 Wm-2 (but 1.82 Wm-2 in Table 3).

*This value has changed slightly due to the LAI bugfix but it is now consistent between this section (Line 441) and Table 3 (Page 14-15).*

 page 14, line 325-328 : It is informative to stress the absolute difference in CO2 ppm. However, maybe one could add that the relative change in CO2 concentration is more relevant for the forcing.

Response: Added that CO2 has a logarithmic dependency on concentration – See line 444

- page 15, line 337 : low-clouds -> low clouds.

Response: Corrected – line 458

- page 15, line 338 : piClim-CO2phys : the "2" should not be an index.

*Response: Corrected – See line 459*

- page 15, line 340-341 : low-level cloud -> low-level clouds.

Response: Corrected – See line 463

- page 15, line 337-341 : The first sentence gives the impression that the balance comes from two terms. However, the second sentence adds that the balance (or closure) comes from other terms.

Response: These have now been combined into one sentence. See line 449 onwards

- page 15, line 342 : found a much larger effect : this looks like a dramatic message, but it is just because the forcing is stronger (it is a 4xCO2 experiment). Therefore the reader is a bit in doubt whether he captures what the authors want to say.

*Response: Correct, the physiological forcing simply scales in the same way as the total CO2 ERF. All reference to it being stronger has been removed – See line 460.*

- page 15, line 354 : weren't -> were not.

Response: Done – See line 483

- page 16, line 366 : correlated to -> correlated with.

*Response: This section has been removed now, because similar responses were not found following the LAI bugfix – See line 496*

- page 16, line 378 : SAM : this abbreviation has not been defined.

Response: No longer referring to the SAM – See lines 509 onwards

- page 17, line 380-381 : "and a reduction of associated ... " : this sentence is slightly confusing, as through the reduction in high clouds, the outgoing LW radiation can be stronger again.

Response: This section has been removed, as indicated above – See lines 509 onwards.

- page 17, line 386 : near surface wind : maybe one can specify the altitude. Is it at 10 m?

Response: This figure has now been replaced with one from the piClim-HC experiment

- page 18, line 408 : are of the order -> are in the order.

Response: Corrected – See line 571

- page 18, line 417 : The major driver ... is greenhouse gases (GHGs) -> are [although I am not sure].

Response: Replaced with "major drivers ..... are" – See line 590

- page 18, line 417 : which is offset by aerosol : This is a slightly unlogical construction: I would think that forcings can be offset, but not GHGs.

Response: Now corrected this to specify that it is the forcing from GHGs that is offset – See line 590

- page 18, line 418 : key metrics -> key values.

Response: Changed – See line 591

- page 19, line 424 : "in e)" -> I would advance that slightly.

Response: Updated – See line 508 onwards

- page 19, line 433-437 : is the value 2.82 Wm-2 representing the 1850-2011 estimate? (does it already include the correction for going from 1750 to 1850?)

Response: It does. The sentence has been modified to make this clearer: "This latter estimate of 2.82  $W m^{-2}$  has been adjusted ...." – See line 164

- page 19, line 436 : e.g. CH4 : I assume this is also valid for CO2. Why not mentioning CO2?

Response: Added – See line 616

- page 19, line 440 : However, there is a discrepancy in ERF of 0.35 W m-2 ... which cannot ... -> However, the discrepancy of 0.35 Wm-2 ... cannot ...

*Response: This discrepancy no longer exists, following the re-runs of the simulations with the LAI bugfix. See line 619 onwards*

- page 20, line 443 : is this non-linearity similar to (or larger/smaller than) the one which one sees in the RF formulas of AR3 for N2O and CH4?

Response: No longer applicable, as indicated above.

- page 20, line 450 : The rapid adjustments (RA) ... includes -> The rapid adjustments (RAs) ... include.

Response: Corrected – See line 633

- page 20, line 453-454 : are there no other sources of uncertainty : the lifetime of aerosols? Their vertical profile?

Response: Now include other sources of uncertainty - See line 640

- page 20, line 454-455 : twice "sources" in this sentence.

Response: Altered - See line 644

- page 21, line480 : AEROCOM II -> AEROCOM Phase II.

Response: Now added – See line 672

- page 21, line 481 : The BC ERF was +0.32 W m-2 -> The BC ERF is +0.32 W/m2.

Response: Corrected – See line 672

- page 21, line 481 : and small negative offset -> and a small negative offset.

Response: Corrected – See line 673

- page 21, line 484 : in upper-level cloud -> in upper-level clouds.

Response: Changed – See line 676

- page 21, Fig 7b : should the orange bar represent -0.14 Wm-2? It looks larger.

Response: We checked and they are consistent

- page 23, line 525-526 : Are there two experiments with prescribed CDNC : piClimcontrol-fixedCDNC and piClim-aer-fixedCDNC? "By comparison with the main piClimaer" : shouldn't be added "and piClim-control"?

Response: These additional simulations were paired experiments. To add clarity, we altered the text as follows: "To further understand which processes contribute most to the aerosol ERF, a series of additional control and perturbation experiments were conducted ..."

and

"By comparison with the main piClim-aer/piClim-control experiment pair, this indicated a ...."

See lines 718-719, 723, and 730 of the marked-up manuscript

- page 23, line 532-533 : To complete the breakdown, ... : I assume this was on the main piClimcontrol and piClim-aer simulations, and not on the ones with fixed CDNC. This is maybe not so clear from the text.

*Response: Text changed to: "To complete the breakdown, the method in Ghan (2013) was applied to the main piClim-aer/piClim-control experiment pair to derive ..." – See line 730*

- page 24, line 566 : "over the South Pacific" : looking at the figure, it is not so clear that the South Pacific stands out more than other regions.

Response: *Response: This has been amended to Eastern tropical pacific. (See line 767 of the marked-up document)*

- page 24, line 573 : is increased -> are increased.

Response: Corrected (Line 775 of marked-up document)

- page 25, line 596 : tropics -> subtropics.

Response: Corrected (Line 803 of marked-up document)

- page 25, line 602-603 : due to experimental setup -> due to the experimental setup.

*Response: Part of that section has been removed to improve clarity (Lines 807-812 of marked-up document)*

- page 25, line 609 : 15 % -> 15% (there is a blanco space between "15" and "%" in the text).

Response: This is intentional and follows the recommendations of the Bureau international des poids et mesures. From the SI Brochure, §5.3.7 states "When it is used, a space separates the number and the symbol %." We use this consistently throughout the manuscript.

- page 26, Eq. 9, and line 630 : CH should not be written in italic.

*Response: Corrected although not showing as a tracked change – see Eqn. (9) on page 40 and see line 843 of the marked-up document*

- page 26, line 630 : where ... is ... the concentrations -> concentration.

Response: Corrected - see line 841 of the marked-up document

- page 26, line 631 : there is apparently no blanco space after "piClim-control,".

*Response: Corrected although not showing as a tracked change – see line 843 of the marked-up document*

- page 28, line 669-670 : I suggest to write "hydroxyl radical" and "nitrate radical".

Response: Added - see line 887 of the marked-up document

- page 28, line 674: it doesn't -> it does not.

Response: Corrected - see line 893 of the marked-up document

 page 28, line 680-681 : maybe add at which altitude. It is mentioned in the caption of the figure, but it is maybe informative to mention it also in the text.

Response: Added - see line 898 of the marked-up document

- page 29, line 699 : Global distributions of the multi-annual distributions : twice "distributions".

Response: Corrected - see line 919 of the marked-up document

- page 29, line 707 : is changes to OH -> are changes to OH.

Response: Corrected - see line 929 of the marked-up document

- page 30, line 710-714 : how is the ari from NOx calculated? How in general are the ari/aci from NOx and VOC calculated?

Response: We use the Ghan (2013) approach to calculate the aerosol instantaneous radiative effect (IRE) in a simulation by taking the difference in radiative fluxes between two radiation calls, i.e., one call including aerosols (F) and the other without aerosols (Fclean), i.e., F - Fclean. The aerosol IRF is then calculated from the difference in the aerosol IRE between 2 simulations, where one simulation is piClim-control and the other is, for example, piClim-NOx, i.e.,  $\Delta$ (F - Fclean). We also use this approach to estimate the forcing due to changes in the cloud radiative effect (CRE) by calculating the CRE in each simulation using "clean" radiation calls and the difference between all-sky and clear-sky fluxes, i.e., Fclean – Fcs, clean. The cloud forcing is then calculated as the difference in the CRE between 2 simulations, i.e.,  $\Delta$ (F clean – Fcs, clean). This breakdown is detailed in Eqns. (6)-(8) on pages 12-13 of the marked-up document.

- page 30, line 731 : as was the case with NOx : is meant here that the same mechanisms are active related to OH? As the change of OH is however opposite, the forcing is also opposite.

*Response: Yes, the reviewer is correct. It is the same mechanism but the effect is in the opposite sense for COC/CO as for NOx. This has been re-written as follows:*

"Via the same mechanisms as was the case with NOx, ...." - see line 954 of the marked-up document

- page 30, line 735 : and CO2 -> and CO2 response.

Response: Corrected - see line 959 of the marked-up document

 page 30, line 735-739 : maybe one can mention explicitly that part of this message was already given earlier (on page 30, line 718-720).

Response: It wasn't clear that we could do this without affecting the flow of that paragraph so, we have opted not to change the text.

- page 31, line 753-754 : Is this true : are these the two main reasons (the fact that there are interactions, and the fact that there is non-linearity)?

Response: This has been changed as follows: "This is due to the uncertainty in the individual forcings (e.g., Bellouin et al., 2019) but the interaction between individual forcings, as well as the non-linear response of climate feedbacks due to aerosol-cloud interactions (Feichter et al., 2004; Deng et al., 2016; Collins et al., 2017; Shim et al., 2019) may play a role." - see line 978 onwards of the markedup document

page 31, line 764 : closely correlated : the correlation seems not that high (-0.44). In addition, from
 Figs. 12a and 12b it is not so easy to see that there is an anti-correlation. So I would not write
 "closely" correlated.

Response: Removed "closely" - see line 992 of the marked-up document

- page 31, line 767 : "may be" : can this not be said with more certainty? It seems like a sound explanation.

Response: Changed, as suggested - see line 997 of the marked-up document

- page 31, line 770 : with good correlation with -> correlating well with.

*Response: This has been removed due to an error with the high-level cloud fraction diagnostic from the model simulations - see line 1000 onwards of the marked-up document*

- page 31, line 771-772 : cloud -> clouds (twice).

*Response: This whole sentence has now been removed, as indicated above - see line 998 onwards of the marked-up document*

- page 32, line 778 : ", (h)" -> ", and (h)".

*Response: The number of panels in Figure 12 has now changed but the caption does follow this recommendation, albeit no longer for panel (h) - see page 48 of the marked-up document*

- page 33, line 782 : This study also attempts ... : it seems to be a bit a sudden introduction. Maybe one can first introduce the topic in general, and then say that it is also a focus of this study.

Response: This has now been addressed by including more of an introduction and referring to what has been done in other sections, e.g., GHGs, aerosols, and O3 precursor gases. See line 1018 onwards of the marked-up document

- page 33, line 782 : interaction between the aerosol and O3 forcings : is it really an interaction between the forcings which causes this?

Response: This has been corrected to say that the non-linear interactions between chemistry and aerosol may lead to non-linearities in the forcings - See lines 1021 onwards of the marked-up document

- page 33, line 784 : particularly in the net CS components : it is a bit strange to focus on the the net CS component, as the difference in SE CRE is even bigger.

*Response: Agreed, this has been changed and the non-linearity in the SW CRE explicitly mentioned. See line 1031 onwards of the marked-up document*

- page 33, line 784 : Firstly we calculate the aerosol IRFs. How are the IRFs calculated?

Response: We now refer to Eqn. (7) - See line 1025 of the marked-up document

- page 33, line 792 (not shown) and Fig. 13: it might be interesting to add a figure panel with the differences in O3 profiles.

Response: Given there is little impact on the LWcs' component of the ERF, we do not think that the differences in O3 are contributing to the non-linearity. As a result, we do not include a plot of differences in O3 profiles.

 - page 34, line 805-807 : when reading this sentence, is seems that more attention is given to the +0.07 Wm-2 effect (i.e., the change from -0.39 to -0.32 Wm-2), than on the value of -0.32 Wm-2 itself. Response: The apparent strong land use ERF in UKESM1 did indeed warrant more attention. However, this somewhat changed as a result of the new simulations following the discovery of the bug in the LAI seasonal cycle. Nevertheless, we now include some discussion of the different components to the newly quantified ERF of -0.17  $\pm$  0.04 W m-2. See lines 1057 onwards

 page 34, line 817 : and its cloud-free, aerosol-free, component -> and its cloud-free and aerosolfree component.

Response: Done - See line 1084 of the marked-up document

- page 35, line 822 : The seasonality of ... cause -> causes.

Response: Done - See line 1092 of the marked-up document

- page 35, line 830 : hist -> historical (the official name of this CMIP6 experiment).

Response: Done - See line 1102 of the marked-up document

- page 35, line 840 : As summarized above ... : maybe this is not a very good introduction - I don't think that "summarized" is the best way to refer to earlier text.

Response: Changed - See line 1113 of the marked-up document

- page 35, line 848-850 : What is the motivation for this sentence? Why is it mentioned that ozone is prescribed, but not, e.g., that oxidants like OH, NO3, and H2O2 are sometimes prescribed in models?

Response: Oxidants are now also mentioned - See line 1124 of the marked-up document

- page 36, line 857 : AF19 : this abbreviation is used only three times - I would think that it does not make so much sense to define it.

Response: Abbreviation now removed – See lines 1132 onwards of the marked-up document

- page 36, line 866 : what is GC3.1? Should probably be HadGEM-GC3.1.

Response: Now corrected – See line 1142 of the marked-up document

- page 36, line 866 : (well within the uncertainty range) : maybe one can add which uncertainty range is meant.

Response: Added - See line 1143 of the marked-up document

SECTION 5

- page 37, line 887 : paper -> study.

*Response: Changed – See line 1166 of the marked-up document*

- page 37, line 900 : may result : cannot it be expressed more firmly?

Response: No longer applicable with the updated simulations – See line 1182

- page 37, line 900 : coming from cloud top -> coming from cloud tops.

Response: No longer applicable with the updated simulations

page 38, line 942 : reproduces -> reproduces well.

Response: Added - See line 1123 of the marked-up document

**- page 38, line 949 : we consider -> we suggest.**

Response: Changed - See line 1242 of the marked-up document

**Assessment of pre-industrial to present-day anthropogenic climate forcing in UKESM1**

Fiona M. O'Connor1, N. Luke Abraham2,3, Mohit Dalvi1, Gerd Folberth1, Paul Griffiths2,3, Catherine Hardacre1, Ben T. Johnson1, Ron Kahana1, James Keeble2,3, Byeonghyeon Kim4, Olaf Morgenstern5, Jane P. Mulcahy1, Mark G. Richardson6, Eddy Robertson1, Jeongbyn Seo4, Sungbo Shim4, Joao C. Teixeira1, Steven Turnock1, Jonny Williams5, Andy Wiltshire1, and Guang Zeng5

1Met Office, Exeter, United Kingdom

5

[revised manuscript text omitted]
 O3 precursors and the contribution from stratospheric O3 depletion on the modelled changes in tropospheric O3 (e.g., Søvde et al., 2011; 2012).
Despite these uncertainties in estimating tropospheric O3 RF, the even larger uncertainty in aerosol forcing (Myhre et al., 2013a; Bellouin et al., 20202019) accounts for the majority of the uncertainty in the total anthropogenic forcing.

Aerosol forcing involves a wide range of physical processes. These include (i) direct changes to the radiation budget through scattering and absorption of both shortwave (SW) and longwave (LW) radiation (e.g., Haywood and Boucher, 2000), (ii)

- 110 indirect impacts on the radiation budget by changing the microphysical properties of clouds (Twomey et al., 1977), and (iii) changes in the distribution of cloud cover or condensate that follow on from perturbations in cloud microphysics (Albrecht 1989) or radiative heating by aerosols (Hansen et al., 1997). Direct aerosol RF can be calculated using offline radiative transfer models in a similar manner to greenhouse gas (GHG) and O3 forcing, whereas assessing impacts of aerosols on clouds requires simulations in atmospheric models. The 5th assessment report (AR5) of the IPCC recommended the effective radiative forcing
- (ERF) framework as a suitable metric for assessing the overall aerosol forcing as it enables the more complex cloud impacts to be evaluated as part of the climate's rapid adjustments (RAs) (Myhre et al., 2013a). To simplify terminology, AR5 also made a clear distinction between components of the forcing driven by aerosol-radiation interactions (ari; i.e., the direct or IRF) and aerosol-cloud interactions (aci) (that include all indirect or semi-direct cloud-related forcings). Despite wide-ranging and on-going research, the role of aerosols remains the leading source of uncertainty in estimates of climate forcing, due to the difficulty in constraining the sensitivity of clouds to changing microphysical processes (Bellouin et al., 2020).
- In the case of land use, RF estimates have been made using single general circulation model (GCM) simulations with a doublecall to the radiation scheme (e.g., Betts et al., 2007) or by comparing paired simulations that include RAs (e.g., Andrews et al., 2016). However, the choice of RF calculation is not the major source of differences in RF estimates. Similar to O3, uncertainty in PI land cover is a major source of uncertainty in land use RF (e.g., de Noblet-Ducoudré et al., 2012). Historically, deforestation has been the dominant type of land use change, and this causes a positive RF due to increased carbon dioxide (CO2) emissions and a negative RF due to increased surface albedo. Here, we include the effects of land use CO2 emissions in the CO2 ERF estimates and the land use ERF is due to biophysical changes, predominately albedo. Deforestation has a much larger effect on albedo in snowy regions and model biases in snow cover also contribute to uncertainty in land use RF (Pitman
- 130 et al., 2011). Land use RF estimates also vary due to different time periods being considered (Myhre et al., 2013a), because, unlike many other forcing agents, there was substantial land use change before the industrial revolution.
  - 4

---

## Author Response (AR2)

Author Responses to the Reviews Received from acp-2019-1152: "Assessment of pre-industrial to present-day anthropogenic climate forcing in UKESM1" by F. M. O'Connor

Dear Editor,

On behalf of my co-authors, I would also like to thank the two anonymous reviewers for their positive and constructive reviews of the revised manuscript.

In response, we iterate the reviewers' comment in black with grey shading and our response to their comment will be *black italics*. And we append a revised marked-up manuscript to illustrate the changes implemented.

Regards,

Fiona O'Connor (on behalf of all co-authors)

**Response to Reviewer #1:**

This is an excellent revision and I thank the authors for such a positive response to the review comments.

*Response: Thank you for your positive comments and for your detailed constructive feedback on the original version of the manuscript.*

I am very happy to recommend acceptance of the paper. It can be accepted as is, but I have a few rather minor comments/suggestions that the authors may wish to attend to.

*Response: Thank you*

There is a further matter that I think requires an Editor decision. The handling of the unpublished Morgenstern paper is much better in the new version, especially in acknowledging that UKESM may be an outlier. Nevertheless, unlike (I think) all the other unpublished papers in the new version, it is not available in Discussion form, or available on a preprint service (e.g. as is the Pincus paper). I wonder if it should be insisted that it be made available in this form, if acceptance remains an issue or is likely to be significantly delayed.

*Response: The Morgenstern et al. paper has now been published and the reference updated in the manuscript (Lines 1478-1479).*

Comments

37: "it is outside the range of previous estimates, and". Maybe indicate which way it is outside the range? More negative or less negative?

*Response: We have now added that the land use ERF is more negative than the range of previous estimates (Line 37).*

41: RA – since this abbreviation is used only once in the abstract, I suggest not using it all.

*Response: Now removed (Lines 18 & 42).*

200: Figure caption. "HC" in header – It seems that HC is being used before being defined. As it is not in any way a standard acronym (I still think "hydrocarbon" (or "Hadley Centre!")) I suggest spelling this out in the caption.

*Response: The table caption now includes an explanatory note for the piClim-HC experiment (Lines 204-205) and further clarification has been included in the main body of the text (Line 237).*

419: N2O forcing. With apologies, I might have spotted this in the first version. There is a further possibility to explain the higher ERF/RF here, that goes beyond the RF and ERF. The Etminan expressions crudely account for the changing concentrations of methane (and CO2) as overlapping gases in their parameter M-bar (and C-bar). Now in experiment piClim-N2O, the background concentrations of CH4 and CO2 are unchanging, and hence it would be more appropriate to compare the UKESM forcing against the Etminan forcing with M-bar and C-bar concentrations specified at their 1850 levels. My guesstimate is this would only increase the N2O forcing by 5-10%, so closing the gap a bit, and the authors may, in any case, already have accounted for this. The same applies for the other forcings (CO2 and CH4) calculated from the expressions in Etminan, but the effects of these changing overlaps must be much smaller

*Response: Using Year-1850 concentrations for both M0 and M (for the calculation of Mbar) for methane (and likewise for CO2) instead of Year-1850 and Year-2014 concentrations makes a difference of less than 2 % in the estimate of the Year-2014 N2O RF using the Etiminan et al. (2016) expression. Indeed, when the Etiminan et al. RF estimate is rounded to two decimal figures, as is used in the manuscript, the difference is no longer discernible. As a result, no change to the existing text has been made.*

463: "having a stronger global O3 decline" – given my comment above about the (non) availability of the Morgenstern paper, I suggest that this statement could be made more quantitative – how much stronger is the decline? And similarly, is that decline significantly stronger than what are (compared to many constituents) quite good observations of that decline?

*Response: As indicated above, the Morgenstern et al. (2020) is now published and the Keeble et al. (2020) discussion paper is also available. For example, Keeble et al. shows a comparison of total column ozone trends from 1980-2000 for different models and different observational datasets. UKESM1 has a global mean total column ozone trend over the 1980-2000 period that is stronger than the observed trend and that from other models by a factor of 1.4 or more. This has been added to the manuscript (Lines 470-471).*

898: typo "lland"

*Response: Corrected (Line 909)*

900: "stronger than observed" – perhaps some quantitative hint would be useful here?

*Response: We have now added that the surface SW response to total deforestation was generally 5 W m$^{-2}$ too large in HadGEM2-ES in comparison with observations (Lines 911-912).*

909: typo "andaerosol"

*Response: Corrected (Line 922)*

1063: assuming that this is the UK body, then "National Environmental Research Council" is doubly wrong. It is "Natural" and "Environment" (it is correct later in the Acknowledgements,

*Response: Thank you for spotting this double error – Now corrected (Line 1077)*

**Response to Reviewer #3:**

I have one principle comment :
The fact that the differences shown in global maps which are not statistically significant have been removed from the plots ("masked out in white"). I do not think this is a good approach.

*Response: The following figures (and captions) were updated, such that areas of the globe that did not show a statistically significant ERF (or its components) at the 95 % confidence level were stippled instead of being masked with white:*

*Fig. 3 (Page 18), Fig. 4 (Page 21), Fig. 5 (Page 23), Fig. 6 (Page 25), Fig. 8 (Page 28), Fig. 12 (Page 39), and Fig. 14 (Page 42)*

Then I have a list of small comments and suggestions (on language, typos, different ERF values in text and tables, ...) :
Eqs (4), (6) and (7) : formulas which consist out of several lines, only need a "," or "." after the last line. So Eqs. (4), (6) and (7) do not need a "," at the end of the line.

*Response: Done (Lines 300, 309, and 310)*

"Year" : "Year" is often written in capital letters - I do not think that is necessary

*Response: All occurrences of the word "year" are now in lower case (Lines 21, 37, 163, 187, 194, and 210; Table 3 on page 13; Lines 340, 348, 374, 375, 480, 580, 826, 827, 852, and 916)*

line 18-19 : the combination "by ..., it ..." : I would possibly skip both "by" and "it"

*Response: Done as suggested (Lines 18-19)*

Table 1 : the expression "4x1850" is a bit strange (but maybe ok to keep it)

*Response: Now changed to "4x 1850 value" in Table 1 (Page 7 and 8)*

line 210-211 : possibly add "and" before "surface seawater"

*Response: Added (Line 212)*

line 241 : "LBC, Finally" : "," -> "."

*Response: Thank you for spotting this. Now corrected (Line 244)*

line 278 : maybe add "the" before "National Institute of Meteorological Science"

*Response: Added (Line 270)*

line 318 : (ERFcs' ) : white space can be removed

*Response: Done (Line 320)*

line 320 : modelled-derived -> model-derived

*Response: Done (Line 322)*

line 324 : considered -> taken into account

*Response: Done (Lines 326-327)*

Table 3 : LWcs', SWcs', NETcs' : I suggest to write also the subscript straight (as, e.g., on line 399)

*Response: Corrected in Table 3 (Page 13) and elsewhere in the manuscript (e.g., Table 4 on Page 18)*

Table 3 : on the line with GHG : I would write $CO_2$, $N_2O$, and $CH_4$ with lowered indices

*Response: Done (Table 3, Page 14)*

Table 3 : on the line with O3 precursors : I would write $O_3$ with lowered indices

*Response: Done (Table 3, Page 14)*

line 361 : "signal.Further" : white space needed between "signal." and "Further"

*Response: Done (Line 364)*

Figure 3, caption : the references to panels like "a)" and "b)" appear here after the description. In most other figures they appear before the description.

*Response: Order changed (Lines 375-376)*

line 397 : Etminan et al., (2016) : "," can be removed

*Response: Done (Line 402)*

line 409 : "cf." : I would not used "cf." here

*Response: Replaced with "and" (Line 414)*

line 411 : 0.14+/-0.02, whereas 0.16 in Table 3

*Response: Corrected (Line 416)*

line 421 : 0.25+/-0.03, whereas +/- 0.04 in Table 3

*Response: Corrected (Line 426)*

line 424 : "of0.09" : white space needed between "of" and "0.09"

*Response: White space already exists although perhaps not so obvious (Line 429) – no change implemented*

line 425 : ( 0.25 : whte space between "(" and "0.25"

*Response: Done (Line 430)*

line 425 : 0.25 +/- 0.03, whereas +/- 0.04 in Table 3

*Response: Corrected (Line 430)*

line 473 : 0.74 +/- 0.03, whereas +/- 0.02 in Table 3

*Response: Corrected (Line 481)*

line 476 : 0.12 +/- 0.03, whereas +/- 0.02 in Table 3

*Response: Corrected (Line 485)*

line 481 : "and are due to indirect effects" -> "and is due to indirect effects"

*Response: Corrected (Line 489)*

line 509 : aren't the SW CRE and LW CRE values interchanged compared to Table 3?

*Response: Indeed they were interchanged and were inconsistent with the text – Now corrected (Lines 518-519)*

line 516 : "CH4," : the "," is also written as a subscript

*Response: Corrected (Line 525)*

line 562 : -0.46 +/- 0.01 : the value in the figure looks more negative than just -0.46 (might be also related to the resolution of the figure)

*Response: Thank you for spotting this. Having gone back over the analysis, we found that the plot is correct but that the value of -0.46 in the text was not. We have now corrected this (-0.49), making it consistent with Fig. 7. (Line 571)*

line 601 : "And (iii) Internal" : "Internal" should not start with a capital letter

*Response: Corrected (Line 611)*

line 606-621 : the numbers in this paragraph do not have uncertainty estimates : -0.70 (line 612), -0.15 (line 615), 0.05 (line 619) and -0.99 (line 620)

*Response: These have now been added (Lines 622, 623, 625, 629, and 630)*

line 636 : 0.21 +/- 0.04 : "+/- 0.04" is written in red

*Response: Corrected (Line 646)*

line 688 : "toODSs" : white space needed between "to" and "ODSs"

*Response: Corrected (Line 698)*

Figure 10, caption : piClim-NOX -> piClim-NOx

*Response: Corrected (Line 713)*

line 710 : "take account of" : I would think that "take into account" or "account for" sounds better

*Response: Changed as suggested (Line 720)*

line 794 : 0.33 +/- 0.03 whereas 0.04 in Table 3

*Response: Corrected (Line 804)*

line 803 : Thornhill et al. (2020) also shows ... : I would write "show" (plural)

*Response: Changed as suggested (Line 813)*

line 840-843 : after the "... three ... are discussed: ...", one would except that all 3 experiments follow in the second part of the sentence. Here only one experiment follows, and the next seperate sentence treats the two other experiments.

*Response: This has now been changed (Lines 851-852)*

line 875 : twice "-1.0" -> twice "-1.00" (to be in agreement with the precision of the uncertainty estimate)

*Response: Done (Line 886)*

line 882 : "Figure 13" -> "Fig. 13"

*Response: Done (Line 893)*

line 888 : "lland" -> "land"

*Response: Done (Line 909)*

line 909 : "andaerosol-free" : white space between "and" and "aerosol-free"

*Response: Done (Line 922)*

line 914 : reference "Noblet-Ducoudre et al. (2012)" is not found in reference list

*Response: Added (Lines 1513-1515)*

line 915 : "increased from about 1850-1970 in North America and from 1900-1980 in ..." : I suggest to write : "increased from about 1850 to 1970 in North America and from 1900 to 1980 in ..." or "increased over the period 1850-1970 in North America and over the period 1900-1980 in ..."

*Response: Changed as suggested (Lines 928-929)*

line 908-927 : the numbers in these paragraphs have no uncertainty estimate : -0.28 (line 918), -0.08 (line 918), and -0.20 (line 927).

*Response: Uncertainties have now been added (Lines 930 and 931)*

line 1027 : what's -> what is

*Response: Changed (Line 1040)*

line 1044 : "with a fully coupled ESM" : if "fully-coupled" refers to coupling with the ocean, then this sentence is maybe a bit strange : the quantification of the ERf has happened with the atmosphere-only configuration of the model

*Response: Reference to fully-coupled now removed (Line 1058)*

[revised manuscript text omitted]